# Hard labels sampled from sparse targets mislead rotation invariant algorithms

Avrajit Ghosh [1]  Bin Yu [1]  Manfred K. Warmuth [2] [*]  Peter Bartlett [3] [*]

## Abstract

One of the most common machine learning setups is logistic regression. In many classification models, including neural networks, the final prediction is obtained by applying a logistic link function to a linear score. In binary logistic regression, the feedback can be either soft labels, corresponding to the true conditional probability of the data (as in distillation), or sampled hard labels (taking values $\pm 1$). We point out a fundamental problem that arises even in a particularly favorable setting, where the goal is to learn a noise-free soft target of the form $\sigma(\mathbf{x}^\top \mathbf{w}^\star)$. In the over-constrained case (i.e. the number of samples $n$ exceeds the input dimension $d$) with examples $(\mathbf{x}_i, \sigma(\mathbf{x}_i^\top \mathbf{w}^\star))$, it is sufficient to recover $\mathbf{w}^\star$ and hence achieve the Bayes risk. However, we prove that when the examples are labeled by hard labels $y_i$ sampled from the same conditional distribution $\sigma(\mathbf{x}_i^\top \mathbf{w}^\star)$ and $\mathbf{w}^\star$ is $s$-sparse, then rotation-invariant algorithms are provably suboptimal: they incur an excess risk $\Omega\left(\frac{d-1}{n}\right)$, while there are simple non-rotation invariant algorithms with excess risk $O(\frac{s \log d}{n})$. The simplest rotation invariant algorithm is gradient descent on the logistic loss (with early stopping). A simple non-rotation-invariant algorithm for sparse targets that achieves the above upper bounds uses gradient descent on the weights $u_i, v_i$, where now the linear weight $w_i$ is reparameterized as $u_i v_i$.

## 1. Introduction

A fundamental objective in machine learning is to learn a task from finite samples efficiently. One such task is to estimate a vector $\mathbf{w}^\star \in \mathbb{R}^d$ from a finite collection of paired data $\{(\mathbf{x}_i, y_i)\}_{i=1}^n$, where training instances $\mathbf{x}_i \in \mathbb{R}^d$ and the labels are generated as a function of the linear score

---
[*]Equal contribution  [1]University of California, Berkeley. [2]Google Research [3]Google Deepmind. Correspondence to: Avrajit Ghosh <ghoshavr@berkeley.edu>.

*Proceedings of the 43rd International Conference on Machine Learning*, Seoul, South Korea. PMLR 306, 2026. Copyright 2026 by the author(s).

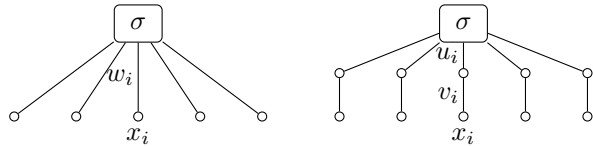

*Figure 1.* A sigmoided linear neuron (left) and its "spindlified" reparameterization (right). Gradient Descent (GD) on the left network is rotation invariant, whereas GD on the right network is not.

$(\mathbf{x}_i^\top \mathbf{w}^\star)$. Much prior work has focused on the underconstrained case ($n \leq d$), particularly when the label depends on a single feature of $\mathbf{x}_i$ or on a sparse linear combination of features. In this setting, rotation-invariant algorithms are known to perform poorly compared to non-rotation-invariant methods that are biased toward sparse solutions (Warmuth & Vishwanathan, 2005; Ng, 2004; Li et al., 2021; Warmuth et al., 2021). In some cases, it has even been shown that embedding the instances via a fixed feature map $\Phi(\mathbf{x}_i)$ does not alleviate this limitation for rotation-invariant algorithms (Warmuth & Vishwanathan, 2005). In this paper, we focus on the overconstrained case ($n \geq d$). Surprisingly, even in this setting there remains a performance gap between rotation-invariant and non-rotation-invariant algorithms for sparse linear regression with additive Gaussian noise. Recent work (Warmuth et al., 2025) has shown that when noise is added to a sparse linear model, then rotation invariant algorithms are prevented from recovering sparse targets efficiently. Canonical examples of rotation-invariant algorithms are neural networks with a fully connected input layer, initialized from a rotation-invariant distribution and trained using gradient descent. In contrast, when each linear weight $w_i$ is expressed as a product of two parameters $w_i = u_i v_i$ (Figure-1), then GD on the resulting network is not rotation invariant any more.

We focus on a more important setting: sparse logistic regression. Logistic modeling is the most common final prediction stage in neural networks for classification, and understanding how models trained with gradient descent learn sparse targets in this setting is therefore of central interest. We consider the simplest instance of this problem, namely the binary generalized linear model, where the conditional distribution of the label is given by a logistic sigmoid link.

Specifically, labels are generated according to

$$\mathbb{P}(y_i = 1 \mid \mathbf{x}_i) = \sigma(\mathbf{x}_i^\top \mathbf{w}^\star). \tag{1}$$

We assume that the inputs are drawn i.i.d. from an isotropic Gaussian distribution, $\mathbf{x} \sim \mathcal{N}(\mathbf{0}, \mathbf{I})$, the target vector $\mathbf{w}^\star$ has finite norm and is $s$-sparse, so that the probabilities $\sigma(\mathbf{x}^\top \mathbf{w})$ are bounded away from 0 and 1.

In the overconstrained case ($n > d$), learning may proceed in two ways: (a) The learner uses the true conditional probabilities (*soft-labels*) $\sigma(\mathbf{x}_i^\top \mathbf{w}^\star)$ as targets. In this case, rotation-invariant algorithms are sufficient to recover the sparse target vector $\mathbf{w}^\star$. (b) The learner observes only hard labels taking values $\pm 1$, sampled from the same distribution (1). We show that this distinction is important. When training relies only on hard labels, rotation-invariant algorithms are provably suboptimal: no such algorithm can achieve the Bayes risk, and any rotation-invariant procedure incurs an excess risk of order $\Omega\left(\frac{d-1}{n}\right)$. By contrast, a simple non-rotation-invariant algorithm (gradient descent on the *spindly network* of Figure 1, left) achieves a much smaller excess risk of order $O\left(\frac{s \log d}{n}\right)$.

This gap is non-trivial because the underlying label-generating model is noise-free, and the target conditional probability $\sigma(\mathbf{x}^\top \mathbf{w}^\star)$ contains sufficient information to recover $\mathbf{w}^\star$ in the overconstrained case (Proposition-2.1). Nevertheless, when the learner accesses this target only through sampled hard labels, the sampling process introduces enough randomness about $\mathbf{w}^\star$, and the performance gap arises without any additional noise.

Our techniques for obtaining a lower bound for rotation invariant algorithms and an upper bound for non-rotation invariant algorithms are novel for logistic loss and these results do not follow directly from the analogous results for square loss (Warmuth et al., 2025). For example, to establish the excess risk lower-bound, we must resort to geometric properties of the logistic loss to analyze posterior concentration on a sphere. For the excess risk upper bound, we develop a new *state-dependent* Riccati-type ODE describing the dynamics of a non-rotation invariant algorithm under logistic loss and use an ODE enveloping argument to separate signal and noise coordinates.

**Notation:** For positive functions $f(x)$ and $g(x)$, we write $f(x) = O(g(x))$ if there exists a constant $C > 0$ such that $f(x) \leq C g(x)$ for all sufficiently large $x$, and $f(x) = \Omega(g(x))$ if there exists a constant $c > 0$ such that $f(x) \geq c g(x)$ for all sufficiently large $x$. The notation $f(x) = \Theta(g(x))$ means both $f(x) = O(g(x))$ and $f(x) = \Omega(g(x))$. We use $f(x) \lesssim g(x)$ to denote $f(x) \leq C g(x)$ for an absolute constant $C > 0$, and we write $f(x) \asymp g(x)$ when both $f(x) \lesssim g(x)$ and $g(x) \lesssim f(x)$ hold, meaning the two functions are comparable up to constant factors. For

vectors $\mathbf{u}, \mathbf{v} \in \mathbb{R}^d$, we denote by $\mathbf{u} \odot \mathbf{v}$ their Hadamard (elementwise) product.

## 2. Preliminaries

*Well-specified Data model:* Assume data $\mathcal{D} := \{(\mathbf{x}_i, y_i)\}_{i=1}^n \in \mathbb{R}^d \times \{-1, +1\}$ are drawn i.i.d. from a Gaussian distribution $\mathbf{x}_i \sim \mathcal{N}(\mathbf{0}, \mathbf{I})$ and labels are generated according to the true conditional probability (1):

$$\mathbb{P}(y_i = 1 \mid \mathbf{x}_i) = \sigma(\mathbf{x}_i^\top \mathbf{w}^\star) := \frac{1}{1 + e^{-\mathbf{x}_i^\top \mathbf{w}^\star}}. \tag{2}$$

The oracle parameter $\mathbf{w}^\star \in \mathbb{R}^d$ is $s$-sparse and has unit Euclidean norm, that is, $\|\mathbf{w}^\star\|_0 = s < d$ and $\|\mathbf{w}^\star\|_2 = 1$ (without loss of generality). If the input vector lies entirely in the inactive subspace of $\mathbf{w}^\star$, the conditional label distribution is $\mathbb{P}(y = 1 \mid \mathbf{x}) = 1/2$, so the labels contain no information; effective learning therefore requires ignoring these directions and focusing on recovering the signal on the sparse active support.

*Population risk:* For parameter $\mathbf{w} \in \mathbb{R}^d$, the population *logistic* risk is defined by the negative log-likelihood of (2)

$$\mathcal{L}(\mathbf{w}) := \mathbb{E}\big[\ell(y\langle \mathbf{x}, \mathbf{w}\rangle)\big], \quad \ell(t) := \log(1 + e^{-t}), \tag{3}$$

where the expectation is over the joint distribution of $(\mathbf{x}, y)$. The population risk implicitly has access to the true conditional probability (2) since it is an expectation over $\mathbb{P}(y|\mathbf{x})$. Under the well-specified sparse data model in (2), $\mathcal{L}(\mathbf{w})$ is strictly convex and admits a unique and finite global minimizer at $\mathbf{w}^\star$. The Bayes risk is thus $\mathcal{L}(\mathbf{w}^\star)$.

*Empirical risk:* The empirical risk is the average over the $n$ observed samples in $\mathcal{D}$:

$$\widehat{\mathcal{L}}(\mathbf{w}) := \frac{1}{n} \sum_{i=1}^n \ell(y_i \langle \mathbf{x}_i, \mathbf{w}\rangle). \tag{4}$$

We also define a *soft-label* empirical risk, when we have access to the true conditional distribution of the label given each input.

$$\widehat{\mathcal{L}}_{\text{soft}}(\boldsymbol{w}) := \frac{1}{n} \sum_{i=1}^n \mathbb{E}_{y|X=\boldsymbol{x}_i}\big[\ell(y\langle \boldsymbol{x}_i, \boldsymbol{w}\rangle)\big]. \tag{5}$$

Under this data model assumption and loss definition, we state the following proposition:

**Proposition 2.1.** *If the design matrix* $\mathbf{X} := (\mathbf{x}_1, \mathbf{x}_2, .., \mathbf{x}_n)^T$ *has full column rank and* $n > d$, *then the empirical soft-label risk* $\widehat{\mathcal{L}}_{\text{soft}}(\boldsymbol{w})$ (5) *has the same unique global minimizer* $\mathbf{w}^\star$ *as the population risk.*

We defer the proof to C.1 in the appendix. At a high level, the argument is straightforward. The empirical soft-label

risk has a stationary point at $\mathbf{w}^\star$, and full column rank of the design matrix implies strict convexity further implying $\mathbf{w}^\star$ is the unique global minimizer. Proposition C.1 shows that, in the over-constrained case, access to soft labels makes learning trivial: minimizing the empirical risk uniquely recovers the oracle $\mathbf{w}^\star$.

In contrast, when only sampled hard labels are observed, the empirical risk (4) behaves differently. Even in the over-constrained case, the randomness due to the label sampling alters the landscape and the unique global minimizer is not at the oracle $\mathbf{w}^\star$. Proposition 2.2 formalizes this distinction:

**Proposition 2.2.** *Under the well-specified sparse logistic model* (2), *with* $n > d$, *let* $\mathbf{X} := (\mathbf{x}_1, \mathbf{x}_2, \ldots, \mathbf{x}_n)^\top$ *denote the design matrix.*

1. *There exist constants* $c_1, c_2 > 0$, *depending only on* $\|\mathbf{w}^\star\|$, *such that*

$$\mathbb{P}(\{(\mathbf{x}_i, y_i)\}_{i=1}^n \text{ is linearly separable}) \leq c_1 e^{-c_2 n}.$$

2. *On the event that the data is not linearly separable,* $\widehat{\mathcal{L}}(\mathbf{w})$ *is coercive, i.e.*

$$\|\mathbf{w}\| \to \infty \quad \Longrightarrow \quad \widehat{\mathcal{L}}(\mathbf{w}) \to \infty,$$

*and if* $\mathbf{X}$ *has full column rank, then* $\widehat{\mathcal{L}}$ *is strictly convex and admits a unique global minimizer, different than* $\mathbf{w}^*$.

We defer the full proof to Appendix C.2 and provide a brief proof sketch showing that the data is non-separable when $n > d$. Since $\|\mathbf{w}^\star\|$ is finite, the logistic model assigns nonzero probability to both labels on a set of inputs that occurs with positive probability, so the labels are random rather than deterministic. When $n > d$, many sampled labels therefore disagree with any fixed linear separator, and because the number of linear separations in $\mathbb{R}^d$ grows only polynomially with $n$, a union bound implies that, with high probability, no separator fits all labels and the data are not linearly separable.

When the data are not separable, the empirical logistic loss diverges along every ray, which ensures coercivity. If the design matrix has full column rank, the loss is strictly convex and therefore admits a unique global minimizer. Moreover, with sampled hard labels, $\mathbf{w}^\star$ fails the first order optimality condition $\nabla \widehat{\mathcal{L}}(\mathbf{w}^\star) = \mathbf{0}$. Since strict convexity implies the minimizer is the unique stationary point, it follows that the unique global minimizer differs from $\mathbf{w}^\star$.

*Gradient flow on single layer:* The empirical risk $\widehat{\mathcal{L}}(\mathbf{w})$ is minimized using gradient flow.

$$\dot{\mathbf{w}}(t) = -\nabla \widehat{\mathcal{L}}(\mathbf{w}(t)), \qquad \mathbf{w}(0) = \mathbf{0}. \qquad (6)$$

From Proposition 2.2, since the loss is strictly convex and has a unique global minimizer, this flow is guaranteed to

reach the unique minimum (by LaSalle invariance principle (Khalil & Grizzle, 2002)). In Section-3.2 we show that this flow trajectory is rotation-invariant.

*Gradient flow on the spindly network:* $\widehat{\mathcal{L}}(\mathbf{w})$ is minimized using a Hadamard-factored parameterization $\mathbf{w}(t) = \mathbf{u}(t) \odot \mathbf{v}(t)$, initialized as $(\mathbf{u}(0), \mathbf{v}(0)) = (\alpha \mathbf{1}, \alpha \mathbf{1})$. This parameterization is also referred to as a *spindly network* or a two-layer diagonal linear network. The training dynamics are analyzed by studying the gradient flow trajectories of $\mathbf{u}(t)$ and $\mathbf{v}(t)$ separately:

$$\dot{\mathbf{u}}(t) = -\nabla_{\mathbf{u}} \widehat{\mathcal{L}}(\mathbf{w}(t)), \qquad \dot{\mathbf{v}}(t) = -\nabla_{\mathbf{v}} \widehat{\mathcal{L}}(\mathbf{w}(t)). \qquad (7)$$

In contrast to single-layer gradient flow, this spindlified flow is *not* rotation-invariant: the coordinate-wise parameterization induces learning dynamics that differentiate directions according to their alignment with the underlying signal. Given an estimator $\mathbf{w}$, its statistical performance is measured by the excess risk

$$\mathcal{E}(\mathbf{w}) := \mathcal{L}(\mathbf{w}) - \mathcal{L}(\mathbf{w}^\star), \qquad (8)$$

which quantifies the gap between the risk of an estimator $\mathbf{w}$ and the minimum achievable risk $\mathcal{L}(\mathbf{w}^\star)$, i.e., the Bayes risk.

# 3. Lower Bound for Rotation-Invariant Algorithms

Consider empirical risk minimization with logistic loss trained on two datasets 1) Dataset-A: $\{(\mathbf{x}_i, y_i)\}_{i=1}^n$ and 2) Dataset-B: $\{(\boldsymbol{U}\mathbf{x}_i, y_i)\}_{i=1}^n$ where $\boldsymbol{U}$ is a rotation matrix and the labels are identical in both datasets. Let $\hat{\mathbf{w}}$ denote the minimizer learned from Dataset A. Since the loss depends only on inner products and satisfies

$$\ell(y \langle \mathbf{x}, \mathbf{w} \rangle) = \ell(y \langle \boldsymbol{U}\mathbf{x}, \boldsymbol{U}\mathbf{w} \rangle), \qquad (9)$$

the minimizer learned from Dataset B is $\boldsymbol{U}\hat{\mathbf{w}}$. As a consequence, for any test input $\mathbf{x}_{\text{te}}$, the conditional probability $p(\mathbf{x}_{\text{te}}, \hat{\mathbf{w}})$ at $\mathbf{x}_{\text{te}}$ induced by the model trained on Dataset A coincides with the conditional probability at the rotated input $\boldsymbol{U}\mathbf{x}_{\text{te}}$ of the model trained on Dataset B, that is,

$$p(\mathbf{x}_{\text{te}}, \hat{\mathbf{w}}) = p(\boldsymbol{U}\mathbf{x}_{\text{te}}, \boldsymbol{U}\hat{\mathbf{w}}). \qquad (10)$$

This leaves the prediction unchanged when the training data $\mathbf{x}_i$ and the test data $\mathbf{x}_{\text{te}}$ is rotated by $\boldsymbol{U}$. We formally define rotation invariance as follows:

**Definition 3.1.** *An algorithm* $\mathcal{A}$ *is called* rotation invariant *if, for an orthogonal matrix* $\boldsymbol{U} \in \mathbb{R}^{d \times d}$, *training dataset* $\{(\mathbf{x}_i, y_i)\}_{i=1}^n$, *and test input* $\mathbf{x}_{\text{te}}$, *the predictor satisfies*

$$p(\mathbf{x}_{\text{te}}; \mathcal{A}(\{(\mathbf{x}_i, y_i)\}_{i=1}^n)) = p(\boldsymbol{U}\mathbf{x}_{\text{te}}; \mathcal{A}(\{(\boldsymbol{U}\mathbf{x}_i, y_i)\}_{i=1}^n)).$$

*That is, rotating both the training data and the test input by the same orthogonal transformation leaves the induced prediction unchanged.*

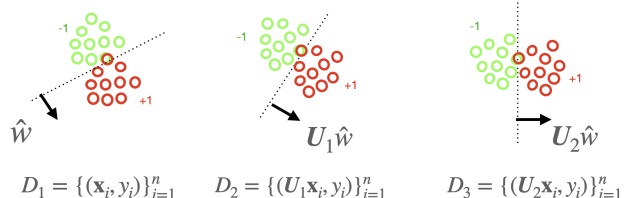

$D_1 = \{(\mathbf{x}_i, y_i)\}_{i=1}^n$    $D_2 = \{(\boldsymbol{U}_1\mathbf{x}_i, y_i)\}_{i=1}^n$    $D_3 = \{(\boldsymbol{U}_2\mathbf{x}_i, y_i)\}_{i=1}^n$

*Figure 2.* Gradient flow on single layer is rotation invariant. Rotating the data $\mathbf{x}_i$ (with labels unchanged), induces the same rotation on the estimator $\hat{\mathbf{w}}$.

**Proposition 3.2.** *Gradient flow on single layer* (6) *is rotation invariant.*

We defer the formal proof to Lemma E.1 in the appendix. Intuitively, training on a rotated dataset $\{(\boldsymbol{U}\mathbf{x}_i, y_i)\}_{i=1}^n$ rotates the empirical gradient by the same transformation, so the gradient flow trajectory is rotated pointwise as $\mathbf{w}(t) \mapsto \boldsymbol{U}\mathbf{w}(t)$ for all $t$, including any early-stopped solution.

Consequently, a rotation-invariant algorithm produces identical induced conditional probabilities under joint rotations of the training data and test input. As a result, such an algorithm cannot distinguish whether the data were generated by $\mathbf{w}^\star$ or its rotated version $\boldsymbol{U}\mathbf{w}^\star$, since both induce identical prediction after corresponding rotation of the inputs. To formalize this limitation, we reduce the performance of any rotation-invariant algorithm to the Bayes optimal predictor of a rotationally symmetrized model, that gives equal importance to every direction.

### 3.1. Reduction to a Rotationally Symmetrized Observation Model

Let $\tilde{\boldsymbol{X}} = [\boldsymbol{X}; \boldsymbol{x}_{te}]$ denote the augmented design matrix and $\tilde{\boldsymbol{y}} = [\boldsymbol{y}; y_{te}]$ the corresponding augmented label vector for the test example $(\boldsymbol{x}_{te}, y_{te})$, where $\boldsymbol{X} \in \mathbb{R}^{n \times d}$ denotes the training design matrix and $\boldsymbol{y} \in \mathbb{R}^n$ the corresponding label vector. We denote by $q(\tilde{\boldsymbol{y}}|\tilde{\boldsymbol{X}})$ the joint conditional distribution over $n$ training outcomes and a test outcome induced by the Gaussian logistic model (2). We define a rotated distribution for any rotation $\boldsymbol{U}$ as follows:

$$q_{\boldsymbol{U}}(\tilde{\boldsymbol{y}}|\tilde{\boldsymbol{X}}) := q(\tilde{\boldsymbol{y}}|\tilde{\boldsymbol{X}}\boldsymbol{U}^\top). \tag{11}$$

Then a symmetrized observation model is obtained by averaging this rotated model $q_{\boldsymbol{U}}(\tilde{\boldsymbol{y}}|\tilde{\boldsymbol{X}})$ over all rotations $\boldsymbol{U} \in \mathbb{R}^{d \times d}$ with respect to the Haar measure $\rho_H$ on the orthogonal group:

$$\bar{q}(\tilde{\boldsymbol{y}} \mid \tilde{\boldsymbol{X}}) := \int q_{\boldsymbol{U}}(\tilde{\boldsymbol{y}} \mid \tilde{\boldsymbol{X}}) \, d\rho_H(\boldsymbol{U}). \tag{12}$$

This symmetrization makes the direction $\mathbf{w}^\star$ unidentifiable. Conditioned on observing the finite training samples $(\tilde{\boldsymbol{X}}, \tilde{\boldsymbol{y}})$,

the estimator now becomes $\boldsymbol{U}\mathbf{w}^\star$ with its posterior distribution given by $p(\boldsymbol{U} \mid \tilde{\boldsymbol{X}}, \boldsymbol{y})$. Since this rotation $\boldsymbol{U}$ is uncertain, prediction at a new test point $\boldsymbol{x}_{te}$ must account for this posterior uncertainty over the rotation. The Bayes-optimal real-valued score therefore minimizes the logistic loss after averaging over all rotations consistent with the observed training data

$$s^\star(\boldsymbol{x}_{te} \mid \boldsymbol{X}, \boldsymbol{y}) \in$$
$$\arg\min_{s \in \mathbb{R}} \int \mathbb{E}_{y_{te} \sim q_{\boldsymbol{U}}(y_{te}|\tilde{\boldsymbol{X}},\boldsymbol{y})} \big[\ell(y_{te}\, s)\big]\, p(\boldsymbol{U} \mid \tilde{\boldsymbol{X}}, \boldsymbol{y}) \, d\boldsymbol{U}.$$

Here, $q_{\boldsymbol{U}}(y_{te} \mid \tilde{\boldsymbol{X}}, \boldsymbol{y})$ denotes the conditional distribution of the test label induced by rotated distribution $q_{\boldsymbol{U}}(\tilde{\boldsymbol{y}} \mid \tilde{\boldsymbol{X}})$, given the observed training labels $\boldsymbol{y}$. The Bayes risk under the symmetrized model (12) is the expected logistic loss of this Bayes-optimal predictor $s^\star$ and is given by $\mathcal{L}_B(\bar{q})$.

$$\mathcal{L}_B(\bar{q}) = \mathbb{E}_{\tilde{\boldsymbol{X}} \sim \mathcal{N}(0,\mathbf{I}),\, \tilde{\boldsymbol{y}} \sim \bar{q}(\cdot|\tilde{\boldsymbol{X}})} \Big[\ell\big(y_{te}\, s^\star(\boldsymbol{x}_{te} \mid \boldsymbol{X}, \boldsymbol{y})\big)\Big].$$

Theorem 3.3 shows that the performance of any rotation-invariant learning algorithm on the original problem $q$, is lower bounded by $\mathcal{L}_B(\bar{q})$, the Bayes risk under the symmetrized model.

**Theorem 3.3.** *Let $q(\tilde{\boldsymbol{y}} \mid \tilde{\boldsymbol{X}})$ be the joint conditional distribution over the training outcomes and one test outcome and $\hat{s}(\cdot \mid \boldsymbol{X}, \boldsymbol{y})$ be a rotation-invariant learning algorithm trained on $(\boldsymbol{X}, \boldsymbol{y})$. Define the expected loss as*

$$\mathcal{L}_{\hat{s}}(q) := \mathbb{E}_{\tilde{\boldsymbol{X}} \sim \mathcal{N}(0,\mathbf{I}),\, \tilde{\boldsymbol{y}} \sim q(\cdot|\tilde{\boldsymbol{X}})} \Big[\ell\big(y_{te}\, \hat{s}(\boldsymbol{x}_{te} \mid \boldsymbol{X}, \boldsymbol{y})\big)\Big].$$

*Then this loss is lower bounded by the Bayes risk of the symmetrized observation model:*

$$\mathcal{L}_{\hat{s}}(q) \geq \mathcal{L}_B(\bar{q}). \tag{13}$$

The proof logic is as follows. For a rotation-invariant algorithm $\hat{s}(\cdot \mid \boldsymbol{X}, \boldsymbol{y})$, replacing $q$ by any rotated model $q_{\boldsymbol{U}}$ does not change its expected loss, since the Gaussian input distribution and the algorithm are rotation invariant. Therefore, its expected loss is identical under every rotated model $q_{\boldsymbol{U}}$. Because the symmetrized model $\bar{q}$ is obtained by averaging these rotated models, it follows that

$$\mathcal{L}_{\hat{s}}(q) = \mathcal{L}_{\hat{s}}(\bar{q}).$$

Under the symmetrized model, the Bayes predictor achieves the minimal possible risk, implying

$$\mathcal{L}_{\hat{s}}(q) = \mathcal{L}_{\hat{s}}(\bar{q}) \geq \mathcal{L}_B(\bar{q}).$$

We refer to Theorem-1 in (Warmuth et al., 2021) for the full proof. So, by Theorem 3.3, we can now lower bound the excess risk (8) of any rotation invariant algorithm by just

lower bounding the difference of symmetrized Bayes risk $\mathcal{L}_B(\bar{q})$ and the population Bayes risk $\mathcal{L}(\mathbf{w}^\star)$:

$$\mathcal{L}_{\hat{s}}(q) - \mathcal{L}(\mathbf{w}^\star) \geq \underbrace{\mathcal{L}_B(\bar{q}) - \mathcal{L}(\mathbf{w}^\star)}_{\text{Bayes risk gap}} \qquad (14)$$

The following lemma reduces the *Bayes risk gap* (14) to an expectation with respect to a posterior distribution induced by the empirical likelihood and a uniform spherical prior.

**Lemma 3.4.** *Under the assumptions stated, the Bayes risk gap* (14) *satisfies*

$$\mathcal{L}_B(\bar{q}) - \mathcal{L}(\mathbf{w}^\star) = \mathbb{E}_\mathcal{D} \, \mathbb{E}_{\boldsymbol{w}(\mathcal{D}) \sim \Pi(\cdot|\mathcal{D})} \Big[ \mathcal{L}(\boldsymbol{w}(\mathcal{D})) - \mathcal{L}(\mathbf{w}^\star) \Big],$$

*where $\Pi(\boldsymbol{w} \mid \mathcal{D})$ is the posterior distribution induced by a uniform prior on the unit sphere $\mathbb{S}^{d-1}$:*

$$\Pi(\boldsymbol{w} \mid \mathcal{D}) \;\propto\; \exp\big(-n\,\widehat{\mathcal{L}}(\boldsymbol{w})\big)\,\mathbf{1}_{\{\|\boldsymbol{w}\|_2=1\}}.$$

We provide the proof in Appendix E.4. Under the symmetrized observation model $\bar{q}$, the direction of the target vector $\boldsymbol{w}^\star$ is averaged uniformly, so evaluating risk under $\bar{q}$ is equivalent to averaging over all directions $\boldsymbol{w}$ in the original conditional model $q$, resulting in an expectation with respect to the induced posterior $\Pi(\boldsymbol{w} \mid \mathcal{D})$ supported on the unit sphere.

### 3.2. Excess risk on posterior over uniform sphere:

Computing the Bayes predictor under a uniform prior on the sphere is difficult since the posterior mean does not admit a closed-form expression. In the squared-loss setting, this difficulty can be bypassed by relating the spherical prior to a Gaussian prior, which reduces the analysis to ridge regression (as shown by Warmuth et al. (2025)). No analogous reduction is available for logistic loss, where the posterior induced by a uniform spherical prior does not simplify to a tractable form. As a result, the spherical posterior must be analyzed directly through its geometric and curvature properties, making the problem fundamentally more difficult than in the squared-loss case. In the following theorem, we derive a lower bound of $\Omega(\frac{d-1}{n})$ on the excess risk on spherical posterior and draw a proof sketch.

**Theorem 3.5.** *Let $\mathcal{D} = \{(\mathbf{x}_i, y_i)\}_{i=1}^n$ be drawn i.i.d. from data model* (2) *and $n \gtrsim d + \log(\frac{1}{\delta})$. Then, with probability at least $1 - \delta$ over the draw of $\mathcal{D}$, any rotation-invariant learning algorithm outputting $\mathbf{w}(\mathcal{D})$ has an excess risk lower bound over the spherical posterior,*

$$\mathbb{E}_{\boldsymbol{w}(\mathcal{D}) \sim \Pi(\cdot|\mathcal{D})} \left[ \mathcal{L}(\mathbf{w}(\mathcal{D})) - \mathcal{L}(\mathbf{w}^\star) \right] \geq \frac{c(d-1)}{n} \qquad (15)$$

*for an absolute constant $c$.*

Due to space constraints, we defer the full proof to Appendix F.1 and only outline the main steps here. The key

idea is a change of variable that reduces the analysis on the unit sphere (in $\mathbf{w}$) to an equivalent problem in a local Euclidean coordinate system ($\mathbf{z}$). In these coordinates, the proof separates into two components (a) establishing a lower bound on the population excess risk as a function of $\mathbf{z}$, and (b) showing that the induced posterior distribution $\Pi(\mathbf{z}|\mathcal{D})$ is anti-concentrated in $\mathbf{z}$. This allows us to control second moments of the posterior and apply standard covariance bounds such as the Cramér-Rao inequality.

1. To characterize the posterior distribution $\Pi(\mathbf{w}|\mathcal{D})$ on the unit sphere $\mathbb{S}^{d-1}$, we use the chart variable map $\mathbf{w}(\mathbf{z}) = \frac{\mathbf{w}^\star + \mathbf{z}}{\|\mathbf{w}^\star + \mathbf{z}\|_2}$ and work on the domain $T := \{\mathbf{z} \in \mathbb{R}^d : \langle \mathbf{z}, \mathbf{w}^\star \rangle = 0\}$.

2. We characterize the population Hessian (at $\mathbf{w}^\star$) on $T$ (written as $\mathbf{H}_T^\star$) and prove it is isotropic. Then we prove a quadratic lower-bound on the population excess risk on a restricted domain, $\|\mathbf{z}\|_2 \leq r$ utilizing a modified self-concordance hypothesis (Proposition-1 in (Bach, 2010)):

$$\mathcal{L}(\mathbf{w}(\mathbf{z})) - \mathcal{L}(\mathbf{w}^\star) \gtrsim \mathbf{z}^T \mathbf{H}_T(\mathbf{w}^\star) \mathbf{z}. \qquad (16)$$

3. Next to account for the change of variable, we derive the density $\Pi(\mathbf{z}|\mathcal{D})$ from $\Pi(\mathbf{w}|\mathcal{D})$ using its associated Jacobian. We prove that the posterior $\Pi(\mathbf{z}|\mathcal{D})$ is locally log-concave and is globally smooth in $\mathbf{z} \in T$. Defining $V(\mathbf{z}) = -\log \Pi(\mathbf{z}|\mathcal{D})$ as the negative log-likelihood, we prove for a positive $M > m > 0$:

$$\frac{1}{m}\mathbf{I}_T \;\succeq\; \nabla^2 V(\mathbf{z}) \;\succeq\; \frac{1}{M}\,\mathbf{I}_T, \qquad (17)$$

and we derive $m$ and $M$ explicitly for this problem.

4. We lower-bound the excess risk on $\mathbf{z}$ as follows:

$$\mathbb{E}_{\Pi(\mathbf{z})}[\mathcal{L}(\mathbf{w}(\mathbf{z})) - \mathcal{L}(\mathbf{w}^\star)] \;\gtrsim$$
$$\underbrace{\Pi(\|\mathbf{z}\|_2 \leq r \mid \mathcal{D})}_{\text{A}} \underbrace{\mathbb{E}_{\Pi(\cdot|\mathcal{D}, \|\mathbf{z}\|_2 \leq r)}\big[\mathbf{z}^\top \mathbf{H}_T^\star \mathbf{z}\big]}_{\text{B}}$$

and control the factors $\Pi(\|\mathbf{z}\|_2 \leq r|\mathcal{D})$ and $\mathbb{E}_{\Pi(\cdot|\mathcal{D}, \|\mathbf{z}\|_2 \leq r)}\big[\mathbf{z}^\top \mathbf{H}_T^\star \mathbf{z}\big]$ respectively. In particular, we prove the following events occur with high probability:

(A): $\quad \Pi(\|\mathbf{z}\|_2 \leq r|\mathcal{D}) \;\geq\; 1 - \exp(-c_1 m r^2),$

(B): $\quad \mathbb{E}_{\Pi(\cdot|\mathcal{D}, \|\mathbf{z}\|_2 \leq r)}\big[\mathbf{z}^\top \mathbf{H}_T^\star \mathbf{z}\big] \geq \frac{c_2}{n + c_3 d} \mathrm{Tr}(\mathbf{H}_T^\star).$

The lower bound for (A) follows from local log-concavity of $\Pi(\mathbf{z} \mid \mathcal{D})$. For (B), we use the curvature bound $\nabla^2 V(\mathbf{z}) \succeq \frac{1}{M}\mathbf{I}_T$, which gives an upper bound on the covariance of $\Pi(\mathbf{z} \mid \mathcal{D})$ via the Cramér-Rao inequality. This covariance bound, in turn, implies a lower bound on the corresponding second-order moment.

5. Using the lower bounds for these two factors, combining all the events (using a union bound), and using

$\text{Tr}(\mathbf{H}_T^\star) = d - 1$ ( $\mathbf{H}_T^\star$ is isotropic), we show that the lower bound in Theorem 3.5 holds for $n \gtrsim d + \log(\frac{1}{\delta})$ with probability $1 - \delta$ over the draw of dataset $\mathcal{D}$.

Theorem 3.5 gives a lower-bound over a single instance draw of $\mathcal{D}$ with high probability. Let $\mathcal{E}$ denote that event (over the draw of $\mathcal{D}$) on which the lower bound in Theorem 3.4 holds, so that $\mathbb{P}_{\mathcal{D}}(\mathcal{E}) \geq 1 - \delta$. Then applying Lemma 3.4, we get for some constant $c > 0$:

$$
\begin{aligned}
&\mathbb{E}_{\mathcal{D}} \mathbb{E}_{\boldsymbol{w}(D) \sim \Pi(\cdot|\mathcal{D})} \big[ \mathcal{L}(\boldsymbol{w}(\mathcal{D})) - \mathcal{L}(\boldsymbol{w}^\star) \big] \\
&= \mathbb{E}_{\mathcal{D}} \big[ \mathbf{1}_{\mathcal{E}} \, \mathbb{E}_{\boldsymbol{w}(\mathcal{D}) \sim \Pi(\cdot|\mathcal{D})} \big[ \mathcal{L}(\boldsymbol{w}(\mathcal{D})) - \mathcal{L}(\boldsymbol{w}^\star) \big] \big] \\
&\quad + \mathbb{E}_{\mathcal{D}} \big[ \mathbf{1}_{\mathcal{E}^c} \, \mathbb{E}_{\boldsymbol{w}(\mathcal{D}) \sim \Pi(\cdot|\mathcal{D})} \big[ \mathcal{L}(\boldsymbol{w}(\mathcal{D})) - \mathcal{L}(\boldsymbol{w}^\star) \big] \big] \\
&\geq \mathbb{P}_{\mathcal{D}}(\mathcal{E}) \cdot \frac{c(d-1)}{n} \; \geq \; (1-\delta) \frac{c(d-1)}{n},
\end{aligned}
$$

where $\mathbf{1}_{\mathcal{E}}$ denotes the occurrence of event $\mathcal{E}$. This finally establishes that the Bayes risk gap in Lemma 3.4 is $\Omega(\frac{d-1}{n})$. By (14), the expected loss for any rotation invariant algorithm given an observation model $q$ is also $\Omega(\frac{d-1}{n})$. In the next section, we show that this lower-bound barrier can be overcome by a simple reparameterization of the weights.

## 4. Spindly Dynamics in Logistic Regression

We study the training dynamics of a non-rotation invariant algorithm, where we reparameterize the weight vector as the Hadamard product of two vectors, $\mathbf{w}(t) = \mathbf{u}(t) \odot \mathbf{v}(t)$, and perform gradient flow on $\mathbf{u}$ and $\mathbf{v}$ as

$$
\dot{\mathbf{u}}(t) = -\nabla_{\mathbf{u}} \widehat{\mathcal{L}}(\mathbf{w}(t)), \quad \dot{\mathbf{v}}(t) = -\nabla_{\mathbf{v}} \widehat{\mathcal{L}}(\mathbf{w}(t)). \quad (18)
$$

By the conservation law property of gradient flow (proof in Lemma-58), the joint dynamics on $\mathbf{u}$ and $\mathbf{v}$ lead to following dynamics on the predictor $\mathbf{w}$:

$$
\dot{\mathbf{w}}(t) = -|\mathbf{w}(t)| \odot \nabla \widehat{\mathcal{L}}(\mathbf{w}(t)), \quad (19)
$$

with a balanced initialization $\mathbf{u}(0) = \mathbf{v}(0) = \alpha \mathbf{1}$. It is immediate that the dynamics (18) is not rotation invariant, since the elementwise weighting $|\mathbf{w}(t)|$ depends on the coordinate system and is not preserved under orthogonal transformations of $\mathbf{w}$. The dynamics of (18) has been well studied in the linear regression setting by several past works such as (Saxe et al., 2013; Pesme & Flammarion, 2023; Jacot et al., 2021). Consider the square loss with Gaussian design matrix, where $\mathbb{E}[\mathbf{x}\mathbf{x}^\top] = \boldsymbol{\Sigma} = \text{diag}(\lambda_1, \dots, \lambda_d)$,

$$
\mathcal{L}(\mathbf{w}) = \tfrac{1}{2} \mathbb{E}\big[(y - \mathbf{x}^\top \mathbf{w})^2\big], \qquad y = \mathbf{x}^\top \mathbf{w}^\star.
$$

In this case the population gradient is linear in $\mathbf{w}$, that is, $\nabla \mathcal{L}(\mathbf{w}) = \boldsymbol{\Sigma}(\mathbf{w} - \mathbf{w}^\star)$, and the spindly dynamics decouple across coordinates. Writing $\theta_k(t) := w_k(t)$, each coordinate follows a one-dimensional Riccati differential equation of the form

$$
\dot{\theta}_k(t) = \lambda_k(\theta_k^\star \theta_k(t) - \theta_k^2(t)), \quad (20)
$$

which admits a closed-form solution. This decoupling under Gaussian design explains why the dynamics of Hadamard-parameterized linear models are fully tractable. In contrast, for logistic loss the population gradient does not decouple, even under the well-specified model,

$$
\nabla \mathcal{L}(\mathbf{w}) = \mathbb{E}_{\mathbf{x},y}\big[-y \, \mathbf{x} \, \sigma(-y \, \mathbf{x}^\top \mathbf{w})\big], \quad (21)
$$

so each coordinate of the gradient depends on the entire vector $\mathbf{w}$ through the link function $\sigma(\cdot)$. As a result, the induced spindly dynamics are no longer coordinate-separable and do not admit a closed-form solution analogous to the Riccati dynamics in linear regression (20). In Theorem 4.1, we show that Stein's Lemma allows us to write the dynamics as a *state-dependent* Riccati ODE.

**Theorem 4.1.** *Consider the sparse data-generation model* (1). *Let* $S := \text{supp}(\mathbf{w}^\star)$ *denote the non-zero index set of* $\mathbf{w}^\star$ *and denote by* $\mathbf{x}_S$ *the subvector of* $\mathbf{x}$ *on* $S$.
*(i) Exact population gradient: For every* $\mathbf{w} \in \mathbb{R}^d$ *and each coordinate* $j \in [d]$,

$$
a(\mathbf{w}) := \mathbb{E}_{\mathbf{x}}\big[\sigma'(\mathbf{x}^\top \mathbf{w})\big], \quad a^\star := \mathbb{E}_{\mathbf{x}}\big[\sigma'(\mathbf{x}_S^\top \mathbf{w}_S^\star)\big], \quad (22)
$$

*and we have the gradient coordinate-wise forms by Stein's lemma:*

$$
\begin{aligned}
[\nabla \mathcal{L}(\mathbf{w})]_j &= w_j \, a(\mathbf{w}), & j \in S^c, \quad (23) \\
[\nabla \mathcal{L}(\mathbf{w})]_j &= w_j \, a(\mathbf{w}) \; - \; w_j^\star \, a^\star, & j \in S. \quad (24)
\end{aligned}
$$

*(ii) Empirical gradient noise: Define the gradient noise vector*

$$
\boldsymbol{\zeta}(\mathbf{w}) := \widehat{\mathcal{L}}(\mathbf{w}) - \mathcal{L}(\mathbf{w}), \qquad \zeta_j(\mathbf{w}) := \mathbf{e}_j^\top \boldsymbol{\zeta}(\mathbf{w}).
$$

*Then each* $\zeta_j(\mathbf{w})$ *is the average of* $n$ *i.i.d. centered sub-Gaussian random variables. Then for any* $\delta \in (0, 1)$, *with probability at least* $1 - \delta$,

$$
\max_{1 \leq j \leq d} |\zeta_j(\mathbf{w})| \; \leq \; 4\sqrt{\frac{1}{n} \log\Big(\frac{2d}{\delta}\Big)} \; =: \gamma. \quad (25)
$$

Equations (23) and (24) provide a coordinatewise expression for the population gradient, in which each component is proportional to the corresponding parameter value and modulated by a common coupling factor $a(\mathbf{w}) := \mathbb{E}_{\mathbf{x}}\big[\sigma'(\mathbf{x}^\top \mathbf{w})\big]$ which depends on the derivative of the link function. This simplification, obtained via Stein's lemma, allows the induced spindly dynamics to be written as a system of *coupled, state-dependent* Riccati differential equations:

$$
\begin{aligned}
\dot{w}_i(t) &= \zeta_i \, w_i(t) - a(\mathbf{w}(t)) \, w_i(t)^2, & i \in S^c, \\
\dot{w}_i(t) &= \big(w_i^\star \, a^\star + \zeta_i\big) \, w_i(t) - a(\mathbf{w}(t)) \, w_i(t)^2, & i \in S,
\end{aligned}
\quad (26)
$$

where the coordinate-wise evolution is coupled through the shared scalar $a(\mathbf{w}(t))$.

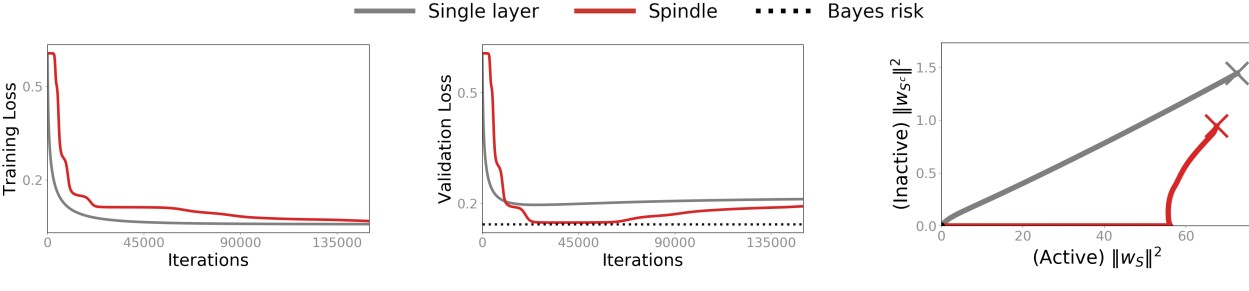

(a) Training loss.    (b) Validation loss and Bayes risk.    (c) Norm on the inactive and active subspace.

*Figure 3.* Training dynamics of sparse logistic regression under a single-layer (gray) and the spindly parameterization (red). Under early stopping, the spindly parameterization achieves lower validation error than the single-layer model. (c) Evolution of the squared norm of the weights on the active coordinates $S$ and inactive coordinates $S^c$. Along the spindly trajectory (red), the weights on inactive coordinates remain small while the iterate passes by a sparse solution.

# 5. Upper bound Excess Risk for the Spindly Network

Analyzing the dynamics of (26), we prove that early stopping induces statistical signal-noise separation. At a proper stopping time, the coupled dynamics amplify the active coordinates, $i \in S$, so that $w_i(t)$ concentrates near $w_i^\star$, while the inactive coordinates, $i \in S^c$, are strongly suppressed. As a result, the predictor $\mathbf{w}(t)$ remains close to the sparse oracle $\mathbf{w}^\star$, leading to a sharp upper bound on the excess risk. We first define the following notation:

**Signal-curvature and Noise parameters.** Let $S \subset [d]$ denote the support of the sparse parameter $\mathbf{w}^\star$. We define:

$$a^\star := \mathbb{E}_\mathbf{x}\big[\sigma'\big(\mathbf{x}_S^\top \mathbf{w}_S^\star\big)\big], \qquad w_{\min}^\star := \min_{i \in S} |w_i^\star|. \quad (27)$$

Here, $a^\star$ is the curvature of the logistic loss along the signal subspace, while $w_{\min}^\star$ is the minimum signal strength among the active coordinates. Without loss of generality, we assume $w_i^\star > 0$ for all $i \in S$. Fixing a confidence level $\eta \in (0, 1)$, we also define variables $\gamma, \delta$ and $\epsilon$ as follows:

$$\gamma := 4\sqrt{\frac{1}{n}\log\Big(\frac{2d}{\eta}\Big)}, \quad \delta := \frac{1}{2} - \frac{\gamma}{a^* w_{min}^\star}, \quad \epsilon := 1 - 4a^*.$$

Considering a relatively weaker assumption on the sample size and a signal-curvature lower bound:

$$a^\star w_{\min}^\star = \Theta(1), \quad n \geq \frac{256}{(a^* w_{\min}^\star)^2}\log(\frac{2d}{\eta}) \quad (28)$$

which ensures $\delta \in [\frac{1}{4}, \frac{1}{2})$ and $\epsilon \in (0, 1)$. On this event, with probability $(1 - \eta)$, the *sampling noise magnitude* is bounded by $\gamma$. The parameter $\delta$ acts as a signal–noise separation margin, guaranteeing that the empirical gradient noise $\gamma$ is dominated by the population signal along the active coordinates $a^* w_{min}^\star$. Lastly, $\epsilon$ signifies the curvature deviation along the signal subspace from its maximum value. With the definitions in place, we state the following theorem:

**Theorem 5.1.** *Under assumption* (28) *with probability atleast* $1 - \eta$ *over the sample draw* $\mathcal{D}$, *there exists constants* $c_1, c_2 > 0$ *depending only on* $w_{\min}^\star, a^\star, \epsilon$, *such that for all times:*

$$t \in \Big[\frac{\log d}{2w_{\min}^\star a^\star} - c_1, \ \frac{\log d}{2w_{\min}^\star a^\star} + c_1\Big],$$

*the spindly iterate satisfies*

$$\|\mathbf{w}(t) - \mathbf{w}^\star\|_2^2 \leq \epsilon^2 \|\mathbf{w}^\star\|_2^2 + \frac{16s\log(\frac{2d}{\eta})}{n(w_{\min}^\star a^\star)^2} + \frac{c_2}{d^{\delta + \frac{1}{2}}}. \quad (29)$$

The theorem shows that although the spindly dynamics under logistic regression (26) do not admit a closed-form solution, we can analyze the coupled dynamics and obtain an upper bound on the excess risk. The resulting bound is dominated by the term $O\big(\frac{s\log d}{n}\big)$ which captures the logarithmic dependence on the dimension $d$. We defer the full proof to Appendix D.4. It involves the following steps:

1) *Monotonicity of envelope ODEs:* First, we prove that the curvature along the trajectory is uniformly controlled, in the sense that $a(\mathbf{w}(t)) \in [a^\star, 1/4]$. This allows us to envelope the intractable ODE system in (26) between upper-bound and lower-bound closed-form ODE-systems. For example, for $i \in S$, we define two ODEs

$$\dot{w}_i^{\text{up}}(t) = \big(w_i^\star a^\star + \zeta_i\big) w_i^{\text{up}}(t) - a^\star w_i^{\text{up}}(t)^2 \quad (30)$$

$$\dot{w}_i^{\text{low}}(t) = \big(w_i^\star a^\star + \zeta_i\big) w_i^{\text{low}}(t) - \frac{1}{4} w_i^{\text{low}}(t)^2 \quad (31)$$

and prove that $w_i^{\text{low}}(t) \leq w_i(t) \leq w_i^{\text{up}}(t)$ for all $0 \leq t \leq T$, with both envelope trajectories monotone on this interval. An analogous construction applies to the coordinates in $i \in S^c$.

2) *Early-stopping time:* We choose the stopping time so that the lower envelope of the weakest active coordinate

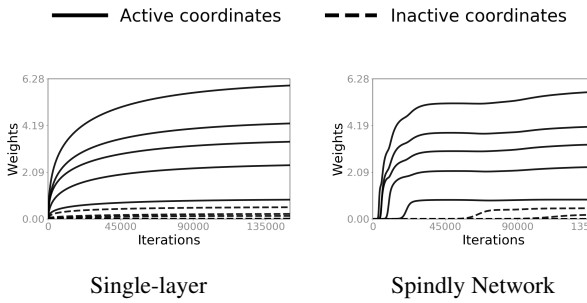

*Figure 4.* Growth of active and inactive coordinates under rotation-invariant and spindly parameterizations.

reaches a prescribed fraction of its target value. Recalling $w_{\min}^\star := \min_{i \in S} w_i^\star$, and defining $i_{\min} \in \arg\min_{i \in S} w_i^\star$, we define the stopping time $T(\varepsilon)$ as:

$$T(\epsilon) := \frac{1}{2 w_{\min}^\star a^\star} \log\left( \frac{4 w_{\min}^\star a^\star d - 1}{\frac{4 a^\star}{(1-\epsilon) w_{\min}^\star} - 1} \right), \quad (32)$$

which is when the lower-envelope of coordinate $i_{\min}$ reaches proportion $(1 - \epsilon)$ of the target $w_{\min}^\star$, that is,

$$w_{i_{\min}}^{\mathrm{low}}(T(\epsilon)) = (1 - \epsilon) w_{\min}^\star.$$

In Lemma D.7, we show that once the smallest active coordinate reaches $(1 - \epsilon)$ of its target, all other lower-envelope active coordinates are automatically closer to their respective targets.

3) *Active coordinate error at $T(\epsilon)$.* By the envelope comparison, at the stopping time $T(\epsilon)$ the weakest active coordinate satisfies

$$(1 - \epsilon) w_{\min}^\star \leq w_{i_{\min}}(T) \leq w_{\min}^\star + \frac{\zeta_{i_{\min}}}{a^\star w_{\min}^\star},$$

and all other active coordinates $i \in S$ satisfy analogous bounds with some $\epsilon_S < \epsilon$. Summing the resulting errors over $i \in S$ gives

$$\sum_{i \in S} \left( w_i(T) - w_i^\star \right)^2 \ \leq \ \epsilon^2 \| \mathbf{w}^\star \|_2^2 \ + \ \frac{16 s \log(\frac{2d}{\delta})}{n \left( w_{\min}^\star a^\star \right)^2}. \quad (33)$$

4) *Inactive coordinate error at $T(\epsilon)$:* We show that at $T(\epsilon)$, the inactive coordinate remains suppressed. So, the upper bound ODE for $w_i(t)$ with $i \in S^c$ remains suppressed,

$$w_i^{\mathrm{up}}(t) = \frac{\zeta_i / a^*}{1 + \left( \frac{\zeta_i d}{a^*} - 1 \right) e^{-\zeta_i t}} \leq \frac{2}{d} e^{\zeta_i t}. \quad (34)$$

Using expression for $T(\epsilon)$, and summing over all $i \in S^c$,

$$\sum_{i \in S^c} w_i^2(T) \leq \frac{c_2}{d^{\frac{1}{2} + \delta}} \quad (35)$$

Summing (35) and (33), we get the final statement (29). Since the logistic risk $\mathcal{L}$ is $\frac{1}{4}$-smooth, the excess risk relates as

$$\mathcal{L}(\mathbf{w}) - \mathcal{L}(\mathbf{w}^\star) \leq \frac{1}{8} \| \mathbf{w} - \mathbf{w}^\star \|_2^2$$

which implies the excess risk is also $O(\frac{s \log d}{n})$. While envelope arguments have been used for matrix Riccati flows (Arous et al., 2025), our analysis is the first to handle spindly dynamics under logistic loss, where the coupling is state-dependent and no closed-form solution exists like in linear regression.

The only condition required for obtaining the excess risk upper bound is the sample size requirement $n \geq \frac{256}{(a^\star w_{\min}^\star)^2} \log\left( \frac{2d}{\eta} \right)$. When the signal–curvature parameter satisfies $a^\star w_{\min}^\star = \Theta(1)$ this condition scales as $n \gtrsim \log d$, and is therefore automatically satisfied in the overdetermined regime $n \gtrsim d$.

**Rate Separation in Sparse Logistic Regression:** In overconstrained case $n \gtrsim d$ under a well-specified logistic model, the maximum likelihood estimator (MLE) exists and is unique. Classical asymptotic theory (Van der Vaart, 2000) implies that the MLE achieves an excess risk of order $d/n$. More recently, Chardon et al. (2024) established a non-asymptotic version of this result, showing as soon as MLE exists, its excess risk is $O(d/n)$. Algorithms whose iterates converge to the MLE therefore admit this $d/n$ excess risk. Our result focuses on the trajectory rather than convergence to the unique MLE.

In sparse-logistic regression, the minimax excess risk scales as $O(\frac{s \log(ed/s)}{n})$ under standard conditions, showing that MLE rate is suboptimal in this setting, motivating the search for algorithmic trajectories that can attain this rate through early stopping.

Our result fills this gap stating that rotation invariant algorithm (with early stopping) still suffer an excess risk with a *lower bound* $\Omega(\frac{d-1}{n})$ matching the rate for MLE. However, a non-rotation invariant algorithm with a proper early-stopping time can achieve the rate $O(\frac{s \log d}{n})$, which matches the sharp sparse-logistic rate $O(\frac{s \log(ed/s)}{n})$ up to the usual logarithmic simplification.

## 6. Numerical Experiments

We consider sparse binary logistic regression with dimension $d = 50$, sparsity level $s = 5$, and $n = 1000$ i.i.d. training samples drawn from an isotropic Gaussian design with labels generated according to the true conditional distribution (2) and a test set of size $10^5$. We compare standard single-layer logistic regression with a spindly-parameterized model having effective weights $\mathbf{w} = \mathbf{u} \odot \mathbf{v}$. From Figure 3,

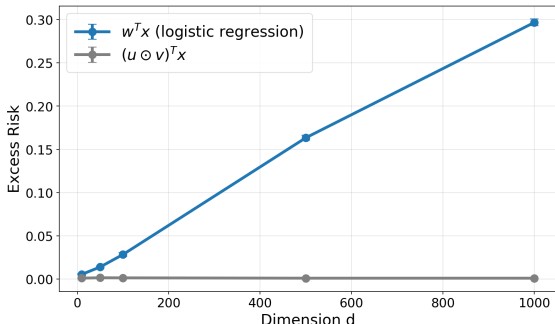

*Figure 5.* Excess risk $\mathcal{L}(\hat{\mathbf{w}}) - \mathcal{L}(\mathbf{w}^*)$ plots averaged over sample draws of $\mathcal{D}$. The best estimator $\hat{\mathbf{w}}$ was obtained using early-stopping for both the algorithms.

the spindly model achieves lower validation error under early stopping. Figure 4 illustrates the contrasting evolution of active and inactive coordinates under the two parameterizations.

In Figure 5, we plot the excess risk $\mathcal{L}(\hat{\mathbf{w}}) - \mathcal{L}(\mathbf{w}^\star)$ as a function of the dimension $d$. For each value of $d$, we report the best validation performance achieved via early stopping for both the single-layer and spindly parameterizations, averaged over 10 independent draws of the dataset $\mathcal{D}$. The observed scaling of the excess risk for both algorithms is consistent with the lower and upper-bound rates established in our theoretical analysis.

## 7. Related works

Performance bounds for logistic regression have been proven with a large number of different methodologies.

*Regret bounds for on-line logistic regression:* One of the earliest research on logistic regression proves regret bounds for the Gradient Descent (GD) and Exponentiated Gradient (EG) algorithms (Helmbold et al., 1999; Kivinen & Warmuth, 2001). The bounds exemplify the different performance between both types of update on dense and sparse logistic regression problems. Note that EG is mirror descent based on the logarithm link and the EG update family can be reparameterized as GD by reparameterizing the weights $w_i$ as $u_i v_i$ (Amid & Warmuth, 2020a; Chizat, 2022). We focus on excess risk rather than online regret, since excess risk evaluates performance with respect to the population minimizer, which is known and fixed under the data-generating distribution.

*Excess risk on logistic regression:* Using self-concordant analysis, Bach (2010) extends classical generalization arguments to logistic loss. In the underdetermined regime, several works establish benign overfitting or margin-based generalization guarantees for logistic regression, typically via implicit bias toward the max-margin solution (Monta-

nari et al., 2019; Chatterji & Long, 2021; Cao et al., 2021; Muthukumar et al., 2021; Shamir, 2022). In contrast, in the overdetermined regime ($n \gtrsim d$), recent work provides sharp *upper bounds* on the excess logistic risk of order $\tilde{O}(d/n)$ (Ostrovskii & Bach, 2021; Kuchelmeister & van de Geer, 2024; Chardon et al., 2024; Paik et al., 2025). To the best of our knowledge, however, no prior work establishes *excess-risk lower bounds* for *overdetermined* logistic regression.

*Rotation invariance and sparse recovery in the underdetermined case:* Several works reveal rotation invariance breaking dynamics that promote sparsity and achieve near-optimal statistical rates (Pesme & Flammarion, 2023; Vaskevicius et al., 2019; Woodworth et al., 2020; Amid & Warmuth, 2020a). Related analyses extend to multiplicative updates and reparameterized gradient flows (Warmuth et al., 2021; Amid et al., 2022). Earlier works (Warmuth & Vishwanathan, 2005) also show that staying in the span of the training set can restrict learning sparse features. For sparse logistic regression, Matsumoto & Mazumdar (2025) provides sample complexity result for non-rotation invariant algorithm such as binary iterative hard thresholding. Ng (2004) proves lower bounds for rotationally invariant algorithms under separable (realizable) setting. For more detailed recent treatment in linear regression see (Warmuth et al., 2021).

*Lower bounds for rotation-invariant algorithms:* Most closely related to our work, Warmuth et al. (2025) establish excess-risk lower bounds for rotation-invariant algorithms in noisy linear regression in the overdetermined regime. Our work provides the first analogous lower bound for *logistic regression*, showing that rotation invariance fundamentally limits statistical efficiency even when $n \gtrsim d$. The effect comes without externally added noise and is a consequence of using hard labels sampled from a sparse target. We provide a more detailed literature review in Appendix A.

## 8. Conclusion

This work demonstrates a fundamental limitation of rotation invariant algorithms when the loss is the logistic loss and the labels are sampled from a sparse target. A general class of rotation invariant algorithms includes gradient descent on feed forward neural networks with a fully connected input layer that is initialized with a rotation invariant distribution. Our lower bound holds across this entire class of algorithms. Analogous results have been shown before for the square loss (Warmuth et al., 2025). We believe this phenomenon is fundamental but underappreciated: it shows that standard optimization procedures can remain statistically suboptimal when their symmetry conflicts with the nature of data. In Appendix B, we provide experiments showing that the gap persists even for anisotropic input distributions. Proving a corresponding lower bound in this setting remains open.

## Impact Statement

This paper presents theoretical work whose primary goal is to advance the understanding of optimization dynamics and statistical limits of learning algorithms in sparse logistic regression. Our results contribute to the foundations of machine learning theory by clarifying when and why certain algorithmic symmetries lead to suboptimal statistical performance. The work is methodological in nature and does not involve new data, applications, or deployment considerations. We do not anticipate direct negative societal impacts arising from this work.

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

# A. Expanded Related works

*Asymptotic implicit bias in underdetermined linear/logistic regression:* The empirical success of gradient-based optimization in deep learning has spurred extensive research into the algorithmic bias of gradient descent, often referred to as implicit regularization. In simple convex settings, this phenomenon manifests in well-understood forms: in underdetermined linear regression, gradient descent converges to the minimum $\ell_2$-norm solution, while in classification with linearly separable data, gradient descent diverges in norm but converges in direction to the maximum-margin classifier (Soudry et al., 2018; Ji & Telgarsky, 2019). These foundational results have motivated a broad line of work characterizing the implicit bias of gradient descent in the underdetermined regime under various optimization geometries, including homogeneous deep networks (Gunasekar et al., 2018; Lyu & Li, 2019; Ji & Telgarsky, 2018a), non-homogeneous networks (Cai et al., 2025), mirror descent (Sun et al., 2023), steepest descent (Tsilivis et al., 2024), and the effect of initialization scale (Moroshko et al., 2020). Subsequent studies have further investigated convergence rates and asymptotic risk behavior under these settings (Nacson et al., 2019; Ji & Telgarsky, 2018b; 2021).

*Statistical benefits of implicit bias for linear regression:* The statistical consequences of implicit regularization in the underdetermined setting have been extensively studied in linear regression. In particular, the minimum $\ell_2$-norm interpolating solution can achieve vanishing excess risk despite fitting noisy training data, a phenomenon known as benign overfitting (Bartlett et al., 2020; Tsigler & Bartlett, 2023), under suitable assumptions on the data covariance. Beyond minimum-norm interpolation, early stopping of gradient descent has been shown to yield vanishing excess risk for general covariance structures (Bühlmann & Yu, 2003; Yao et al., 2007), with analogous guarantees established for stochastic gradient descent and its variants (Li et al., 2023; Wu et al., 2022; Zou et al., 2021). Moreover, early stopping can provide statistical advantages over ridge regression, achieving improved excess-risk performance in certain regimes (Wu et al., 2025b; Ali et al., 2019).

*Excess risk calculation in underdetermined logistic regression:* Analyzing statistical benefits in logistic regression via excess-risk bounds is substantially more delicate than in linear regression due to the non-quadratic nature of the logistic loss. Nonetheless, Bach (2010) showed that generalization guarantees for squared loss can be extended to logistic loss using tools from self-concordant analysis. In the separable (underdetermined) regime, a growing body of work has established forms of benign overfitting for logistic regression, typically through the implicit bias toward the max-margin estimator, with guarantees on classification risk or related notions of generalization (Montanari et al., 2019; Chatterji & Long, 2021; Cao et al., 2021; Wang & Thrampoulidis, 2022; Muthukumar et al., 2021; Shamir, 2022; Wu et al., 2025a). Interesting generalization behaviours such as grokking emerge near the edge of separability (Beck et al., 2024).

*Excess-risk upper bounds for overdetermined logistic regression:* In overdetermined logistic regression ($n \gtrsim d$), the existence of a finite-sample maximum likelihood estimator enables sharp characterizations of excess-risk *upper bounds*. Several recent works establish rates of order $\tilde{O}(d/n)$ for the population logistic risk in this regime (Ostrovskii & Bach, 2021; Kuchelmeister & van de Geer, 2024; Chardon et al., 2024; Paik et al., 2025). Specifically, Ostrovskii & Bach (2021) leverage the self-concordance of the logistic loss to derive non-asymptotic excess-risk bounds, Kuchelmeister & van de Geer (2024) analyze refined guarantees in low-noise or small-sample regimes, and Chardon et al. (2024) provide excess-risk bounds for the MLE under general conditions. Complementarily, Hsu & Mazumdar (2024) characterize the minimax sample complexity for parameter estimation in logistic regression with Gaussian design, revealing sharp phase transitions as a function of the inverse temperature. Excess-risk upper bounds for *regularized* logistic regression have also been studied in related settings (Bach, 2014; Marteau-Ferey et al., 2019; Tsigler et al., 2025). To the best of our knowledge, however, no prior work establishes excess-risk *lower bounds* for overdetermined logistic regression, a gap addressed by our results.

*Implicit regularization in sparse linear regression:* For sparse linear regression, a growing body of work studies the training dynamics of *non-rotationally invariant* algorithms, revealing rich saddle-to-saddle dynamics that promote sparsity during optimization (Pesme & Flammarion, 2023; Pesme et al., 2024; Even et al., 2023; Gunasekar et al., 2017; Jacot et al., 2021; Saxe et al., 2013; Gissin et al., 2019; Poon & Peyré, 2021; Ghosh et al., 2025). Under Gaussian or near-Gaussian design assumptions (e.g., restricted isometry–type conditions), it has been shown that such symmetry-breaking algorithms can achieve *near-optimal statistical rates* for sparse recovery (Vaskevicius et al., 2019; 2020; Zhou & Ge, 2023; Woodworth et al., 2020). Relatedly, the training dynamics of *multiplicative update* methods and reparameterizations have also been extensively analyzed, highlighting their implicit bias toward sparse solutions (Amid & Warmuth, 2020a;b; Amid et al., 2022; Kivinen & Warmuth, 1997; Majidi et al., 2021; Warmuth et al., 2021).

*Online regret bounds for logistic loss:* While our work measures the statistical excess risk under an i.i.d. data-generating distribution, previous work such as (Shamir, 2020) studies *online regret* for logistic loss. Regret evaluates performance on a realized sequence of seen examples and compares loss of the algorithm to that of the best fixed parameter chosen in

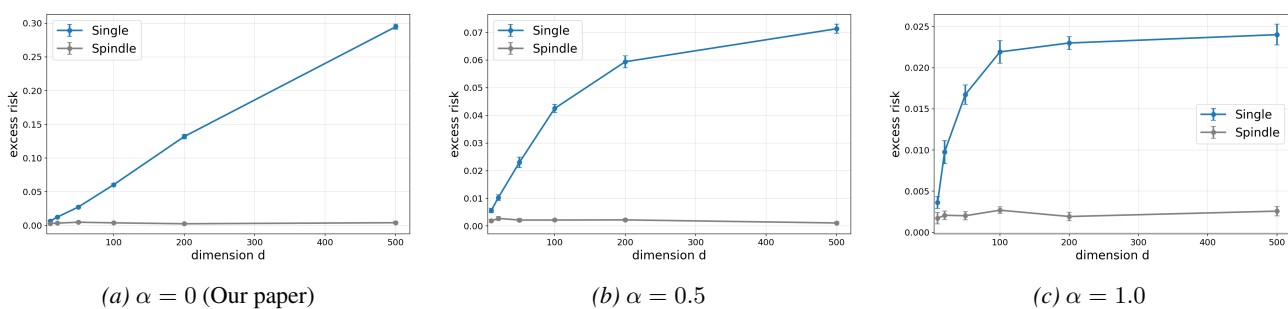

*Figure 6.* Excess risk $\mathcal{L}(\hat{\mathbf{w}}) - \mathcal{L}(\mathbf{w}^*)$ averaged over independent draws of the dataset $\mathcal{D}$. The input distribution is Gaussian $\mathbf{x} \sim \mathcal{N}(0, \Sigma)$ with a damped covariance spectrum, where the eigenvalues of $\Sigma$ decay according to a power law $\lambda_i \propto i^{-\alpha}$.

*hindsight* for that same sequence.

*Sparse logistic regression:* In the underdetermined regime, Matsumoto & Mazumdar (2025) establish sample-complexity guarantees for sparse logistic regression, providing purely statistical recovery results under sparsity assumptions. The authors propose Binary Iterative Hard Thresholding, a non-rotation-invariant, sparsity-enforcing algorithm that achieves optimal sample complexity for sparse binary GLMs. Earlier, Ng (2004) proved lower bounds for rotationally invariant algorithms under deterministic hard labels. For more detailed recent treatment in linear regression see (Warmuth et al., 2021). To the best of our knowledge, however, a principled analysis of the training dynamics of non-rotationally invariant algorithms, together with a characterization of the resulting statistical advantages for sparse recovery in logistic regression remains open.

To the best of our knowledge, Warmuth et al. (2025) is the first work to establish a lower bound for rotationally invariant algorithms in noisy linear regression in the overdetermined regime, demonstrating that even when the sample size $n$ exceeds the ambient dimension $d$, noise can fundamentally mislead rotation-invariant procedures. An analogous excess-risk lower bound for logistic regression in the overdetermined setting has remained open; establishing such a result constitutes the first contribution of our work.

Our second contribution is to investigate the training dynamics and corresponding statistical advantages of non-rotation invariant algorithm, focusing on a two-layer diagonal linear network in sparse logistic regression. We identify the primary challenges in analyzing the dynamics of depth-2 diagonal linear networks under the logistic loss and show how our analysis enables the derivation of finite-sample excess-risk upper bounds for such networks under early stopping.

## B. Excess risk gap beyond isotropic Gaussian data

We also include an experiment for sparse logistic regression beyond the isotropic Gaussian design. Specifically, we draw $\mathbf{x} \sim \mathcal{N}(\mathbf{0}, \boldsymbol{\Sigma})$, where $\boldsymbol{\Sigma} = \mathrm{diag}(\lambda_1, \ldots, \lambda_d)$ and the eigenvalues follow a power-law decay, $\lambda_j \asymp j^{-\alpha}$ for $\alpha \geq 0$. The case $\alpha = 0$ recovers the isotropic setting, where the excess risk of rotation-invariant methods grows approximately linearly with the ambient dimension $d$, consistent with our theory. For $\alpha > 0$, the design is anisotropic, so the isotropic lower-bound argument no longer applies directly. Nevertheless, the experiments show that a performance gap persists. In this regime, the risk depends on the alignment between the oracle parameter $\mathbf{w}^\star$ and the eigenspaces of $\boldsymbol{\Sigma}$. Intuitively, symmetrization spreads estimation error across coordinates, but directions with small eigenvalues contribute less to prediction risk. Consequently, as seen in Figures 6, the growth of excess risk with $d$ becomes sublinear because error propagation is attenuated along low-variance directions. Formalizing this anisotropic phenomenon theoretically is an interesting open direction.

We list two other open problems as future work:

1. Can our lower bounds for rotation invariant algorithms be generalized to the case when hard labels are sampled from other exponential family distributions such as the Poisson and Exponential distributions. Our preliminary experiments show that there is again a performance gap between rotation invariant algorithms and GD on the spindlified network. Quantifying and proving this gap for general exponential families is a challenging goal.

2. In the previous setting of over-constrained noisy sparse linear regression (Warmuth et al., 2025), a large variety of non-rotation invariant algorithms were shown to decisively beat rotation invariant algorithms. This include two-sided versions of the exponentiated gradient algorithm (a.k.a. mirror descent with the $\mathrm{arcsinh}$ link function) and priming

(Warmuth & Amid, 2023). Also Lasso achieves the same feat. We believe that a number of these algorithm can be adapted to learn $s$ sparse logistic regression with the same upper excess risk bound.

## C. Proof of Propositions

Consider the learning problem where the task is to learn a target $\mathbf{w}^\star \in \mathbb{R}^d$ through a supervised learning problem where $n > d$ samples are observed. Let $\boldsymbol{x} \in \mathbb{R}^d$ be an input drawn i.i.d. from an isotropic distribution. Conditioned on $\boldsymbol{x}$, the label $y \in \{+1, -1\}$ is generated according to

$$\mathbb{P}(y = +1 \mid \boldsymbol{x}) = \sigma(\langle \boldsymbol{x}, \boldsymbol{w}^\star \rangle), \qquad \mathbb{P}(y = -1 \mid \boldsymbol{x}) = 1 - \sigma(\langle \boldsymbol{x}, \boldsymbol{w}^\star \rangle),$$

We distinguish two learning settings.

*1) Soft-label (conditional-expectation) learning:* Suppose that instead of observing a single realization of $y$, the learner has access to the true conditional distribution of the label given each input. The empirical soft-label risk is defined as

$$\widehat{\mathcal{L}}_{\text{soft}}(\boldsymbol{w}) := \frac{1}{n} \sum_{i=1}^n \mathbb{E}_{y \mid \boldsymbol{X} = \boldsymbol{x}_i} \big[ \ell(y \langle \boldsymbol{x}_i, \boldsymbol{w} \rangle) \big]. \tag{36}$$

For the logistic loss $\ell(t) = \log(1 + \mathrm{e}^{-t})$, this can be written equivalently as

$$\widehat{\mathcal{L}}_{\text{soft}}(\boldsymbol{w}) = \frac{1}{n} \sum_{i=1}^n \Delta_H(\sigma(\langle \boldsymbol{x}_i, \boldsymbol{w}^\star \rangle),\, \sigma(\langle \boldsymbol{x}_i, \boldsymbol{w} \rangle)), \tag{37}$$

up to an additive constant independent of $\boldsymbol{w}$, where $\Delta_H(\cdot, \cdot)$ denotes the Bernoulli cross-entropy.

*2) Sampled-label (hard-label) learning:* Alternatively, the learner observes a single sampled label

$$y_i \in \{+1, -1\}, \qquad y_i = +1 \text{ with prob } \sigma(\langle \boldsymbol{x}, \boldsymbol{w}^\star \rangle), \quad y_i = -1 \text{ with prob } 1 - \sigma(\langle \boldsymbol{x}, \boldsymbol{w}^\star \rangle).$$

The corresponding empirical risk is

$$\widehat{\mathcal{L}}_{\text{hard}}(\boldsymbol{w}) := \frac{1}{n} \sum_{i=1}^n \ell(y_i \langle \boldsymbol{x}_i, \boldsymbol{w} \rangle) = \frac{1}{n} \sum_{i=1}^n \Delta_H(y_i,\, \sigma(\langle \boldsymbol{x}_i, \boldsymbol{w} \rangle)). \tag{38}$$

*3) Semi-soft label learning.* We also consider an intermediate setting between hard-label and soft-label learning. For each input $\boldsymbol{x}_i$, the learner observes $m \geq 1$ independent Monte Carlo samples

$$y_{i,1}, \ldots, y_{i,m} \sim \mathbb{P}(y \mid \boldsymbol{X} = \boldsymbol{x}_i),$$

drawn from the true conditional label distribution. The corresponding empirical risk is defined as

$$\widehat{\mathcal{L}}_m(\boldsymbol{w}) := \frac{1}{n} \sum_{i=1}^n \frac{1}{m} \sum_{j=1}^m \ell(y_{i,j} \langle \boldsymbol{x}_i, \boldsymbol{w} \rangle). \tag{39}$$

Let $\mathcal{L}(\boldsymbol{w})$ denote the population loss, which by assumption admits a unique minimizer $\boldsymbol{w}^\star$ which is given by expectaion over joint distribution over the population data $(\boldsymbol{x}, y)$:

$$\mathcal{L}(\boldsymbol{w}) := \mathbb{E}_{\boldsymbol{x}} \mathbb{E}_{y \mid \boldsymbol{X} = \boldsymbol{x}} \big[ \ell(y \langle \boldsymbol{x}, \boldsymbol{w} \rangle) \big] = \mathbb{E}_{(\boldsymbol{x}, y)} \big[ \ell(y \langle \boldsymbol{x}, \boldsymbol{w} \rangle) \big]. \tag{40}$$

We establish the following statements in the overdetermined regime $n > d$: the empirical soft-label risk $\widehat{\mathcal{L}}_{\text{soft}}(\boldsymbol{w})$ shares the same global minimizer as the population loss $\mathcal{L}(\boldsymbol{w})$, namely $\boldsymbol{w}^\star$.

**Proposition C.1.** *If the design matrix has full column rank, then the empirical soft-label risk*

$$\widehat{\mathcal{L}}_{\text{soft}}(\mathbf{w}) := \frac{1}{n} \sum_{i=1}^n \mathbb{E}_{y \mid \mathbf{X} = \mathbf{x}_i} \big[ \ell(y \langle \mathbf{x}_i, \mathbf{w} \rangle) \big]$$

*has the same unique global minimizer $\mathbf{w}^\star$ as the population risk.*

*Proof.* Expanding the conditional expectation over $y \in \{-1, +1\}$ gives

$$\widehat{\mathcal{L}}_{\text{soft}}(\mathbf{w}) = \frac{1}{n} \sum_{i=1}^{n} \Big( \sigma(\langle \mathbf{x}_i, \mathbf{w}^\star \rangle), \ell(\langle \mathbf{x}_i, \mathbf{w} \rangle) + (1 - \sigma(\langle \mathbf{x}_i, \mathbf{w}^\star \rangle)) \, \ell(-\langle \mathbf{x}_i, \mathbf{w} \rangle) \Big). \tag{41}$$

Using $\ell'(t) = -\sigma(-t)$ and $\sigma(-t) = 1 - \sigma(t)$, a direct differentiation yields

$$
\begin{aligned}
\nabla \widehat{\mathcal{L}}_{\text{soft}}(\mathbf{w}) &= \frac{1}{n} \sum_{i=1}^{n} \Big( \sigma(\langle \mathbf{x}_i, \mathbf{w}^\star \rangle) \, \ell'(\langle \mathbf{x}_i, \mathbf{w} \rangle) \, \mathbf{x}_i + (1 - \sigma(\langle \mathbf{x}_i, \mathbf{w}^\star \rangle)) \, \ell'(-\langle \mathbf{x}_i, \mathbf{w} \rangle) \, (-\mathbf{x}_i) \Big) \\
&= \frac{1}{n} \sum_{i=1}^{n} \Big( -\sigma(\langle \mathbf{x}_i, \mathbf{w}^\star \rangle) \, \sigma(-\langle \mathbf{x}_i, \mathbf{w} \rangle) \, \mathbf{x}_i + (1 - \sigma(\langle \mathbf{x}_i, \mathbf{w}^\star \rangle)) \, \sigma(\langle \mathbf{x}_i, \mathbf{w} \rangle) \, \mathbf{x}_i \Big) \\
&= \frac{1}{n} \sum_{i=1}^{n} \Big( \sigma(\langle \mathbf{x}_i, \mathbf{w} \rangle) - \sigma(\langle \mathbf{x}_i, \mathbf{w}^\star \rangle) \Big) \mathbf{x}_i. \tag{42}
\end{aligned}
$$

Evaluating (42) at $\mathbf{w} = \mathbf{w}^\star$ gives $\nabla \widehat{\mathcal{L}}_{\text{soft}}(\mathbf{w}^\star) = \mathbf{0}$.

It remains to argue uniqueness. The Hessian is

$$\nabla^2 \widehat{\mathcal{L}}_{\text{soft}}(\mathbf{w}) = \frac{1}{n} \sum_{i=1}^{n} \sigma'(\langle \mathbf{x}_i, \mathbf{w} \rangle) \, \mathbf{x}_i \mathbf{x}_i^\top, \qquad \sigma'(t) = \sigma(t) (1 - \sigma(t)) > 0, \tag{43}$$

so for any $\mathbf{v} \neq \mathbf{0}$,

$$\mathbf{v}^\top \nabla^2 \widehat{\mathcal{L}}_{\text{soft}}(\mathbf{w}) \, \mathbf{v} = \frac{1}{n} \sum_{i=1}^{n} \sigma'(\langle \mathbf{x}_i, \mathbf{w} \rangle) \, (\mathbf{x}_i^\top \mathbf{v})^2.$$

If the design matrix has full column rank, then $\mathbf{x}_i^\top \mathbf{v} = 0$ for all $i$ implies $\mathbf{v} = \mathbf{0}$, hence the right-hand side is strictly positive for all $\mathbf{v} \neq \mathbf{0}$. Thus $\nabla^2 \widehat{\mathcal{L}}_{\text{soft}}(\mathbf{w}) \succ \mathbf{0}$ for all $\mathbf{w}$, so $\widehat{\mathcal{L}}_{\text{soft}}$ is strictly convex and has a unique global minimizer. Since $\mathbf{w}^\star$ is a stationary point, it is this unique minimizer.

Finally, by definition the population risk is

$$\mathcal{L}(\mathbf{w}) := \mathbb{E}_{\mathbf{X}} \, \mathbb{E}_{y|\mathbf{X}} \big[ \ell(y \langle \mathbf{X}, \mathbf{w} \rangle) \big],$$

so under the well-specified model $\mathbf{w}^\star$ is also the (unique) population minimizer. Therefore the empirical soft-label risk has the same unique global minimizer $\mathbf{w}^\star$ as the population risk. □

**Proposition C.2.** *Given the well-specified logistic model and letting $\mathbf{X} \in \mathbb{R}^{n \times d}$ be the design matrix and assume $n > d$. Define the empirical logistic loss*

$$\widehat{\mathcal{L}}(\mathbf{w}) := \frac{1}{n} \sum_{i=1}^{n} \log(1 + \exp(-y_i \, \mathbf{x}_i^\top \mathbf{w})).$$

*1) Then there exist constants $c_1, c_2 > 0$, depending only on $\|\mathbf{w}^\star\|$, such that*

$$\mathbb{P}(\{(\mathbf{x}_i, y_i)\}_{i=1}^{n} \text{ is linearly separable}) \leq c_1 e^{-c_2 n}.$$

*2) On this event, the empirical loss $\widehat{\mathcal{L}}$ is coercive. Moreover, if $\mathbf{X}$ has full column rank, then $\widehat{\mathcal{L}}$ is strictly convex and admits a unique global minimizer (different than $\mathbf{w}^\star$).*

*Proof.* Let $Z := \mathbf{x}^\top \mathbf{w}^\star$. Since $\mathbf{x} \sim \mathcal{N}(\mathbf{0}, \mathbf{I}_d)$, we have

$$Z \sim \mathcal{N}(0, \|\mathbf{w}^\star\|^2).$$

Fixing a constant $a > 0$ and defining region where labels are highly uncertain

$$\mathcal{A} := \{ \mathbf{x} \in \mathbb{R}^d : |\mathbf{x}^\top \mathbf{w}^\star| \leq a \}.$$

On $\mathcal{A}$ we have the sigmoid prediction bounded as ,

$$\sigma(\mathbf{x}^\top \mathbf{w}^\star) \in [\sigma(-a), \sigma(a)] = [\delta, 1 - \delta], \qquad \delta := \sigma(-a) \in (0, 1/2).$$

Defining $p_0$ as the probability with which $\mathcal{A}$ occurs and given as

$$p_0 := \mathbb{P}(\mathcal{A}) = \mathbb{P}(|Z| \leq a) = 2 \, \Phi\left(\frac{a}{\|\mathbf{w}^\star\|}\right) - 1,$$

where $\Phi$ denotes the standard Gaussian cumulative distribution function.

$N_\mathcal{A} := \sum_{i=1}^n \mathbf{1}\{\mathbf{x}_i \in \mathcal{A}\}$ which is the count of data being in that set. Then it follows a binomial distribution as follows:

$$N_\mathcal{A} \sim \mathrm{Binomial}(n, p_0). \tag{44}$$

Applying the Chernoff's bound for Binomial random variable

$$\mathbb{P}\left(N_\mathcal{A} \leq \frac{p_0}{2}n\right) \leq \exp\left(-\frac{p_0}{8}n\right). \tag{45}$$

On the event $\{N_\mathcal{A} \geq (p_0/2)n\}$, we choose a subset $S \subset [n]$ of size such that $\mathbf{x}_i \in \mathcal{A}$ for all $i \in S$

$$m := \left\lfloor \frac{p_0}{2}n \right\rfloor \tag{46}$$

Conditioning on $\{\mathbf{x}_i\}_{i \in S}$, the labels $\{y_i\}_{i \in S}$ are independent and satisfy

$$\mathbb{P}(y_i = 1 \mid \mathbf{x}_i) \in [\delta, 1 - \delta] \qquad \text{for all } i \in S. \tag{47}$$

Hence, for any fixed labeling pattern $(\bar{y}_i)_{i \in S} \in \{\pm 1\}^m$,

$$\mathbb{P}((y_i)_{i \in S} = (\bar{y}_i)_{i \in S} \mid (\mathbf{x}_i)_{i \in S}) \leq (1 - \delta)^m. \tag{48}$$

Fixing the feature vectors $\{\mathbf{x}_i\}_{i \in S}$. If the full dataset is linearly separable, then the restriction $(y_i)_{i \in S}$ must be realizable by some linear classifier in $\mathbb{R}^d$.

For $m$ points in $\mathbb{R}^d$, the number of distinct labelings realizable by linear classifiers is at most polynomial in $m$: specifically, there exists a constant $C_d > 0$, depending only on $d$, such that the number of realizable labelings is at most $C_d \, m^{d+1}$.

On the other hand, for any fixed labeling $(\bar{y}_i)_{i \in S}$,

$$\mathbb{P}((y_i)_{i \in S} = (\bar{y}_i)_{i \in S} \mid (\mathbf{x}_i)_{i \in S}) \leq (1 - \delta)^m.$$

Taking a union bound over all realizable labelings yields, on the event $\{N_\mathcal{A} \geq (p_0/2)n\}$,

$$\mathbb{P}(\{(\mathbf{x}_i, y_i)\}_{i=1}^n \text{ is linearly separable} \mid (\mathbf{x}_i)_{i \in S}) \leq C_d \, m^{d+1} (1 - \delta)^m.$$

Since $m = \lfloor (p_0/2)n \rfloor = \Theta(n)$ and $\delta \in (0, 1/2)$, the exponential decay of $(1 - \delta)^m$ dominates the polynomial factor $m^{d+1}$. Hence there exist constants $c_1, c_2 > 0$, depending only on $d$ and $\|\mathbf{w}^\star\|$, such that

$$C_d \, m^{d+1} (1 - \delta)^m \leq c_1 e^{-c_2 n}.$$

Combining this bound with the Chernoff inequality for $\mathbb{P}(N_\mathcal{A} < (p_0/2)n)$ completes the proof.

*Non separability implies coercivity.* Fix any unit vector $\mathbf{u} \in \mathbb{S}^{d-1}$ and consider $\mathbf{w} = t\mathbf{u}$ with $t \to \infty$. Non separability means that for every $\mathbf{u}$ there exists an index $i$ such that $y_i \mathbf{x}_i^\top \mathbf{u} \leq 0$. For this $i$,

$$\log\big(1 + \exp(-y_i \mathbf{x}_i^\top (t\mathbf{u}))\big) = \log\big(1 + \exp(-t \, y_i \mathbf{x}_i^\top \mathbf{u})\big) \geq \log 2, \tag{49}$$

and if $y_i \mathbf{x}_i^\top \mathbf{u} < 0$ the term diverges linearly in $t$. Thus $\widehat{\mathcal{L}}(t\mathbf{u}) \to \infty$ for every direction $\mathbf{u}$, which is coercivity.

*Full column rank implies strict convexity and uniqueness.* For each $i$, the function $\mathbf{w} \mapsto \log(1 + \exp(-y_i \mathbf{x}_i^\top \mathbf{w}))$ is convex and twice differentiable. Hence

$$\nabla^2 \widehat{\mathcal{L}}(\mathbf{w}) = \frac{1}{n} \sum_{i=1}^{n} \sigma(z_i)(1 - \sigma(z_i)) \, \mathbf{x}_i \mathbf{x}_i^\top. \tag{50}$$

If $\mathbf{X}$ has full column rank, then for any nonzero $\mathbf{v} \in \mathbb{R}^d$,

$$\sum_{i=1}^{n} (\mathbf{v}^\top \mathbf{x}_i)^2 = \|\mathbf{X}\mathbf{v}\|^2 > 0. \tag{51}$$

Since all weights $\sigma(z_i)(1 - \sigma(z_i))$ are strictly positive, we obtain

$$\mathbf{v}^\top \nabla^2 \widehat{\mathcal{L}}(\mathbf{w}) \, \mathbf{v} > 0 \qquad \text{for all } \mathbf{v} \neq \mathbf{0}. \tag{52}$$

Thus $\widehat{\mathcal{L}}$ is strictly convex.

Combining strict convexity with coercivity, this implies existence and uniqueness of the global minimizer.

*The minimizer is not* $\mathbf{w}^\star$ *(by contradiction).* On the event that the data are not linearly separable and $X$ has full column rank, the empirical risk $\widehat{\mathcal{L}}$ is strictly convex and coercive, and therefore admits a unique global minimizer $\widehat{\mathbf{w}}$. We show that $\mathbf{w}^\star$ cannot be a stationary point of $\widehat{\mathcal{L}}$ under the sampled-label model (2). Assume for contradiction that $\mathbf{w}^\star$ is a stationary point of the hard-label empirical risk, i.e.

$$\nabla \widehat{\mathcal{L}}(\mathbf{w}^\star) = 0.$$

Then for every direction $\mathbf{v} \in \mathbb{R}^d$,

$$\mathbf{v}^\top \nabla \widehat{\mathcal{L}}(\mathbf{w}^\star) = 0.$$

Fix a coordinate direction $\mathbf{v} = e_j$. The stationarity condition implies

$$\sum_{i=1}^{n} x_{ij} \, y_i \, \sigma\big(-y_i \mathbf{x}_i^\top \mathbf{w}^\star\big) = 0. \tag{53}$$

Condition on the design matrix $X$. Under the data model (2),

$$\mathbb{P}(y_i = 1 \mid X) = \sigma(\mathbf{x}_i^\top \mathbf{w}^\star) =: p_i \in (0, 1).$$

Define the scalar random variable

$$Z_i := y_i \, \sigma\big(-y_i \mathbf{x}_i^\top \mathbf{w}^\star\big),$$

which satisfies

$$Z_i = \begin{cases} 1 - p_i, & y_i = 1, \\ -p_i, & y_i = -1, \end{cases} \qquad \mathbb{E}[Z_i \mid X] = 0, \qquad \mathrm{Var}(Z_i \mid X) = p_i(1 - p_i) > 0.$$

With this notation, (53) can be written as

$$S_j := \sum_{i=1}^{n} x_{ij} \, Z_i = 0.$$

We now compute the conditional variance of $S_j$ given $X$. Since the labels $\{y_i\}_{i=1}^{n}$ are conditionally independent,

$$\mathrm{Var}(S_j \mid X) = \sum_{i=1}^{n} x_{ij}^2 \, \mathrm{Var}(Z_i \mid X) = \sum_{i=1}^{n} x_{ij}^2 \, p_i(1 - p_i).$$

Because $X$ has full column rank, there exists at least one index $j$ such that $\sum_{i=1}^{n} x_{ij}^2 > 0$. Moreover, $p_i(1 - p_i) > 0$ for all $i$ under the logistic model. Hence

$$\mathrm{Var}(S_j \mid X) > 0.$$

However, if the stationarity condition were true, then $S_j$ would be identically equal to zero, which would force $\mathrm{Var}(S_j \mid X) = 0$. This is a contradiction.

Therefore, $\mathbf{w}^\star$ cannot be a stationary point of $\widehat{\mathcal{L}}$. Since $\widehat{\mathcal{L}}$ is strictly convex, its unique minimizer $\widehat{\mathbf{w}}$ must be the unique stationary point, and hence

$$\widehat{\mathbf{w}} \neq \mathbf{w}^\star.$$

$\square$

## D. Excess Risk Upper bound for the Spindly Network.

We consider a binary classification problem under a well-specified sparse logistic model. Let data $(\mathbf{x}_i, y_i) \in \mathbb{R}^d \times \{-1, +1\}$ be drawn i.i.d. from a distribution $\mathcal{D}$, where the covariates follow a general Gaussian distribution $\mathbf{x}_i \sim \mathcal{N}(\mathbf{0}, \mathbf{\Sigma})$ and labels are generated according to the conditional proability:

$$\mathbb{P}(y_i = 1 \mid \mathbf{x}_i) = \sigma(\langle \mathbf{x}_i, \mathbf{w}^* \rangle) \tag{54}$$

for a ground truth weight vector $\mathbf{w}^* \in \mathbb{R}^d$ that is $s$-sparse, i.e., $\|\mathbf{w}^*\|_0 = s < d$, $\|\mathbf{w}^*\|_2 < \infty$ and $\sigma(t) = 1/(1 + e^{-t})$ is the logistic sigmoid. If the feature vector lies entirely in the inactive subspace of $\mathbf{w}^\star$, the conditional label distribution reduces to $\mathbb{P}(y = 1 \mid \mathbf{x}) = 0.5$, yielding pure noise with no label information, so effective learning requires suppressing variance in these directions while recovering signal on the sparse active support.

**Population and Empirical risk.** For a parameter $\mathbf{w} \in \mathbb{R}^d$, we measure the population logistic risk as

$$\mathcal{L}(\mathbf{w}) := \mathbb{E}\big[\ell(y\langle \mathbf{x}, \mathbf{w}\rangle)\big], \quad \ell(t) := \log(1 + e^{-t}), \tag{55}$$

Under the well-specified sparse data model in (54), $\mathcal{L}(\mathbf{w})$ is strictly convex and admits a unique and finite global minimizer at $\mathbf{w}^\star$. Given samples $\{(\mathbf{x}_i, y_i)\}_{i=1}^n$, the empirical risk is the average over the $n$ observed samples

$$\widehat{\mathcal{L}}(\mathbf{w}) := \frac{1}{n} \sum_{i=1}^n \ell(y_i \langle \mathbf{x}_i, \mathbf{w}\rangle). \tag{56}$$

In the overdetermined case $n > d$, the empirical risk is also strictly convex, admits a finite global minimizer, is coercive and the data pair $\{(\mathbf{x}_i, y_i)\}_{i=1}^n$ is non-separable with high probability under the specified data-model in (54).

We study the training dynamics of a non-rotation invariant algorithm, where we reparameterize the weight vector as Hadamard scalar product of two vectors as $\mathbf{w}(t) = \mathbf{u}(t) \odot \mathbf{v}(t)$ to minimize the empirical risk $\widehat{\mathcal{L}}(\mathbf{w}(t))$ and perform gradient flow on $\mathbf{u}$ and $\mathbf{v}$ as

$$\dot{\mathbf{u}}(t) = -\nabla_{\mathbf{u}}\widehat{\mathcal{L}}(\mathbf{w}(t)), \quad \dot{\mathbf{v}}(t) = -\nabla_{\mathbf{v}}\widehat{\mathcal{L}}(\mathbf{w}(t)). \tag{57}$$

By the conservation law property of gradient flow, the joint dynamics on $\mathbf{u}$ and $\mathbf{v}$ lead to following dynamics on $\mathbf{w}$:

$$\dot{\mathbf{w}}(t) = -|\mathbf{w}(t)| \odot \widehat{\mathcal{L}}(\mathbf{w}(t)) \tag{58}$$

with balanced initialization $\mathbf{u}(0) = \mathbf{v}(0) = \alpha\mathbf{1}$.

**Lemma D.1.** *Let $\widehat{\mathcal{L}} : \mathbb{R}^d \to \mathbb{R}$ be continuously and the spindly network has the reparamaterization*

$$\boldsymbol{w}(t) = \boldsymbol{u}(t) \odot \boldsymbol{v}(t).$$

*Gradient flow on the variables $(\boldsymbol{u}, \boldsymbol{v})$ with balanced initialization $\mathbf{u}(0) = \mathbf{v}(0) = \alpha\mathbf{1}$:*

$$\dot{\boldsymbol{u}}(t) = -\nabla_{\boldsymbol{u}}\widehat{\mathcal{L}}(\boldsymbol{w}(t)), \qquad \dot{\boldsymbol{v}}(t) = -\nabla_{\boldsymbol{v}}\widehat{\mathcal{L}}(\boldsymbol{w}(t)). \tag{59}$$

*induce the dynamics on the original predictor*

$$\dot{\boldsymbol{w}}(t) = -|\boldsymbol{w}(t)| \odot \nabla_{\boldsymbol{w}}\widehat{\mathcal{L}}(\boldsymbol{w}(t)).$$

*Proof.* We first compute the gradients with respect to $\boldsymbol{u}$ and $\boldsymbol{v}$. By the chain rule and the relation $\boldsymbol{w} = \boldsymbol{u} \odot \boldsymbol{v}$, for each coordinate $j$,

$$\frac{\partial \widehat{\mathcal{L}}(\boldsymbol{u} \odot \boldsymbol{v})}{\partial u_j} = \frac{\partial \widehat{\mathcal{L}}(\boldsymbol{w})}{\partial w_j} v_j, \qquad \frac{\partial \widehat{\mathcal{L}}(\boldsymbol{u} \odot \boldsymbol{v})}{\partial v_j} = \frac{\partial \widehat{\mathcal{L}}(\boldsymbol{w})}{\partial w_j} u_j.$$

Hence,

$$\nabla_{\boldsymbol{u}} \widehat{\mathcal{L}}(\boldsymbol{u} \odot \boldsymbol{v}) = \boldsymbol{v} \odot \nabla_{\boldsymbol{w}} \widehat{\mathcal{L}}(\boldsymbol{w}), \qquad \nabla_{\boldsymbol{v}} \widehat{\mathcal{L}}(\boldsymbol{u} \odot \boldsymbol{v}) = \boldsymbol{u} \odot \nabla_{\boldsymbol{w}} \widehat{\mathcal{L}}(\boldsymbol{w}).$$

The gradient flow equations therefore become

$$\dot{\boldsymbol{u}} = -\boldsymbol{v} \odot \nabla_{\boldsymbol{w}} \widehat{\mathcal{L}}(\boldsymbol{w}), \qquad \dot{\boldsymbol{v}} = -\boldsymbol{u} \odot \nabla_{\boldsymbol{w}} \widehat{\mathcal{L}}(\boldsymbol{w}).$$

Differentiating $\boldsymbol{w} = \boldsymbol{u} \odot \boldsymbol{v}$ and using the product rule yields

$$\dot{\boldsymbol{w}} = \dot{\boldsymbol{u}} \odot \boldsymbol{v} + \boldsymbol{u} \odot \dot{\boldsymbol{v}}.$$

Substituting the expressions above gives

$$\dot{\boldsymbol{w}} = -(\boldsymbol{v}^{\odot 2} + \boldsymbol{u}^{\odot 2}) \odot \nabla_{\boldsymbol{w}} \widehat{\mathcal{L}}(\boldsymbol{w}),$$

which proves (i).

As gradient flow preserves the balancedness:

$$\frac{d}{dt}(\boldsymbol{u}^{\odot 2} - \boldsymbol{v}^{\odot 2}) = 2\boldsymbol{u} \odot \dot{\boldsymbol{u}} - 2\boldsymbol{v} \odot \dot{\boldsymbol{v}} = -2(\boldsymbol{u} \odot \boldsymbol{v}) \odot \nabla_{\boldsymbol{w}} \widehat{\mathcal{L}}(\boldsymbol{w}) + 2(\boldsymbol{v} \odot \boldsymbol{u}) \odot \nabla_{\boldsymbol{w}} \widehat{\mathcal{L}}(\boldsymbol{w}) = \boldsymbol{0}.$$

Hence $\boldsymbol{u}^{\odot 2} - \boldsymbol{v}^{\odot 2}$ is conserved. Finally, if $\boldsymbol{u}(0) = \boldsymbol{v}(0)$, we have $\boldsymbol{u}(t)^{\odot 2} = \boldsymbol{v}(t)^{\odot 2}$ for all $t$. By symmetry of the ODE system and uniqueness of solutions, this implies $\boldsymbol{u}(t) = \boldsymbol{v}(t)$ for all $t$. In this case,

$$\boldsymbol{w} = \boldsymbol{u}^{\odot 2}, \qquad \boldsymbol{u}^{\odot 2} + \boldsymbol{v}^{\odot 2} = 2\,\boldsymbol{u}^{\odot 2} = 2\,|\boldsymbol{w}|,$$

which yields

$$\dot{\boldsymbol{w}}(t) = -|\boldsymbol{w}(t)| \odot \nabla_{\boldsymbol{w}} \widehat{\mathcal{L}}(\boldsymbol{w}(t)).$$

$\square$

**Theorem D.2.** *Define the population gradient $\mathbf{g}(\mathbf{w}) := \nabla \mathcal{L}(\mathbf{w})$ and empirical gradient $\widehat{\mathbf{g}}_n(\mathbf{w}) := \nabla \widehat{\mathcal{L}}(\mathbf{w})$. Let $S := \mathrm{supp}(\mathbf{w}^\star)$ and denote by $\mathbf{x}_S$ the subvector of $\mathbf{x}$ on $S$, denoted from the sparse data-generation model* (54).

*(i) Exact population gradient: For every $\mathbf{w} \in \mathbb{R}^d$ and each coordinate $j \in [d]$,*

$$g_j(\mathbf{w}) = \mathbb{E}_{\mathbf{x}}\left[ x_j \left( \sigma(\mathbf{x}^\top \mathbf{w}) - \sigma(\mathbf{x}_S^\top \mathbf{w}_S^\star) \right) \right]. \tag{60}$$

*Moreover, with $\sigma'(t) = \sigma(t)(1 - \sigma(t))$*

$$a(\mathbf{w}) := \mathbb{E}_{\mathbf{x}}\left[ \sigma'(\mathbf{x}^\top \mathbf{w}) \right], \quad a^\star := \mathbb{E}_{\mathbf{x}}\left[ \sigma'(\mathbf{x}_S^\top \mathbf{w}_S^\star) \right] \tag{61}$$

*we have the gradient coordinate-wise forms by Stein's lemma:*

$$g_j(\mathbf{w}) = w_j\, a(\mathbf{w}), \qquad\qquad\qquad j \in S^c, \tag{62}$$
$$g_j(\mathbf{w}) = w_j\, a(\mathbf{w}) \ - \ w_j^\star\, a^\star, \qquad\qquad j \in S. \tag{63}$$

*(ii) Sampling noise: Define the gradient noise vector*

$$\boldsymbol{\zeta}(\mathbf{w}) := \widehat{\mathbf{g}}_n(\mathbf{w}) - \mathbf{g}(\mathbf{w}) \in \mathbb{R}^d, \qquad \zeta_j(\mathbf{w}) := \mathbf{e}_j^\top \boldsymbol{\zeta}(\mathbf{w}).$$

*Assume that for the fixed $\mathbf{w}$ of interest, each $\zeta_j(\mathbf{w})$ is the average of $n$ i.i.d. centered sub-Gaussian random variables. Then for any $\delta \in (0, 1)$, with probability at least $1 - \delta$,*

$$\max_{1 \leq j \leq d} |\zeta_j(\mathbf{w})| \ \leq \ 4\sqrt{\frac{\log(2d/\delta)}{n}}. \tag{64}$$

*Proof.* The population logistic gradient is given by

$$\nabla L(\boldsymbol{w}) = \mathbb{E}_{\boldsymbol{x},y}\big[-y\boldsymbol{x}\,\sigma(-y\boldsymbol{x}^\top\boldsymbol{w})\big],$$

hence for each $j \in [d]$,

$$g_j(\boldsymbol{w}) = \mathbb{E}_{\boldsymbol{x},y}\big[-yx_j\,\sigma(-y\boldsymbol{x}^\top\boldsymbol{w})\big]. \tag{65}$$

Apply iterated expectation using the Tower property to (65):

$$g_j(\boldsymbol{w}) = \mathbb{E}_{\boldsymbol{x}}\big[\mathbb{E}_{y|\boldsymbol{x}}\big[-yx_j\,\sigma(-y\boldsymbol{x}^\top\boldsymbol{w})\mid\boldsymbol{x}\big]\big].$$

Since this is conditioned on $\boldsymbol{x}$, we pull out $x_j$:

$$g_j(\boldsymbol{w}) = \mathbb{E}_{\boldsymbol{x}}\big[x_j\,\mathbb{E}_{y|\boldsymbol{x}}\big[-y\,\sigma(-y\boldsymbol{x}^\top\boldsymbol{w})\mid\boldsymbol{x}\big]\big]. \tag{66}$$

Writing $p(\boldsymbol{x}) := \mathbb{P}(y = 1 \mid \boldsymbol{x})$ and since $y \in \{-1,+1\}$,

$$\begin{aligned}
\mathbb{E}_{y|\boldsymbol{x}}\big[-y\,\sigma(-y\boldsymbol{x}^\top\boldsymbol{w})\mid\boldsymbol{x}\big] &= (-1)\sigma(-\boldsymbol{x}^\top\boldsymbol{w})\,p(\boldsymbol{x}) + (+1)\sigma(\boldsymbol{x}^\top\boldsymbol{w})\,(1 - p(\boldsymbol{x}))\\
&= -\sigma(-\boldsymbol{x}^\top\boldsymbol{w})p(\boldsymbol{x}) + \sigma(\boldsymbol{x}^\top\boldsymbol{w})\big(1 - p(\boldsymbol{x})\big).
\end{aligned} \tag{67}$$

Using $\sigma(-\boldsymbol{x}^\top\boldsymbol{w}) = 1 - \sigma(\boldsymbol{x}^\top\boldsymbol{w})$,

$$\begin{aligned}
-\sigma(-\boldsymbol{x}^\top\boldsymbol{w})p(\boldsymbol{x}) + \sigma(\boldsymbol{x}^\top\boldsymbol{w})\big(1 - p(\boldsymbol{x})\big) &= -(1 - \sigma(\boldsymbol{x}^\top\boldsymbol{w}))p(\boldsymbol{x}) + \sigma(\boldsymbol{x}^\top\boldsymbol{w}) - \sigma(z)p(\boldsymbol{x})\\
&= \sigma(\boldsymbol{x}^\top\boldsymbol{w}) - p(\boldsymbol{x}).
\end{aligned} \tag{68}$$

Under the logistic generative model, $p(\boldsymbol{x}) = \sigma(\boldsymbol{x}_S^\top\boldsymbol{w}_S^\star)$. Therefore,

$$\mathbb{E}_{y|\boldsymbol{x}}\big[-y\,\sigma(-y\boldsymbol{x}^\top\boldsymbol{w})\mid\boldsymbol{x}\big] = \sigma(\boldsymbol{x}^\top\boldsymbol{w}) - \sigma(\boldsymbol{x}_S^\top\boldsymbol{w}_S^\star). \tag{69}$$

Substitute (69) into (66) to obtain

$$g_j(\boldsymbol{w}) = \mathbb{E}_{\boldsymbol{x}}\Big[x_j\big(\sigma(\boldsymbol{x}^\top\boldsymbol{w}) - \sigma(\boldsymbol{x}_S^\top\boldsymbol{w}_S^\star)\big)\Big],$$

which proves (60).

Also when $j \in S^c$. Since $x_j$ is independent of $\boldsymbol{x}_S$ and $\mathbb{E}[x_j] = 0$,

$$\mathbb{E}_{\boldsymbol{x}}\big[x_j\,\sigma(\boldsymbol{x}_S^\top\boldsymbol{w}_S^\star)\big] = \mathbb{E}[x_j]\cdot\mathbb{E}_{\boldsymbol{x}_S}\big[\sigma(\boldsymbol{x}_S^\top\boldsymbol{w}_S^\star)\big] = 0.$$

Thus from (60),

$$g_j(\boldsymbol{w}) = \mathbb{E}_{\boldsymbol{x}}\big[x_j\,\sigma(\boldsymbol{x}^\top\boldsymbol{w})\big].$$

*Stein's Lemma reduction of exact population gradient*

We notice that (60) is of the form $\mathbb{E}(xf(x))$ with $x$ being a standard normal, so this can be reduced $\mathbb{E}(f'(x))$ if $f$ is smooth and differentiable.

Furthermore in (60), $\mathbf{x}$ is i.i.d, hence writing $\boldsymbol{x}^\top\boldsymbol{w} = w_jx_j + Z$ where $Z := \sum_{k\neq j} w_kx_k$ is independent of $x_j$. Conditioning on $Z$ and using Stein's lemma for $x_j \sim \mathcal{N}(0,1)$,

$$\mathbb{E}[x_j\,\sigma(w_jx_j + Z)\mid Z] = w_j\,\mathbb{E}[\sigma'(w_jx_j + Z)\mid Z].$$

So, we apply the Stein's lemma, and derive when $i \in S$:

$$g_j(\boldsymbol{w}) = \mathbb{E}_{\boldsymbol{x}}\Big[x_j\big(\sigma(\boldsymbol{x}^\top\boldsymbol{w}) - \sigma(\boldsymbol{x}_S^\top\boldsymbol{w}_S^\star)\big)\Big] = w_j\,\mathbb{E}_{\boldsymbol{x}}\big[\sigma'(\boldsymbol{x}^\top\boldsymbol{w})\big] - w_j^\star\,\mathbb{E}_{\boldsymbol{x}}\big[\sigma'(\boldsymbol{x}_S^\top\boldsymbol{w}_S^\star)\big] \tag{70}$$

Since when $i \in S^c$, $w_j^\star = 0$, the gradient reduces to

$$g_j(\boldsymbol{w}) = \mathbb{E}_{\boldsymbol{x}}\big[x_j\sigma(\boldsymbol{x}^\top\boldsymbol{w})\big] = w_j\,\mathbb{E}_{\boldsymbol{x}}\big[\sigma'(\boldsymbol{x}^\top\boldsymbol{w})\big] \tag{71}$$

*Coordinate-wise sampling noise*

Assume $\boldsymbol{x} \sim \mathcal{N}(\boldsymbol{0}, \boldsymbol{I}_d)$ and $Y \in \{-1, +1\}$. A single-sample coordinate gradient contribution

$$g_j(\boldsymbol{w}; \boldsymbol{x}, y) := -y\, x_j\, \sigma(-y\boldsymbol{x}^\top \boldsymbol{w}),$$

and the centered version

$$z_j(\boldsymbol{w}; \boldsymbol{x}, y) := g_j(\boldsymbol{w}; \boldsymbol{x}, y) - \mathbb{E}[g_j(\boldsymbol{w}; \boldsymbol{x}, y)].$$

Since $0 < \sigma(\cdot) < 1$ and $|y| = 1$, we have

$$|g_j(\boldsymbol{w}; \boldsymbol{x}, y)| = |y|\, |x_j|\, \sigma(-y\boldsymbol{x}^\top \boldsymbol{w}) \le |x_j|.$$

Because $x_j \sim \mathcal{N}(0, 1)$,

$$\mathbb{P}(|g_j(\boldsymbol{w}; \boldsymbol{x}, Y)| \ge t) \le \mathbb{P}(|x_j| \ge t) \le 2e^{-t^2/2}, \qquad \forall\, t \ge 0.$$

The tail bound above implies that $g_j(\boldsymbol{w}; \boldsymbol{x}, y)$ is sub-Gaussian (with an absolute-constant proxy). Concretely, one may take the mgf bound

$$\mathbb{E}\left[e^{\lambda g_j(\boldsymbol{w}; \boldsymbol{x}, y)}\right] \le e^{2\lambda^2}, \qquad \forall\, \lambda \in \mathbb{R},$$

so $g_j(\boldsymbol{w}; \boldsymbol{x}, y)$ is sub-Gaussian with variance proxy 4.

Centering preserves sub-Gaussianity up to an absolute constant. In particular, one can take

$$\mathbb{E}\left[e^{\lambda z_j(\boldsymbol{w}; \boldsymbol{x}, y)}\right] \le e^{4\lambda^2} = \exp\left(\frac{8\lambda^2}{2}\right), \qquad \forall\, \lambda \in \mathbb{R},$$

so $z_j(\boldsymbol{w}; \boldsymbol{x}, y)$ is sub-Gaussian with variance proxy 8.

Since $(z_j(\boldsymbol{w}; \boldsymbol{x}_i, y_i))_{i=1}^n$ are i.i.d. centered sub-Gaussian with variance proxy 8, their average satisfies: for any $\delta \in (0, 1)$,

$$\mathbb{P}\left(|z_j(\boldsymbol{w})| \ge \sqrt{\frac{16 \log(2/\delta)}{n}}\right) \le \delta.$$

Equivalently, with probability at least $1 - \delta$,

$$|z_j(\boldsymbol{w})| \le 4\sqrt{\frac{\log(2/\delta)}{n}}.$$

Apply a union bound over $j = 1, \ldots, d$ (replace $\delta$ by $\delta/d$ in Step 4): with probability at least $1 - \delta$,

$$\max_{1 \le j \le d} |z_j(\boldsymbol{w})| \le 4\sqrt{\frac{\log(2d/\delta)}{n}}.$$

We denote this event as $\mathcal{E}$. $\qquad \square$

**Lemma D.3.** *Under the spindly parameterization with balanced initialization (D.1), the dynamics of logistic regression is represented by the following state-dependent coupled Riccati equations:*

$$\dot{w}_i(t) = \zeta_i\, w_i(t) - a(\mathbf{w}(t))\, w_i(t)^2, \qquad\qquad i \in S^c,$$
$$\dot{w}_i(t) = \left(w_i^\star a^\star + \zeta_i\right) w_i(t) - a(\mathbf{w}(t))\, w_i(t)^2, \qquad\qquad i \in S. \qquad (72)$$

*where* $a(\mathbf{w}(t)) := \mathbb{E}_{\boldsymbol{x}}\left[\sigma'(\boldsymbol{x}^\top \boldsymbol{w}(t))\right]$, $a^\star := \mathbb{E}_{\boldsymbol{x}}\left[\sigma'(\mathbf{x}_S^\top \mathbf{w}_S^\star)\right]$ *and* $\max_{1 \le j \le d} |\zeta_j(\boldsymbol{w})| \le 4\sqrt{\frac{\log(2d/\delta)}{n}}$ *occuring with probability* $(1 - \delta)$.

*Proof.* Decomposing the empirical gradient as population gradient and sampling noise $\hat{\mathcal{L}}(\mathbf{w}) = \mathcal{L}(\mathbf{w}) + \zeta$, we put this expression into the spindly dynamics obtained in Lemma 58 with the Stein's Lemma reduced analytical expression for population gradient and obtain (26). $\qquad \square$

**Theorem D.4.** *Consider the spindly dynamics* (26) *on logistic regression with initialization* $\mathbf{u}(0) = \mathbf{v}(0) = \frac{1}{\sqrt{d}}\mathbf{1}$ *and on the event* $\mathcal{E}$ *where* $\gamma = \max_{1 \le j \le d} |z_j(\boldsymbol{w})| \le 4\sqrt{\frac{\log(2d/\delta)}{n}}$ *occuring with probability* $(1 - \delta)$. *Assume the following conditions hold:*

1. *(Signal-Noise separation) There exists* $\delta \in (0, \frac{1}{2})$ *such that*

$$\frac{\gamma}{w^\star_{\min} a^\star} \le \frac{1}{2} - \delta.$$

2. *(Curvature non-saturation) There exists* $\epsilon \in (0, 1)$ *such that*

$$a^\star \ge \frac{1 - \epsilon}{4}.$$

*Then there exist absolute constants* $c_1, c_2 > 0$ *such that, for all times in the interval* $t \in \left[\frac{\log d}{2w^\star_{\min} a^\star} - c_1, \frac{\log d}{2w^\star_{\min} a^\star} + c_1\right]$, *the spindly trajectory satisfies*

$$\|\mathbf{w}(t) - \mathbf{w}^\star\|^2_2 \le \epsilon^2 \|\mathbf{w}^\star\|^2_2 + \frac{16s\log(\frac{2d}{\delta})}{n\,(w^\star_{\min} a^\star)^2} + \frac{c_2}{d^{\frac{1}{2} + \delta}}. \tag{73}$$

*Proof.* The objective of the proof is to show that the spindly dynamics amplifies the growth of the signal coorindates $i \in S$ and suppress the growth of the inactive coordinates $i \in S^c$ as compared to any rotation invariant algorithm.

1) *Monotonicity and identifiability of envelope ODE's:* The system of ODE's in (29) do not admit a closed form solution, however, we first show that for each index $i$, there exist two monotonic ODE's that admit closed form solution of the form:

$$\dot{w}^{\text{up}}_i(t) = \left(w^\star_i a^\star + \zeta_i\right) w^{\text{up}}_i(t) - a^\star\, w^{\text{up}}_i(t)^2 \tag{74}$$

$$\dot{w}^{\text{low}}_i(t) = \left(w^\star_i a^\star + \zeta_i\right) w^{\text{low}}_i(t) - \frac{1}{4}\, w^{\text{low}}_i(t)^2 \tag{75}$$

In Lemma D.6, we prove that $w^{\text{low}}_i(t) \le w_i(t) \le w^{\text{up}}_i(t)$ for all $0 \le t \le T$, with both envelope trajectories monotone on this interval. Furthermore, both these ODE's have a closed form solution in

$$w^{\text{up}}_i(t) = \frac{w^\star_i + \frac{\zeta_i}{a^\star}}{1 + \left(d\left(w^\star_i + \frac{\zeta_i}{a^\star}\right) - 1\right)e^{-(w^\star_i a^\star + \zeta_i)t}}. \tag{76}$$

$$w^{\text{low}}_i(t) = \frac{4\left(w^\star_i a^\star + \zeta_i\right)}{1 + \left(4d\left(w^\star_i a^\star + \zeta_i\right) - 1\right)e^{-(w^\star_i a^\star + \zeta_i)t}}. \tag{77}$$

An analogous construction applies to the coordinates in $i \in S^c$.

2) *Early-stopping time estimate:* We prove that at a well chosen time $T$, the signal coordinates $i \in S$, grows close to their target $w^\star_i$ while the noise coordinates remain suppressed. In fact, we chose a time so that all the active coordinates $i \in S$, reaches atleast $\epsilon$ close to their respective target $w^\star_i$. And at the same time, the noise coordinates remain small. We choose the stopping time so that the lower envelope of the weakest active coordinate reaches a prescribed fraction of its target value. Stating $w^\star_{\min} := \min_{i \in S} w^\star_i$, $i_{\min} \in \arg\min_{i \in S} w^\star_i$, we define the stopping time $T(\varepsilon)$ as:

$$T(\epsilon) := \frac{1}{2w^\star_{\min} a^\star} \log\left(\frac{4w^\star_{\min} a^\star d - 1}{\frac{4a^\star}{(1-\epsilon)w^\star_{\min}} - 1}\right). \tag{78}$$

when the lower-envelope of coordinate $i_{\min}$ reaches $(1 - \epsilon)$ of target $w^\star_{\min}$, that is

$$w^{\text{low}}_{i_{\min}}(T(\epsilon)) = (1 - \epsilon)\, w^\star_{\min}$$

Furthermore, to ensure reachability of $w_{i_{\min}}^{\text{low}}(t)$ to the level $(1 - \epsilon)\, w_{\min}^\star$, we require

$$\lim_{t \to \infty} w_{i_{\min}}^{\text{low}}(t) = 4(w_{\min}^\star a^\star + \zeta_i) \geq (1 - \epsilon)w_{\min}^\star \tag{79}$$

which is exactly Assumption A(2). In Lemma D.7, we show that once the weakest active coordinate reaches $(1 - \epsilon)$ of its target, all other lower-envelope active coordinates are automatically closer to their respective targets, achieving a strictly smaller relative error. From the lower-envelope and upper-envelope we get that for $i_{\min}$,

$$(1 - \epsilon)w_{\min}^\star \leq w_{i_{\min}}(t) \leq w_{\min}^\star + \frac{\zeta_{i_{\min}}}{a^\star w_{\min}^\star} \tag{80}$$

and similarly for all the other active coordinates $i \in S$, we have for some $\epsilon_S < \epsilon$

$$(1 - \epsilon_S)w_{\min}^\star \leq w_i(t) \leq w_i^\star + \frac{\zeta_i}{a^\star w_i^\star} \tag{81}$$

Hence, summing over all the active coorindates gives:

$$\sum_{i \in S}(w_i(T) - w_i^*)^2 \leq \epsilon^2 \, \|\mathbf{w}^\star\|_2^2 + \frac{16s \log(\frac{2d}{\delta})}{n\,(w_{\min}^\star a^\star)^2} \tag{82}$$

*3) Inactive coordinate suppression:* Fix an inactive index $i \in S^c$. By the envelope comparison for inactive coordinates, we have $0 \leq w_i(t) \leq w_i^{\text{up}}(t)$ for all $t \in [0, T]$, where the inactive upper envelope admits the closed form

$$w_i^{\text{up}}(t) = \frac{\zeta_i/a^\star}{1 + \left(\frac{\zeta_i d}{a^\star} - 1\right)e^{-\zeta_i t}}.$$

Since

$$1 + \left(\frac{\zeta_i d}{a^\star} - 1\right)e^{-\zeta_i t} \geq \left(\frac{\zeta_i d}{a^\star}\right)e^{-\zeta_i t},$$

we obtain the bound

$$w_i^{\text{up}}(t) \leq \frac{1}{d}e^{\zeta_i t}$$

which is the inequality used in (34). Evaluating at $t = T(\varepsilon)$ and squaring gives

$$w_i^2(T) \leq \frac{1}{d^2}e^{2\zeta_i T(\varepsilon)}.$$

Using assumption (A1), we have $\zeta_i \leq \gamma$ for all $i \in S^c$ and $\frac{\gamma}{w_{\min}^\star a^\star} \leq \frac{1}{2} - \delta$. With the definition of the stopping time $T(\varepsilon)$,

$$T(\varepsilon) = \frac{1}{2w_{\min}^\star a^\star} \log\left(\frac{\frac{4w_{\min}^\star a^\star d - 1}{\frac{4a^\star}{(1-\varepsilon)w_{\min}^\star} - 1}}{}\right) \leq \frac{\log d}{2w_{\min}^\star a^\star} + C_\varepsilon,$$

where $C_\varepsilon = \frac{1}{2w_{\min}^\star a^\star} \log\left(\frac{4w_{\min}^\star a^\star}{\frac{4a^\star}{(1-\varepsilon)w_{\min}^\star} - 1}\right) > 0$ (due to Assumption-2) and depends only on $\varepsilon$. Therefore,

$$2\zeta_i T(\varepsilon) \leq \frac{\gamma}{w_{\min}^\star a^\star} \log d + 2\gamma C_\varepsilon \leq (\frac{1}{2} - \delta)\log d + 2\gamma C_\varepsilon,$$

and hence

$$w_i^2(T) \leq \frac{e^{2\gamma C_\varepsilon}}{d^{\frac{3}{2}+\delta}}.$$

Summing over all $i \in S^c$ and using $|S^c| \leq d$ yields

$$\sum_{i \in S^c} w_i^2(T) \leq \frac{c_2}{d^{\frac{1}{2}+\delta}} \tag{83}$$

for a constant $c_2 = e^{2\gamma C_\varepsilon} > 0$ depending on $\varepsilon$.

*4) Combinining the coorindate:* Finally combining (83) and (82), we get the final upper bound as

$$\|\mathbf{w}(t) - \mathbf{w}^\star\|_2^2 \leq \epsilon^2 \|\mathbf{w}^\star\|_2^2 + \frac{16s \log(\frac{2d}{\delta})}{n (w^\star_{\min} a^\star)^2} + \frac{c_2}{d^{\frac{1}{2}+\delta}}. \tag{84}$$

$\square$

**Lemma D.5.** *Let $\boldsymbol{x} \in \mathbb{R}^d$ is isotropic, there exist constants $\rho \in (0,1]$ and $\tau > 0$ such that for all $\boldsymbol{u} \in \mathbb{S}^{d-1}$,*

$$\mathbb{P}(|\boldsymbol{u}^\top \boldsymbol{x}| \leq \tau) \geq \rho.$$

*Due to loss-coercivity and monotonicity under non-separable condition, we have on the trajectory $\sup_{t \in [0,T]} \|\boldsymbol{w}(t)\|_2 \leq R$ for some $R > 0$. Then for all $t \in [0,T]$,*

$$\frac{1}{4} \geq a(\boldsymbol{w}(t)) \geq a^\star, \qquad a^\star := \rho\, \sigma'(\tau R) > 0.$$

*Proof.* The upper bound $a(\boldsymbol{w}) \leq \frac{1}{4}$ holds for all $\boldsymbol{w}$ since $\sigma'(t) \leq \frac{1}{4}$ pointwise for all $t \in \mathbb{R}$.

For the lower bound, fix $\boldsymbol{w}$ with $\|\boldsymbol{w}\|_2 \leq R$ and write $\boldsymbol{w} = \|\boldsymbol{w}\|_2\, \boldsymbol{u}$ with $\boldsymbol{u} \in \mathbb{S}^{d-1}$. Then $\boldsymbol{x}^\top \boldsymbol{w} = \|\boldsymbol{w}\|_2\, \boldsymbol{u}^\top \boldsymbol{x}$. Since $\sigma'$ is even and decreasing on $[0, \infty)$, on the event $\{|\boldsymbol{u}^\top \boldsymbol{x}| \leq \tau\}$ we have

$$\sigma'(\boldsymbol{x}^\top \boldsymbol{w}) \geq \sigma'(\tau \|\boldsymbol{w}\|_2) \geq \sigma'(\tau R).$$

Therefore,

$$a(\boldsymbol{w}) = \mathbb{E}[\sigma'(\boldsymbol{x}^\top \boldsymbol{w})] \geq \mathbb{E}[\sigma'(\boldsymbol{x}^\top \boldsymbol{w})\, \mathbf{1}_{\{|\boldsymbol{u}^\top \boldsymbol{x}| \leq \tau\}}] \geq \sigma'(\tau R)\, \mathbb{P}(|\boldsymbol{u}^\top \boldsymbol{x}| \leq \tau) \geq \rho\, \sigma'(\tau R) = a^\star.$$

Applying this bound pointwise to $\boldsymbol{w}(t)$ yields the claim. $\square$

**Lemma D.6.** *For the system of ODE's in (58) for each index $w_i(t)$ ($i \in S$), there exists monotonic ODE's $w_i^{low}(t)$ and $w_i^{up}(t)$:*

$$\dot{w}_i^{up}(t) = (w_i^\star a^\star + \zeta_i)\, w_i^{up}(t) - a^\star\, w_i^{up}(t)^2 \tag{85}$$

$$\dot{w}_i^{low}(t) = (w_i^\star a^\star + \zeta_i)\, w_i^{low}(t) - \frac{1}{4}\, w_i^{low}(t)^2 \tag{86}$$

*such that $w_i^{low}(t) \leq w_i(t) \leq w_i^{up}(t)$ for all $0 \leq t \leq T$.*

*Proof.* From Lemma D.5, along the trajectory we have the curvature strip

$$a^\star \leq a(\boldsymbol{w}(t)) \leq \frac{1}{4}, \qquad \forall\, t \in [0,T].$$

Fix an index $i \in S$ and write the $i$th coordinate ODE in (58) in the form

$$\dot{w}_i(t) = (w_i^\star a^\star + \zeta_i)\, w_i(t) - a(\boldsymbol{w}(t))\, w_i(t)^2, \qquad t \in [0,T]. \tag{87}$$

Using $a(\boldsymbol{w}(t)) \leq \frac{1}{4}$ and $w_i(t)^2 \geq 0$, we obtain the pointwise differential inequality

$$\dot{w}_i(t) = (w_i^\star a^\star + \zeta_i)\, w_i(t) - a(\boldsymbol{w}(t))\, w_i(t)^2 \geq (w_i^\star a^\star + \zeta_i)\, w_i(t) - \frac{1}{4}\, w_i(t)^2.$$

Similarly, using $a(\boldsymbol{w}(t)) \geq a^\star$ we obtain

$$\dot{w}_i(t) \leq (w_i^\star a^\star + \zeta_i)\, w_i(t) - a^\star\, w_i(t)^2.$$

Define $w_i^{low}$ and $w_i^{up}$ as the solutions of

$$\dot{w}_i^{low}(t) = (w_i^\star a^\star + \zeta_i)\, w_i^{low}(t) - \frac{1}{4}\, (w_i^{low}(t))^2, \qquad w_i^{low}(0) = w_i(0),$$

and

$$\dot{w}_i^{\mathrm{up}}(t) = \left(w_i^\star a^\star + \zeta_i\right) w_i^{\mathrm{up}}(t) - a^\star \left(w_i^{\mathrm{up}}(t)\right)^2, \qquad w_i^{\mathrm{up}}(0) = w_i(0).$$

The right-hand sides are locally Lipschitz in the scalar state variable, so solutions exist and are unique on $[0, T]$. By the standard comparison principle for scalar ODEs, the above differential inequalities together with the common initial condition imply

$$w_i^{\mathrm{low}}(t) \;\le\; w_i(t) \;\le\; w_i^{\mathrm{up}}(t), \qquad \forall\, t \in [0, T].$$

This proves the claim. □

**Lemma D.7.** *Let $i_{\min} \in \arg\min_{i \in S} w_i^\star$ and fix $\varepsilon \in (0, 1)$. If at some time $T$,*

$$w_{i_{\min}}^{\mathrm{low}}(T) = (1 - \varepsilon)\, w_{\min}^\star, \qquad w_{\min}^\star := \min_{i \in S} w_i^\star,$$

*then for every $i \in S$,*

$$\left| w_i^{\mathrm{low}}(T) - w_i^\star \right| \;\le\; \varepsilon\, w_i^\star.$$

*Proof.* Fix $i \in S$ and write $\beta = \beta_i$. The ODE

$$\dot{w}(t) = \beta w(t) - \frac{1}{4} w(t)^2, \qquad w(0) = \frac{1}{d},$$

has the explicit solution

$$w(t) = \frac{4\beta}{1 + (4d\beta - 1)e^{-\beta t}}.$$

Define the relative gap to the equilibrium level $4\beta$ by

$$r_\beta(t) := 1 - \frac{w(t)}{4\beta} = \frac{(4d\beta - 1)e^{-\beta t}}{1 + (4d\beta - 1)e^{-\beta t}},$$

which is decreasing in $t$. Equivalently,

$$\frac{r_\beta(t)}{1 - r_\beta(t)} = (4d\beta - 1)e^{-\beta t}.$$

Let $\beta_{\min} = \min_{k \in S} \beta_k$ and choose $j \in \arg\min_{k \in S} \beta_k$, so $\beta_j = \beta_{\min}$. Since $w_j^{\mathrm{low}}(T) = (1 - \varepsilon)w_{\min}^\star \le (1 - \varepsilon)w_j^\star$ and $w_j^{\mathrm{low}}(t)$ is increasing to $4\beta_{\min}$, we have $r_{\beta_{\min}}(T) \le \varepsilon$. Using the odds-ratio identity at time $T$ and comparing $\beta$ with $\beta_{\min}$ yields

$$\frac{r_\beta(T)}{1 - r_\beta(T)} = \frac{4d\beta - 1}{4d\beta_{\min} - 1} \left( \frac{r_{\beta_{\min}}(T)}{1 - r_{\beta_{\min}}(T)} \right)^{\beta/\beta_{\min}}.$$

Since $r_{\beta_{\min}}(T) \le \varepsilon$ and $x \mapsto x/(1 - x)$ is increasing on $(0, 1)$, we get

$$\frac{r_{\beta_{\min}}(T)}{1 - r_{\beta_{\min}}(T)} \le \frac{\varepsilon}{1 - \varepsilon}.$$

Absorbing the fixed prefactor $\frac{4d\beta - 1}{4d\beta_{\min} - 1}$ into the bound (it depends only on $d$ and the $\beta_i$'s), we obtain the clean rate

$$r_\beta(T) \le \varepsilon^{\beta/\beta_{\min}}.$$

Substituting back $r_\beta(T) = 1 - \frac{w_i^{\mathrm{low}}(T)}{4\beta_i}$ and using $4\beta_i \ge w_i^\star$ on the active set gives

$$\frac{w_i^\star - w_i^{\mathrm{low}}(T)}{w_i^\star} \le \varepsilon^{\beta_i/\beta_{\min}},$$

and if $\beta_i > \beta_{\min}$ then $\varepsilon^{\beta_i/\beta_{\min}} < \varepsilon$, proving the strict improvement. Finally, since $w_i^{\mathrm{low}}(T) \le w_i(T)$, the same lower bound transfers to $w_i(T)$. □

# E. Lower Bound for Rotational Invariant Algorithm

A rotationally invariant learning algorithm produces identical predictions under any orthogonal transformation of the input space. Consequently, such an algorithm cannot exploit any specific direction in the data and must treat all directions equally. This forces it to distribute its estimation effort across many irrelevant directions rather than concentrating on the true signal direction.

For logistic risk $\ell(t) = \log(1 + e^{-t})$, $D := \{(\mathbf{x}_i, y_i)\}_{i=1}^n = (\mathbf{X}, \mathbf{y})$, and defining the empirical logistic risk

$$\hat{\mathcal{L}}_{\mathbf{X},\mathbf{y}}(\boldsymbol{w}) := \frac{1}{n} \sum_{i=1}^n \ell\big(y_i \langle \boldsymbol{x}_i, \boldsymbol{w} \rangle\big) = \frac{1}{n} \sum_{i=1}^n \log\big(1 + \exp(-y_i \boldsymbol{x}_i^\top \boldsymbol{w})\big). \tag{88}$$

For any orthogonal matrix $\boldsymbol{U} \in \mathbb{R}^{d \times d}$, we have the identity

$$\hat{\mathcal{L}}_{\boldsymbol{U}\mathbf{X},\mathbf{y}}(\boldsymbol{U}\boldsymbol{w}) = \frac{1}{n} \sum_{i=1}^n \ell\big(y_i \langle \boldsymbol{U}\boldsymbol{x}_i, \boldsymbol{U}\boldsymbol{w} \rangle\big) = \frac{1}{n} \sum_{i=1}^n \ell\big(y_i \langle \boldsymbol{x}_i, \boldsymbol{w} \rangle\big) = \hat{\mathcal{L}}_{\mathbf{X},\mathbf{y}}(\boldsymbol{w}), \tag{89}$$

since $\langle \boldsymbol{U}\boldsymbol{x}_i, \boldsymbol{U}\boldsymbol{w} \rangle = \langle \boldsymbol{x}_i, \boldsymbol{w} \rangle$. Thus, the logistic empirical risk is invariant under the *simultaneous rotation* $(\boldsymbol{x}_i, \boldsymbol{w}) \mapsto (\boldsymbol{U}\boldsymbol{x}_i, \boldsymbol{U}\boldsymbol{w})$, or equivalently,

$$\hat{\mathcal{L}}_{\boldsymbol{U}\mathbf{X},\mathbf{y}}(\boldsymbol{w}) = \hat{\mathcal{L}}_{\mathbf{X},\mathbf{y}}(\boldsymbol{U}\boldsymbol{w}). \tag{90}$$

This means given fixed label $\mathbf{y}$, generated from the true conditional distribution $P(y = 1|\mathbf{x})$, training with rotated input samples, leads to a rotated estimator.

$$\hat{\mathbf{w}}(\mathbf{X}\boldsymbol{U}, \mathbf{y}) = \boldsymbol{U}\hat{\mathbf{w}}(\mathbf{X}, \mathbf{y}) \tag{91}$$

Equivalently, the induced predictions are invariant under rotations of the input: for every test point $\boldsymbol{x}_{\text{te}} \in \mathbb{R}^d$,

$$P(\hat{y}(\boldsymbol{x}_{\text{te}} \mid \mathbf{X}, \mathbf{y}) = 1) = P(\hat{y}(\boldsymbol{U}\boldsymbol{x}_{\text{te}} \mid \boldsymbol{U}\mathbf{X}, \mathbf{y}) = 1). \tag{92}$$

Here $\hat{y}(\boldsymbol{x}_{\text{te}} \mid \mathbf{X}, \mathbf{y})$ denotes the label prediction of $\mathbf{x}_{\text{te}}$ when the estimator was trained using $D = (\mathbf{X}, \mathbf{y})$. Thus (92) shows that rotation of the input leads to identical label prediction.

**Proposition E.1.** *Single-layer gradient flow for empirical logistic risk, initialized at* $\mathbf{w}(0) = \mathbf{0}$, *is rotation invariant: for any orthogonal matrix* $U$, *rotating all inputs by* $U$ *rotates the entire gradient flow trajectory by the same transformation.*

*Proof.* Let Dataset–A be $\{(\mathbf{x}_i, y_i)\}_{i=1}^n$ and Dataset–B be $\{(U\mathbf{x}_i, y_i)\}_{i=1}^n$, where $U \in \mathbb{R}^{d \times d}$ is an orthogonal matrix. Denote the corresponding empirical risks by $\hat{\mathcal{L}}_A$ and $\hat{\mathcal{L}}_B$. Since the logistic loss depends on the data only through inner products, we have

$$\hat{\mathcal{L}}_B(\mathbf{w}) = \hat{\mathcal{L}}_A(U^\top \mathbf{w}), \qquad \nabla \hat{\mathcal{L}}_B(\mathbf{w}) = U \nabla \hat{\mathcal{L}}_A(U^\top \mathbf{w}).$$

Let $\mathbf{w}_A(t)$ be the gradient flow trajectory on Dataset–A,

$$\dot{\mathbf{w}}_A(t) = -\nabla \hat{\mathcal{L}}_A(\mathbf{w}_A(t)), \qquad \mathbf{w}_A(0) = \mathbf{0},$$

and define $\mathbf{w}_B(t) := U\mathbf{w}_A(t)$. Then $\mathbf{w}_B(0) = \mathbf{0}$ and

$$\dot{\mathbf{w}}_B(t) = -U \nabla \hat{\mathcal{L}}_A(\mathbf{w}_A(t)) = -\nabla \hat{\mathcal{L}}_B(\mathbf{w}_B(t)).$$

Thus $\mathbf{w}_B(t)$ coincides with the gradient flow trajectory obtained by training on Dataset–B, and satisfies $\mathbf{w}_B(t) = U\mathbf{w}_A(t)$ for all $t$. Hence, single-layer gradient flow is rotation invariant. $\square$

**Lemma E.2.** *The spindly predictor dynamics*

$$\dot{\mathbf{w}}(t) = -|\mathbf{w}(t)| \odot \nabla \hat{\mathcal{L}}_A(\mathbf{w}(t)), \qquad \mathbf{w}(0) = \alpha \mathbf{1}, \tag{93}$$

*are not rotation-invariant.*

*Proof.* Assume for contradiction that the dynamics are rotation-invariant which means the parameters from the two rotated datasets: Dataset-A and Dataset-B be related by an orthogonal matrix $\mathbf{U}$, and suppose

$$\mathbf{w}_B(t) = \mathbf{U}\mathbf{w}_A(t) \quad \text{for all } t \geq 0. \tag{94}$$

Differentiating gives

$$\dot{\mathbf{w}}_B(t) = \mathbf{U}\dot{\mathbf{w}}_A(t). \tag{95}$$

Using the spindly dynamics on Dataset-A,

$$\dot{\mathbf{w}}_A(t) = -|\mathbf{w}_A(t)| \odot \nabla\widehat{\mathcal{L}}_A(\mathbf{w}_A(t)), \tag{96}$$

we obtain

$$\dot{\mathbf{w}}_B(t) = -\mathbf{U}\big(|\mathbf{w}_A(t)| \odot \nabla\widehat{\mathcal{L}}_A(\mathbf{w}_A(t))\big). \tag{1}$$

Since the loss depends only on inner products,

$$\nabla\widehat{\mathcal{L}}_B(\mathbf{w}) = \mathbf{U}\nabla\widehat{\mathcal{L}}_A(\mathbf{U}^\top\mathbf{w}). \tag{97}$$

Therefore the spindly dynamics on Dataset-B require

$$\dot{\mathbf{w}}_B(t) = -|\mathbf{w}_B(t)| \odot \nabla\widehat{\mathcal{L}}_B(\mathbf{w}_B(t)) = -|\mathbf{U}\mathbf{w}_A(t)| \odot \mathbf{U}\nabla\widehat{\mathcal{L}}_A(\mathbf{w}_A(t)). \tag{2}$$

For (1) and (2) to coincide for all $t$, we must have

$$\mathbf{U}\big(|\mathbf{w}| \odot \mathbf{g}\big) = |\mathbf{U}\mathbf{w}| \odot (\mathbf{U}\mathbf{g}) \quad \text{for all } \mathbf{w}, \mathbf{g}. \tag{98}$$

Let $\mathbf{D}(\mathbf{w}) = \mathrm{diag}(|\mathbf{w}|)$. The above condition is equivalent to

$$\mathbf{U}\mathbf{D}(\mathbf{w}) = \mathbf{D}(\mathbf{U}\mathbf{w})\,\mathbf{U}. \tag{99}$$

Multiplying on the right by $\mathbf{U}^\top$ yields

$$\mathbf{D}(\mathbf{U}\mathbf{w}) = \mathbf{U}\mathbf{D}(\mathbf{w})\mathbf{U}^\top. \tag{3}$$

Now choose $\mathbf{w}$ whose coordinates have distinct magnitudes, so $\mathbf{D}(\mathbf{w})$ has distinct diagonal entries. Then $\mathbf{U}\mathbf{D}(\mathbf{w})\mathbf{U}^\top$ is diagonal if and only if $\mathbf{U}$ is a signed permutation matrix. For a generic orthogonal $\mathbf{U}$, the right-hand side of (3) has off-diagonal entries, while $\mathbf{D}(\mathbf{U}\mathbf{w})$ is diagonal. This contradicts (3).

Hence the assumed rotation-invariance cannot hold. $\qquad\qquad\square$

We let the conditional label distribution induced by the data-generating process as $q(\mathbf{y}|\mathbf{X})$ given $D = (\mathbf{X}, \mathbf{y})$. And define the rotated observation model as $q_\mathbf{U}(\mathbf{y}|\mathbf{X}) = q(\mathbf{y}|\mathbf{X}\mathbf{U})$.

Let $\tilde{\boldsymbol{X}} = [\boldsymbol{X}; \boldsymbol{x}_{\text{te}}]$ denote the augmented design matrix and $\tilde{\boldsymbol{y}} = [\boldsymbol{y}; y_{\text{te}}]$ the corresponding labels. For an orthogonal matrix $\boldsymbol{U} \in \mathbb{R}^{d \times d}$, define the rotated observation model

$$q_{\boldsymbol{U}}(\tilde{\boldsymbol{y}} \mid \tilde{\boldsymbol{X}}) := q(\tilde{\boldsymbol{y}} \mid \tilde{\boldsymbol{X}}\boldsymbol{U}^\top).$$

The symmetrized observation model is obtained by averaging over all rotations:

$$\bar{q}(\tilde{\boldsymbol{y}} \mid \tilde{\boldsymbol{X}}) := \int q_{\boldsymbol{U}}(\tilde{\boldsymbol{y}} \mid \tilde{\boldsymbol{X}})\,\mathrm{d}\rho_H(\boldsymbol{U}),$$

where $\rho_H$ denotes the Haar measure on the orthogonal group.

Given $(\tilde{\boldsymbol{X}}, \boldsymbol{y})$, the posterior distribution on $\boldsymbol{U}$ under the symmetrized observation model is

$$p(\boldsymbol{U} \mid \tilde{\boldsymbol{X}}, \boldsymbol{y}) \; \propto \; q_{\boldsymbol{U}}(\boldsymbol{y} \mid \tilde{\boldsymbol{X}}) \, \mathrm{d}\rho_H(\boldsymbol{U}). \tag{100}$$

The Bayes-optimal real-valued score is defined by integrating the conditional expected logistic loss over the posterior on $\boldsymbol{U}$:

$$s^{\star}(\boldsymbol{x}_{\mathrm{te}} \mid \boldsymbol{X}, \boldsymbol{y}) \; \in \; \arg\min_{s \in \mathbb{R}} \int \mathbb{E}_{y_{\mathrm{te}} \sim q_{\boldsymbol{U}}(\cdot \mid \tilde{\boldsymbol{X}}, \boldsymbol{y})}[\ell(y_{\mathrm{te}}\, s)] \, p(\boldsymbol{U} \mid \tilde{\boldsymbol{X}}, \boldsymbol{y}) \, \mathrm{d}\boldsymbol{U}. \tag{101}$$

The optimal expected loss under the symmetrized observation model is

$$L_B(\bar{q}) = \mathbb{E}_{\tilde{\boldsymbol{X}} \sim \mathcal{N}(0,\mathbf{I}),\, \boldsymbol{y} \sim \bar{q}(\cdot \mid \tilde{\boldsymbol{X}})} \Big[ \ell\big( y_{\mathrm{te}}\, s^{\star}(\boldsymbol{x}_{\mathrm{te}} \mid \boldsymbol{X}, \boldsymbol{y})\big)\Big]. \tag{102}$$

**Theorem E.3.** *Let $q(\tilde{\boldsymbol{y}} \mid \tilde{\boldsymbol{X}})$ be an observation model and $\hat{s}(\cdot \mid \boldsymbol{X}, \boldsymbol{y})$ be a rotation-invariant learning algorithm trained on* $(\boldsymbol{X}, \boldsymbol{y})$. *Define the expected loss*

$$\mathcal{L}_{\hat{s}}(q) \; := \; \mathbb{E}_{\tilde{\boldsymbol{X}} \sim \mathcal{N}(0,\mathbf{I}),\, \tilde{\boldsymbol{y}} \sim q(\cdot \mid \tilde{\boldsymbol{X}})} \Big[ \ell\big( y_{\mathrm{te}}\, \hat{s}(\boldsymbol{x}_{\mathrm{te}} \mid \boldsymbol{X}, \boldsymbol{y})\big)\Big]. \tag{103}$$

*Then this loss is lower bounded by the Bayes risk of the symmetrized observation model:*

$$\mathcal{L}_{\hat{s}}(q) \; \geq \; \mathcal{L}_B(\bar{q}). \tag{104}$$

See (Warmuth et al., 2025) for proof.

**Lemma E.4.** *Under assumptions stated, the Bayes risk gap* (14) *satifies*

$$\mathcal{L}_B(\bar{q}) - \mathcal{L}(\boldsymbol{w}^{\star}) = \mathbb{E}_D \, \mathbb{E}_{\boldsymbol{w}(D) \sim \Pi(\cdot \mid D)} \Big[ \mathcal{L}(\boldsymbol{w}(D)) - \mathcal{L}(\boldsymbol{w}^{\star})\Big], \tag{105}$$

*where $\Pi(\boldsymbol{w} \mid D)$ is the posterior distribution induced by a uniform prior on the unit sphere $\mathbb{S}^{d-1}$.*

*Proof.* Let the excess risk of a rotational invariant algorithm

$$\mathbb{E}_{\tilde{\boldsymbol{X}} \sim \mathcal{N}(0,\mathbf{I}),\, \tilde{\boldsymbol{y}} \sim q(\cdot \mid \tilde{\boldsymbol{X}})} \Big[ \ell\big( y_{\mathrm{te}}\, \hat{s}(\boldsymbol{x}_{\mathrm{te}} \mid \boldsymbol{X}, \boldsymbol{y})\big)\Big] - \mathcal{L}(\mathbf{w}^{\star}) = \mathcal{L}_{\hat{s}}(q) - \mathcal{L}(\mathbf{w}^{\star}) \geq \mathcal{L}_B(\bar{q}) - \mathcal{L}(\mathbf{w}^{\star}) \tag{106}$$

We can further simplify $\mathcal{L}_B(\bar{q}) - \mathcal{L}(\boldsymbol{w}^{\star})$ as

$$\mathcal{L}_B(\bar{q}) - \mathcal{L}(\boldsymbol{w}^{\star}) = \mathbb{E}_{\tilde{\boldsymbol{X}} \sim \mathcal{N}(0,\mathbf{I})} \, \mathbb{E}_{\boldsymbol{U} \sim \rho_H} \, \mathbb{E}_{\tilde{\boldsymbol{y}} \sim q(\cdot \mid \tilde{\boldsymbol{X}}\boldsymbol{U}^{\top})} \Big[ \ell\big( \hat{y}^{\star}(\boldsymbol{x}_{\mathrm{te}} \mid \boldsymbol{X}, \boldsymbol{y}), y_{\mathrm{te}}\big)\Big] - L(\boldsymbol{w}^{\star}) \tag{107}$$

$$= \mathbb{E}_{\tilde{\boldsymbol{X}} \sim \mathcal{N}(0,\mathbf{I})} \, \mathbb{E}_{\boldsymbol{w} \sim \mathrm{Unif}(\mathbb{S}^{d-1})} \, \mathbb{E}_{\tilde{\boldsymbol{y}} \sim q(\cdot \mid \tilde{\boldsymbol{X}}, \boldsymbol{w})} \Big[ \mathcal{L}(\boldsymbol{w}) - \mathcal{L}(\boldsymbol{w}^{\star})\Big]. \tag{108}$$

$$= \mathbb{E}_{\tilde{\boldsymbol{X}} \sim \mathcal{N}(0,\mathbf{I})} \, \mathbb{E}_{\tilde{\boldsymbol{y}} \sim \bar{q}(\cdot \mid \tilde{\boldsymbol{X}})} \, \mathbb{E}_{\boldsymbol{w} \sim p(\boldsymbol{w} \mid \tilde{\boldsymbol{X}}, \tilde{\boldsymbol{y}})} \Big[ \mathcal{L}(\boldsymbol{w}) - \mathcal{L}(\boldsymbol{w}^{\star})\Big] \tag{109}$$

$$= \mathbb{E}_D \, \mathbb{E}_{\mathbf{w}(D) \sim \Pi(\cdot \mid D)} \Big[ \mathcal{L}(\mathbf{w}(D)) - \mathcal{L}(\boldsymbol{w}^{\star})\Big]. \tag{110}$$

where $\Pi(\mathbf{w} \mid D)$ is termed as spherical posterior since the prior is on the unit sphere.

In (108), we exploit the rotational symmetry of the observation model. Since $\mathbf{U} \in O(d)$ be Haar-distributed and the rotated target vector $\mathbf{w} := \mathbf{U}^{\top} \mathbf{w}^{\star}$. We have

$$\big\| \mathbf{U}^{\top} \mathbf{w}^{\star} \big\|_2^2 = \big\| \mathbf{w}^{\star} \big\|_2^2. \tag{111}$$

In particular, if $\|\mathbf{w}^{\star}\|_2 = 1$, then $\mathbf{w} \in \mathbb{S}^{d-1}$. Moreover, by Haar invariance of $\mathbf{U}$, the random vector $\mathbf{w} = \mathbf{U}^{\top}\mathbf{w}^{\star}$ is uniformly distributed on the unit sphere $\mathbb{S}^{d-1}$. Conditional on $\mathbf{w}$, the rotated observation model $q(\cdot \mid \tilde{\mathbf{X}}\mathbf{U}^{\top})$ coincides with the conditional model $q(\cdot \mid \tilde{\mathbf{X}}, \mathbf{w})$. Therefore, averaging over random rotations $\mathbf{U} \sim \rho_H$ is equivalent to averaging over $\mathbf{w} \sim \mathrm{Unif}(\mathbb{S}^{d-1})$, which yields (49). In the last equation, we write the posterior over $\mathbf{w}$ as :

$$\Pi(\mathbf{w} \mid D) \; \propto \; p(\boldsymbol{w}) \, q(\tilde{\boldsymbol{y}} \mid \tilde{\boldsymbol{X}}, \mathbf{w}) \; \propto \; \exp\big( - n\hat{\mathcal{L}}(\boldsymbol{w})\big) \, \mathbf{1}_{\{\|\boldsymbol{w}\|_2 = 1\}}, \tag{112}$$

$\square$

## F. Excess risk calculation on spherical posterior $\Pi(\mathbf{w} \mid D)$

Let $D = \{(\mathbf{x}_i, y_i)\}_{i=1}^n$ be i.i.d. samples with $\mathbf{x}_i \sim \mathcal{N}(\mathbf{0}, \mathbf{I}_d)$ and

$$P(y_i = 1 \mid \mathbf{x}_i) = \sigma(\langle \mathbf{x}_i, \mathbf{w}^\star \rangle).$$

where the ground-truth parameter satisfies $\mathbf{w}^\star \in \mathbb{S}^{d-1} := \{\mathbf{w} \in \mathbb{R}^d : \|\mathbf{w}\|_2 = 1\}$ and the logistic sigmoid function is defined by

$$\sigma(t) := \frac{1}{1 + e^{-t}}. \tag{113}$$

We define the population logistic risk as

$$\mathcal{L}(\mathbf{w}) := \mathbb{E}_{(\mathbf{x}, y)} \Big[ \log\big(1 + e^{-y \langle \mathbf{x}, \mathbf{w} \rangle}\big) \Big], \tag{114}$$

and the empirical logistic risk as

$$\widehat{\mathcal{L}}_n(\mathbf{w}) := \frac{1}{n} \sum_{i=1}^n \log\big(1 + e^{-y_i \langle \mathbf{x}_i, \mathbf{w} \rangle}\big). \tag{115}$$

We define the *spherical posterior* on the unit sphere $\mathbb{S}^{d-1}$ by

$$\Pi(d\mathbf{w} \mid D) \propto \exp\big(-n\,\widehat{\mathcal{L}}_n(\mathbf{w})\big)\, d\mu(\mathbf{w}), \tag{116}$$

where $d\mu$ denotes the uniform surface measure on $\mathbb{S}^{d-1}$.

Our object of interest is the posterior expected excess population risk

$$\mathcal{R}_\Pi(D) := \mathbb{E}_{\mathbf{w} \sim \Pi(\cdot|D)} \Big[ \mathcal{L}(\mathbf{w}) - \mathcal{L}(\mathbf{w}^\star) \Big]. \tag{117}$$

**Theorem F.1.** *Let $\mathcal{D} = \{(\mathbf{x}_i, y_i)\}_{i=1}^n$ be drawn i.i.d. from data model* (2) *and $n \gtrsim d + \log(\frac{1}{\delta})$. Then, with probability at least $1 - \delta$ over the draw of $\mathcal{D}$, any rotation-invariant learning algorithm outputting $\mathbf{w}(\mathcal{D})$ has an excess risk lower bound over the spherical posterior,*

$$\mathbb{E}_{\boldsymbol{w}(\mathcal{D}) \sim \Pi(\cdot|\mathcal{D})} \left[ \mathcal{L}(\mathbf{w}(\mathcal{D})) - \mathcal{L}(\mathbf{w}^\star) \right] \geq \frac{c(d-1)}{n} \tag{118}$$

*for an absolute constant $c$.*

*Proof.* Our proof relies on a technique of analyzing the excess risk and the posterior distribution on the sphere. We list the proof sketch as follows:

1. To characterize the posterior distribution $\Pi(\mathbf{w}|D)$ on the unit sphere $\mathbb{S}^{d-1}$, we use the chart variable map $\mathbf{w}(\mathbf{z}) = \frac{\mathbf{w}^\star + \mathbf{z}}{\|\mathbf{w}^\star + \mathbf{z}\|_2}$ and work on the domain of $\mathbf{z} \in T$, which is the tangent space to $\mathbf{w}^\star$.

2. We use Lemma F.2, F.3 and F.4 to characterize the population Hessian (at $\mathbf{w}^\star$) on the tangent domain $\mathbf{z}$ (given as $\mathbf{H}_T(\mathbf{w}^\star)$) and show it is isotropic.

3. We then use Lemma F.5 and Lemma F.6 to prove a quadratic lower-bound on the population excess risk on the restricted tangent domain $\mathbf{z}$, $\|\mathbf{z}\|_2 \leq r$ utilizing modified self-concordance hypothesis (Proposition-1 in (Bach, 2010)).

$$\mathcal{L}(\mathbf{w}(\mathbf{z})) - \mathcal{L}(\mathbf{w}^\star) \gtrsim \mathbf{z}^T \mathbf{H}_T(\mathbf{w}^\star) \mathbf{z} \tag{119}$$

4. We use Lemma F.10 to show that the Gibbs posterior $\Pi(\mathbf{z}|\mathcal{D})$ induced by the empirical risk on the sphere is locally log-concave. We also show using Lemma F.11 that $\Pi(\mathbf{z}|\mathcal{D})$ is globally smooth in $\mathbf{z} \in T$. We use Lemma F.9 and F.8 to characterize the posterior distribution on the space of $\mathbf{z}$. So jointly using Lemma F.10 and F.11, we have control on the Hessian of the negative log-likelihood of the Gibb's distribution $V(\mathbf{z}) = -\log \Pi(\mathbf{z}|D)$ induced by the empirical risk.

$$\frac{1}{m} \mathbf{I}_T \succeq \nabla^2 V(\mathbf{z}) \succeq \frac{1}{M} \mathbf{I}_T. \tag{120}$$

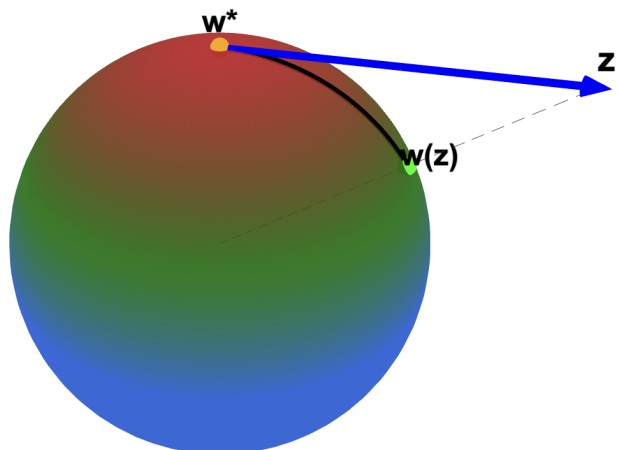

*Figure 7.* Tangent map on the spherical posterior. $\mathbf{w}(\mathbf{z}) = \frac{\mathbf{w}^\star + \mathbf{z}}{\|\mathbf{w}^\star + \mathbf{z}\|_2}$

5. Using Lemma F.14, we use the lower-bound

$$\mathbb{E}_{\Pi(\mathbf{z})}[\mathcal{L}(\mathbf{w}(\mathbf{z})) - \mathcal{L}(\mathbf{w}^\star)] \gtrsim \underbrace{\Pi(\|\mathbf{z}\|_2 \le r \mid D)}_{\text{Factor-1}} \underbrace{\mathbb{E}_{\Pi(\cdot \mid D, \|\mathbf{z}\|_2 \le r)}[\mathbf{z}^\top \mathbf{H}\, \mathbf{z}]}_{\text{Factor-2}}.$$

and control the terms $\Pi(\|\mathbf{z}\|_2 \le r \mid D)$ and $\mathbb{E}_{\Pi(\cdot \mid D, \|\mathbf{z}\|_2 \le r)}[\mathbf{z}^\top \mathbf{H}\, \mathbf{z}]$ respectively. In particular, Lemma F.10 shows us

$$\text{Factor-1:} \quad \Pi(\|\mathbf{z}\|_2 \le r \mid D) \ge 1 - \exp\left(-c_1\left(\tfrac{1}{3}n(1-\varepsilon)\kappa + \tfrac{12}{25}d\right)r^2\right). \tag{121}$$

and Lemma F.12 gives:

$$\text{Factor-2:} \quad \mathbb{E}_{\Pi(\cdot \mid D, \|\mathbf{z}\|_2 \le r)}[\mathbf{z}^\top \mathbf{H}_T^\star\, \mathbf{z}] \ge \frac{c}{n + c_1 d} \operatorname{Tr}(\mathbf{H}_T^\star) \tag{122}$$

6. Putting the lower bounds for these two factors, we derive the final theorem in F.15 and prove the lower bound holds for $n \gtrsim d + \log(\frac{n}{\delta})$ with probability $1 - \delta$ over the draw of dataset $D$.

$\square$

### F.1. Required Lemmas

**Lemma F.2.** *Let $\mathbf{w}^\star \in \mathbb{S}^{d-1}$ and $T$ be the tangent space at $\mathbf{w}^\star$ defined as $T := \{\mathbf{z} \in \mathbb{R}^d : \langle \mathbf{z}, \mathbf{w}^\star \rangle = 0\}$. Then for the orthogonal projector on $T$ (denoted by $\mathbf{P}_T$), we have:*

1. $\mathbf{P}_T = \mathbf{I}_d - \mathbf{w}^\star(\mathbf{w}^\star)^\top$.

2. *Range($\mathbf{P}_T$)=T.*

3. *Ker($\mathbf{P}_T$)=span($\mathbf{w}^\star$)*

*Proof.* Since $\|\mathbf{w}^\star\|_2 = 1$, define $\mathbf{P}_T := \mathbf{I}_d - \mathbf{w}^\star(\mathbf{w}^\star)^\top$, (1) For any $\mathbf{v} \in \mathbb{R}^d$,

$$\mathbf{P}_T\mathbf{v} = \mathbf{v} - \langle \mathbf{v}, \mathbf{w}^\star \rangle \mathbf{w}^\star,$$

which is the orthogonal projection of $\mathbf{v}$ onto the hyperplane orthogonal to $\mathbf{w}^\star$. Hence $\mathbf{P}_T$ is the orthogonal projector onto $T$.

For any $\mathbf{v} \in \mathbb{R}^d$,

$$\langle \mathbf{P}_T\mathbf{v}, \mathbf{w}^\star \rangle = \langle \mathbf{v}, \mathbf{w}^\star \rangle - \langle \mathbf{v}, \mathbf{w}^\star \rangle \|\mathbf{w}^\star\|_2^2 = 0,$$

so $\mathbf{P}_T \mathbf{v} \in T$, implying $\mathrm{Range}(\mathbf{P}_T) \subseteq T$. Conversely, if $\mathbf{z} \in T$, then $\langle \mathbf{z}, \mathbf{w}^\star \rangle = 0$ and thus $\mathbf{P}_T \mathbf{z} = \mathbf{z}$, so $T \subseteq \mathrm{Range}(\mathbf{P}_T)$. Therefore $\mathrm{Range}(\mathbf{P}_T) = T$.

(3) Finally, $\mathbf{P}_T \mathbf{v} = \mathbf{0}$ if and only if

$$\mathbf{v} = \langle \mathbf{v}, \mathbf{w}^\star \rangle \mathbf{w}^\star,$$

which holds if and only if $\mathbf{v} \in \mathrm{span}(\mathbf{w}^\star)$. Hence $\mathrm{Ker}(\mathbf{P}_T) = \mathrm{span}(\mathbf{w}^\star)$. $\qquad \square$

**Lemma F.3.** *The Hessian of the objective at the global minimum $\mathbf{z} = \mathbf{0}$, restricted to $T$, is given by*

$$\mathbf{H}_T^\star := \mathbf{P}_T \mathbf{H}(\mathbf{w}^\star) \mathbf{P}_T,$$

*where $\mathbf{H}(\mathbf{w}^\star)$ denotes the Hessian at the population minimizer $\mathbf{w}^\star$.*

*Proof.* The Hessian of the objective restricted to $T$ is the bilinear form on $T \times T$ given by

$$(\mathbf{u}, \mathbf{v}) \mapsto \mathbf{u}^\top \mathbf{H}(\mathbf{w}^\star) \mathbf{v}, \qquad \mathbf{u}, \mathbf{v} \in T.$$

Fix any $\mathbf{u}, \mathbf{v} \in T$. Since $\mathbf{P}_T$ is the orthogonal projector onto $T$, we have $\mathbf{P}_T \mathbf{u} = \mathbf{u}$ and $\mathbf{P}_T \mathbf{v} = \mathbf{v}$. Therefore,

$$\mathbf{u}^\top \mathbf{H}(\mathbf{w}^\star) \mathbf{v} = (\mathbf{P}_T \mathbf{u})^\top \mathbf{H}(\mathbf{w}^\star)(\mathbf{P}_T \mathbf{v}) = \mathbf{u}^\top \mathbf{P}_T^\top \mathbf{H}(\mathbf{w}^\star) \mathbf{P}_T \mathbf{v}.$$

Using $\mathbf{P}_T^\top = \mathbf{P}_T$ (orthogonal projector), we get

$$\mathbf{u}^\top \mathbf{H}(\mathbf{w}^\star) \mathbf{v} = \mathbf{u}^\top \mathbf{P}_T \mathbf{H}(\mathbf{w}^\star) \mathbf{P}_T \mathbf{v}, \qquad \forall \mathbf{u}, \mathbf{v} \in T.$$

Hence the restriction of $\mathbf{H}(\mathbf{w}^\star)$ to $T$ is represented by the matrix

$$\mathbf{H}_T^\star := \mathbf{P}_T \mathbf{H}(\mathbf{w}^\star) \mathbf{P}_T,$$

$\qquad \square$

**Lemma F.4.** *There exists a constant $\kappa \in (0, 1/4)$ such that*

$$\mathbf{H}_T^\star = \kappa \, \mathbf{P}_T. \quad \kappa = \mathbb{E}\big[\sigma(a)\big(1 - \sigma(a)\big)\big], \qquad a \sim \mathcal{N}(0, 1). \tag{123}$$

*Proof.* Decomposing in orthogonal subspaces, $\mathbf{x} = a\mathbf{w}^\star + \mathbf{u}$, such that $\mathbf{u} \in T$ where $a = \langle \mathbf{x}, \mathbf{w}^\star \rangle \sim \mathcal{N}(0, 1)$, $\mathbf{u} \sim \mathcal{N}(\mathbf{0}, \mathbf{P}_T)$. Then $\mathbf{P}_T \mathbf{x} = \mathbf{u}$, and

$$\mathbf{H}_T^\star = \mathbf{P}_T \, \mathbb{E}\big[\sigma(a)\big(1 - \sigma(a)\big) \mathbf{x}\mathbf{x}^\top\big] \mathbf{P}_T \tag{124}$$

$$= \mathbb{E}\big[\sigma(a)\big(1 - \sigma(a)\big) \mathbf{u}\mathbf{u}^\top\big] \tag{125}$$

$$= \mathbb{E}\big[\sigma(a)\big(1 - \sigma(a)\big)\big] \, \mathbb{E}\big[\mathbf{u}\mathbf{u}^\top\big] \tag{126}$$

$$= \kappa \, \mathbf{P}_T, \tag{127}$$

since $\mathbb{E}[\mathbf{u}\mathbf{u}^\top] = \mathbf{P}_T$. Finally, $0 < \kappa < 1/4$ because $0 < \sigma(t)(1 - \sigma(t)) \leq 1/4$ for all $t \in \mathbb{R}$. Note that although the population Hessian $\mathbf{H}(\mathbf{w}^\star)$ is anisotropic due to its change in the direction of $\mathbf{w}^\star$, its restriction to Tangent space $T$ is isotropic since it completely eliminates the direction of $\mathbf{w}^\star$. $\qquad \square$

**Lemma F.5.** *Let $\mathbf{w}^\star \in \mathbb{S}^{d-1}$ and $T := \{\mathbf{z} \in \mathbb{R}^d : \langle \mathbf{z}, \mathbf{w}^\star \rangle = 0\}$. For $\mathbf{z} \in T$ with $\|\mathbf{z}\|_2 \leq 1/2$, define*

$$\mathbf{w}(\mathbf{z}) := \frac{\mathbf{w}^\star + \mathbf{z}}{\|\mathbf{w}^\star + \mathbf{z}\|_2} \in \mathbb{S}^{d-1}, \qquad \mathbf{v}(\mathbf{z}) := \mathbf{w}(\mathbf{z}) - \mathbf{w}^\star.$$

*Then*

$$\|\mathbf{v}(\mathbf{z})\|_2 \leq \|\mathbf{z}\|_2, \tag{128}$$

$$\|P_T \mathbf{v}(\mathbf{z})\|_2 \geq \frac{3}{4}\|\mathbf{z}\|_2, \tag{129}$$

$$\big|\langle \mathbf{v}(\mathbf{z}), \mathbf{w}^\star \rangle\big| \leq \frac{1}{2}\|\mathbf{z}\|_2^2. \tag{130}$$

*Proof.* Since $\mathbf{z} \perp \mathbf{w}^\star$, we have $\|\mathbf{w}^\star + \mathbf{z}\|_2 = \sqrt{1 + \|\mathbf{z}\|_2^2}$ and thus $\mathbf{w}(\mathbf{z}) = (\mathbf{w}^\star + \mathbf{z})/\sqrt{1 + \|\mathbf{z}\|_2^2}$. Then

$$\mathbf{v}(\mathbf{z}) = \frac{\mathbf{z}}{\sqrt{1 + \|\mathbf{z}\|_2^2}} + \left( \frac{1}{\sqrt{1 + \|\mathbf{z}\|_2^2}} - 1 \right) \mathbf{w}^\star.$$

This gives $P_T \mathbf{v}(\mathbf{z}) = \mathbf{z}/\sqrt{1 + \|\mathbf{z}\|_2^2}$ (from Lemma F.2), hence $\|P_T \mathbf{v}(\mathbf{z})\|_2 = \|\mathbf{z}\|_2/\sqrt{1 + \|\mathbf{z}\|_2^2} \geq (3/4)\|\mathbf{z}\|_2$ for $\|\mathbf{z}\|_2 \leq 1/2$.

We further have

$$\|\mathbf{w}^\star + \mathbf{z}\|_2^2 = \|\mathbf{w}^\star\|_2^2 + \|\mathbf{z}\|_2^2 = 1 + \|\mathbf{z}\|_2^2, \qquad \text{hence} \qquad \|\mathbf{w}^\star + \mathbf{z}\|_2 = \sqrt{1 + \|\mathbf{z}\|_2^2}.$$

A direct computation then yields

$$\|\mathbf{w}^\star(\mathbf{z}) - \mathbf{w}^\star\|_2^2 = 2 - \frac{2}{\sqrt{1 + \|\mathbf{z}\|_2^2}}.$$

Finally, for $u := \|\mathbf{z}\|_2^2 \geq 0$, define

$$f(u) := u - 2 + \frac{2}{\sqrt{1 + u}}.$$

Then $f(0) = 0$ and $f'(u) = 1 - (1 + u)^{-3/2} \geq 0$, so $f(u) \geq 0$ for all $u \geq 0$, i.e.,

$$2 - \frac{2}{\sqrt{1 + \|\mathbf{z}\|_2^2}} \leq \|\mathbf{z}\|_2^2.$$

Therefore,

$$\|\mathbf{v}(\mathbf{z})\|_2 = \|\mathbf{w}(\mathbf{z}) - \mathbf{w}^\star\|_2 \leq \|\mathbf{z}\|_2.$$

We also have

$$\langle \mathbf{v}(\mathbf{z}), \mathbf{w}^* \rangle = \frac{1}{\sqrt{1 + \|\mathbf{z}\|_2^2}} - 1 \qquad (\text{since } \langle \mathbf{z}, \mathbf{w}^* \rangle = 0 \text{ and } \|\mathbf{w}^*\|_2 = 1)$$

$$= - \frac{\|\mathbf{z}\|_2^2}{\sqrt{1 + \|\mathbf{z}\|_2^2}\left(1 + \sqrt{1 + \|\mathbf{z}\|_2^2}\right)}.$$

For $\|\mathbf{z}\|_2 \leq \frac{1}{2}$ we have $\sqrt{1 + \|\mathbf{z}\|_2^2} \geq 1$, so the denominator is at least $1 \cdot (1 + 1) = 2$. Therefore

$$|\langle \mathbf{v}(\mathbf{z}), \mathbf{w}^* \rangle| \leq \frac{\|\mathbf{z}\|_2^2}{2}.$$

$\square$

**Lemma F.6.** *Define the chart excess risk $\Delta(\mathbf{z}) := \mathcal{L}(\mathbf{w}(\mathbf{z})) - \mathcal{L}(\mathbf{w}^\star)$. Then for any $\mathbf{z} \in T$ satisfying*

$$\|\mathbf{z}\|_2 \leq r_{\text{chart}} := \min\left\{ \frac{1}{2}, \frac{1}{R} \right\}, \tag{131}$$

*where $R = \frac{\mathbb{E}[\|\mathbf{x}\|_2^3]}{\kappa \, \mathbb{E}[\|\mathbf{x}\|_2^2]}$, $|g'''(t)| \leq R\|\mathbf{v}\|_2 g''(t)$ holds for the line restriction $g(t) = \mathcal{L}(\mathbf{w}^\star + t\mathbf{v})$, we have the quadratic lower bound*

$$\Delta(\mathbf{z}) \geq \begin{cases} \dfrac{1}{3} \dfrac{1}{\left(1 + \frac{1}{R^2}\right)^2} \mathbf{z}^\top H_T^\star \mathbf{z}, & R \geq 2, \\[2ex] \dfrac{16}{75} \mathbf{z}^\top H_T^\star \mathbf{z}, & R \leq 2. \end{cases}$$

*Proof.* Let $\mathbf{v}(\mathbf{z}) = \mathbf{w}(\mathbf{z}) - \mathbf{w}^\star$ and consider the line restriction $f(t) = \mathcal{L}(\mathbf{w}^\star + t\mathbf{v}(\mathbf{z}))$ for $t \in [0, 1]$. By the modified self-concordance hypothesis Proposition-1 in (Bach, 2010), applied at $\mathbf{w}^\star$ yields

$$\mathcal{L}(\mathbf{w}^\star + \mathbf{v}(\mathbf{z})) \geq \mathcal{L}(\mathbf{w}^\star) + \mathbf{v}(\mathbf{z})^\top \nabla \mathcal{L}(\mathbf{w}^\star) + \frac{\mathbf{v}(\mathbf{z})^\top H(\mathbf{w}^\star) \mathbf{v}(\mathbf{z})}{R^2 \|\mathbf{v}(\mathbf{z})\|_2^2} \left( e^{-R\|\mathbf{v}(\mathbf{z})\|_2} + R\|\mathbf{v}(\mathbf{z})\|_2 - 1 \right). \tag{132}$$

holds for

$$R = \frac{\mathbb{E}[\|\mathbf{x}\|_2^3]}{\kappa \, \mathbb{E}[\|\mathbf{x}\|_2^2]}, \qquad \kappa := \mathbb{E}[\sigma(G)(1 - \sigma(G))], \ G \sim \mathcal{N}(0, 1).$$

since defining the line restriction $f(t) := \mathcal{L}(\mathbf{w}^\star + t\mathbf{v})$. for all $t \in \mathbb{R}$, we get $|f'''(t)| \leq R\|\mathbf{v}\|_2 \, f''(t)$.

Since $\mathbf{w}^\star$ minimizes $\mathcal{L}$ on $\mathbb{S}^{d-1}$ and the model is well specified, we have $\nabla \mathcal{L}(\mathbf{w}^\star) = \mathbf{0}$. If $\|\mathbf{z}\|_2 \leq 1/R$, Lemma F.5 gives $\|\mathbf{v}\|_2 \leq \|\mathbf{z}\|_2 \leq 1/R$, hence $u := R\|\mathbf{v}\|_2 \leq 1$ and the scalar bound

$$\frac{e^{-u} + u - 1}{u^2} \geq \frac{1}{3}$$

implies

$$\Delta(\mathbf{z}) = \mathcal{L}(\mathbf{w}^\star + \mathbf{v}(\mathbf{z})) - \mathcal{L}(\mathbf{w}^\star) \geq \frac{1}{3} \mathbf{v}(\mathbf{z})^\top H(\mathbf{w}^\star)\mathbf{v}(\mathbf{z}).$$

Therefore,

$$\Delta(\mathbf{z}) \geq \frac{1}{3} \mathbf{v}^\top H(\mathbf{w}^\star)\mathbf{v} = \frac{1}{3} \frac{1}{(1 + \|\mathbf{z}\|_2^2)} \mathbf{z}^T H_T^* \mathbf{z}$$

Recall that the chart radius satisfies

$$\|\mathbf{z}\|_2 \leq r_{\text{chart}} := \min\left\{\frac{1}{2}, \frac{1}{R}\right\}.$$

Therefore,

$$(1 + \|\mathbf{z}\|_2^2) \leq (1 + r_{\text{chart}}^2), \qquad \frac{1}{(1 + \|\mathbf{z}\|_2^2)} \geq \frac{1}{(1 + r_{\text{chart}}^2)}.$$

Substituting into the previous bound yields

$$\Delta(\mathbf{z}) \geq \frac{1}{3} \frac{1}{(1 + r_{\text{chart}}^2)} \mathbf{z}^\top H_T^\star \mathbf{z}, \qquad r_{\text{chart}} = \min\left\{\frac{1}{2}, \frac{1}{R}\right\}.$$

Equivalently, this gives the following two cases:

$$\Delta(\mathbf{z}) \geq \begin{cases} \dfrac{1}{3} \dfrac{1}{\left(1 + \frac{1}{R^2}\right)} \mathbf{z}^\top H_T^\star \mathbf{z}, & R \geq 2, \\ \dfrac{4}{15} \mathbf{z}^\top H_T^\star \mathbf{z}, & R \leq 2. \end{cases}$$

$\square$

**Lemma F.7.** *There exists an absolute constant $c > 0$ such that for any $\delta \in (0, 1)$, then defining the event,*

$$\mathcal{E}_A := \left\{\max_{1 \leq i \leq n} \|\mathbf{x}_i\|_2 \leq c\left(\sqrt{d} + \sqrt{\log(n/\delta)}\right)\right\}. \tag{133}$$

*where $\mathbf{x}_1, \ldots, \mathbf{x}_n \sim \mathcal{N}(\mathbf{0}, \mathbf{I}_d)$ are i.i.d, we have $\mathbb{P}(\mathcal{E}_A) \geq 1 - \delta$ and define $R_e := c\left(\sqrt{d} + \sqrt{\log(n/\delta)}\right)$.*

*Proof.* Since, $\mathbf{x} \sim \mathcal{N}(\mathbf{0}, \mathbf{I}_d)$. By Theorem 3.1.1 in (Vershynin, 2018), there exists an absolute constant $c > 0$ such that for all $t > 0$,

$$\mathbb{P}\left(\|\mathbf{x}\|_2 \geq c(\sqrt{d} + \sqrt{t})\right) \leq e^{-t}.$$

Applying this bound to each $\mathbf{x}_i$ and using a union bound, we obtain

$$\mathbb{P}\left(\max_{1 \leq i \leq n} \|\mathbf{x}_i\|_2 \geq c(\sqrt{d} + \sqrt{t})\right) \leq n\, e^{-t}.$$

Choosing $t = \log(n/\delta)$ gives

$$\mathbb{P}\left(\max_{1 \leq i \leq n} \|\mathbf{x}_i\|_2 \geq c\left(\sqrt{d} + \sqrt{\log(n/\delta)}\right)\right) \leq \delta.$$

$\square$

**Lemma F.8.** *Let the empirical and population Hessian are respectivelty defined as*

$$\widehat{\mathbf{H}}_T := \frac{1}{n}\sum_{i=1}^{n}\mathbf{P}_T\big(a_i\,\mathbf{x}_i\mathbf{x}_i^\top\big)\mathbf{P}_T, \qquad \mathbf{H}_T^\star := \mathbb{E}\big[\mathbf{P}_T(a\,\mathbf{x}\mathbf{x}^\top)\mathbf{P}_T\big] = \kappa\,\mathbf{P}_T,$$

*where $\mathbf{x}_i \sim \mathcal{N}(\mathbf{0}, \mathbf{I}_d)$ are i.i.d. $a_i = \sigma\big((y_i\langle\mathbf{x}_i, \mathbf{w}^\star\rangle)\big)(1 - \sigma\big((y_i\langle\mathbf{x}_i, \mathbf{w}^\star\rangle)\big)) \in (0, 1/4]$ and $\kappa := \mathbb{E}[a]$. Then define the event $\mathcal{E}_\mathrm{B}$ for an absolute constant $c_1 > 0$,*

$$\mathcal{E}_\mathrm{B} := \left\{(1-\varepsilon)\,\mathbf{H}_T^\star \preceq \widehat{\mathbf{H}}_T \preceq (1+\varepsilon)\,\mathbf{H}_T^\star\right\}, \qquad \text{with} \quad \varepsilon := c_1\sqrt{\frac{d + \log(1/\delta_1)}{n}} \tag{134}$$

*we have $\mathbb{P}(\mathcal{E}_\mathrm{B}) \geq 1 - \delta_1$.*

*Proof.* Write $\mathbf{P}_T\mathbf{x}_i = \boldsymbol{\xi}_i$, where

$$\boldsymbol{\xi}_i \sim \mathcal{N}(\mathbf{0}, \mathbf{I}_{d-1}), \qquad g_i := \langle\mathbf{x}_i, \mathbf{w}^\star\rangle \sim \mathcal{N}(0, 1),$$

and note that $a_i = \sigma(g_i)(1 - \sigma(g_i))$ depends only on $g_i$. Thus

$$\mathbf{P}_T(a_i\mathbf{x}_i\mathbf{x}_i^\top)\mathbf{P}_T \stackrel{d}{=} a_i\,\boldsymbol{\xi}_i\boldsymbol{\xi}_i^\top, \qquad a_i \perp \boldsymbol{\xi}_i, \qquad \mathbb{E}[a_i] = \kappa.$$

Decomposing the Hessian difference into the orthogonal subspaces, we get

$$\widehat{\mathbf{H}}_T - \mathbf{H}_T^\star = \underbrace{\frac{1}{n}\sum_{i=1}^{n} a_i(\boldsymbol{\xi}_i\boldsymbol{\xi}_i^\top - \mathbf{I})}_{=:\mathbf{A}} + \underbrace{\left(\frac{1}{n}\sum_{i=1}^{n}(a_i - \kappa)\right)\mathbf{I}}_{=:\mathbf{B}}.$$

*Control of $\mathbf{B}$.:* Since $a_i \in [0, 1/4]$ are i.i.d., Hoeffding's inequality implies that with probability at least $1 - \delta/2$,

$$\|\mathbf{B}\| = \left|\frac{1}{n}\sum_{i=1}^{n}(a_i - \kappa)\right| \leq \frac{1}{4}\sqrt{\frac{2\log(2/\delta)}{n}}.$$

*Control of $\mathbf{A}$.* Conditionally on $\{a_i\}$, the vectors $\sqrt{a_i}\boldsymbol{\xi}_i$ are independent, mean-zero, subgaussian vectors in $\mathbb{R}^{d-1}$ with covariance $a_i\mathbf{I}$. By the matrix Bernstein inequality for sample covariance matrices of subgaussian vectors (Vershynin, 2018), there exists an absolute constant $C > 0$ such that with probability at least $1 - \delta/2$,

$$\|\mathbf{A}\| \leq C\left(\sqrt{\frac{d + \log(2/\delta)}{n}} + \frac{d + \log(2/\delta)}{n}\right)\max_i a_i \leq C\sqrt{\frac{d + \log(2/\delta)}{n}},$$

where we used $\max_i a_i \leq 1/4$ and $n \gtrsim d$.

Combining the bounds for $\mathbf{A}$ and $\mathbf{B}$ and applying a union bound yields, with probability at least $1 - \delta$,

$$\big\|\widehat{\mathbf{H}}_T - \mathbf{H}_T^\star\big\| \leq c_1\sqrt{\frac{d + \log(1/\delta)}{n}}.$$

$\square$

**Lemma F.9.** *The posterior induced on the tangent coordinates satisfies*

$$\Pi(d\mathbf{z} \mid D) \propto \exp\big(-n\,\widehat{\mathcal{L}}_n(\mathbf{w}(\mathbf{z}))\big)(1 + \|\mathbf{z}\|_2^2)^{-d/2}\,d\mathbf{z}, \qquad \mathbf{z} \in T.$$

*Proof.* On the unit sphere $\mathbb{S}^{d-1}$, the (unnormalized) posterior is

$$\Pi(d\mathbf{w} \mid D) \propto \exp\big(-n\,\widehat{\mathcal{L}}_n(\mathbf{w})\big)\,d\mu(\mathbf{w}),$$

where $d\mu$ denotes surface measure on $\mathbb{S}^{d-1}$. Fix $\mathbf{w}^{\star} \in \mathbb{S}^{d-1}$ and parametrize a neighborhood of $\mathbf{w}^{\star}$ by the normalization chart

$$\mathbf{w}(\mathbf{z}) : T \to \mathbb{S}^{d-1}, \qquad \mathbf{w}(\mathbf{z}) = \frac{\mathbf{w}^{\star} + \mathbf{z}}{\sqrt{1 + \|\mathbf{z}\|_2^2}},$$

where $T = \{\mathbf{z} \in \mathbb{R}^d : \langle \mathbf{z}, \mathbf{w}^{\star} \rangle = 0\}$.

By the change-of-variables formula on manifolds,

$$d\mu(\mathbf{w}) = \sqrt{\det\big(D\psi(\mathbf{z})^{\top} D\psi(\mathbf{z})\big)} \, d\mathbf{z} =: J(\mathbf{z}) \, d\mathbf{z},$$

where $d\mathbf{z}$ is Lebesgue measure on $T$. By Lemma F.16, the Jacobian satisfies

$$J(\mathbf{z}) = \|\mathbf{w}^{\star} + \mathbf{z}\|_2^{-d} = (1 + \|\mathbf{z}\|_2^2)^{-d/2}.$$

Substituting into the posterior we get the required expression. $\qquad\square$

**Lemma F.10.** *Let the induced posterior on $T$ defined as*

$$\Pi(d\mathbf{z} \mid D) \propto \exp\big(-n\,\widehat{\mathcal{L}}_n(\mathbf{w}(\mathbf{z}))\big)\,(1 + \|\mathbf{z}\|_2^2)^{-d/2}\,d\mathbf{z}, \qquad \mathbf{z} \in T.$$

*and the negative log-density of the posterior $\Pi(d\mathbf{z} \mid D)$ defined as*

$$V(\mathbf{z}) := n\,\widehat{\mathcal{L}}_n(\mathbf{w}(\mathbf{z})) + \frac{d}{2}\log(1 + \|\mathbf{z}\|_2^2).$$

*, then under the event $\mathcal{E}_C = \mathcal{E}_A \cap \mathcal{E}_A$ for all radii $0 < r \le r_{\text{slc}} := \min\left\{\frac{1}{2}, \frac{\log 3}{R_e}\right\}$ we have*

$$\nabla^2 V(\mathbf{z}) \succeq \left(\frac{1}{3}n(1-\varepsilon)\kappa + \frac{12}{25}d\right)\mathbf{I}_T, \qquad \forall\,\mathbf{z} \in T : \|\mathbf{z}\|_2 \le r,$$

*where $\mathbf{I}_T$ is the identity on $T \simeq \mathbb{R}^{d-1}$ and $R_e$ defined in Theorem F.7.*

*Proof.* Writing $V(\mathbf{z}) = V_{\text{loss}}(\mathbf{z}) + V_{\text{jac}}(\mathbf{z})$ with

$$V_{\text{loss}}(\mathbf{z}) := n\,\widehat{\mathcal{L}}_n(\mathbf{w}(\mathbf{z})), \qquad V_{\text{jac}}(\mathbf{z}) := \frac{d}{2}\log(1 + \|\mathbf{z}\|_2^2).$$

We separately accumulate the curvature term from the Jacobian and the loss as follows:

*Curvature term from the Jacobian:*

$$\nabla^2 V_{\text{jac}}(\mathbf{z}) = d\Big(\frac{1}{1 + \|\mathbf{z}\|_2^2}\mathbf{I}_T - \frac{2}{(1 + \|\mathbf{z}\|_2^2)^2}\mathbf{z}\mathbf{z}^{\top}\Big).$$

Hence

$$\lambda_{\min}\big(\nabla^2 V_{\text{jac}}(\mathbf{z})\big) = d \cdot \frac{1 - \|\mathbf{z}\|_2^2}{(1 + \|\mathbf{z}\|_2^2)^2}.$$

In particular, for $\|\mathbf{z}\|_2 \le 1/2$,

$$\nabla^2 V_{\text{jac}}(\mathbf{z}) \succeq \frac{12}{25}\,d\,\mathbf{I}_T.$$

*Curvature term from the empirical loss:*

Defining the segment direction $\mathbf{v}(\mathbf{z}) := \mathbf{w}(\mathbf{z}) - \mathbf{w}^{\star}$ and the segment

$$\gamma_{\mathbf{z}}(t) := \mathbf{w}^{\star} + t\,\mathbf{v}(\mathbf{z}), \qquad t \in [0, 1],$$

we consider the directional curvature

$$h_{\mathbf{z},\mathbf{u}}(t) := \mathbf{u}^{\top}\nabla^2\widehat{\mathcal{L}}_n\big(\gamma_{\mathbf{z}}(t)\big)\mathbf{u}, \qquad t \in [0, 1].$$

For logistic loss and on the event $\mathcal{E}_A$, the empirical modified self-concordance bound (Proposition-1 in (Bach, 2010)) gives

$$\left|h'_{\mathbf{z},\mathbf{u}}(t)\right| \leq R_e \left\|\mathbf{v}(\mathbf{z})\right\|_2 h_{\mathbf{z},\mathbf{u}}(t), \qquad \forall\, t \in [0,1].$$

Equivalently, since $h_{\mathbf{z},\mathbf{u}}(t) > 0$,

$$-R_e \|\mathbf{v}(\mathbf{z})\|_2 \;\leq\; \frac{d}{dt}\log h_{\mathbf{z},\mathbf{u}}(t) \;\leq\; R_e \|\mathbf{v}(\mathbf{z})\|_2, \qquad \forall\, t \in [0,1].$$

Integrating from 0 to 1 yields

$$\log h_{\mathbf{z},\mathbf{u}}(1) - \log h_{\mathbf{z},\mathbf{u}}(0) \geq -R_e \|\mathbf{v}(\mathbf{z})\|_2,$$

and hence

$$h_{\mathbf{z},\mathbf{u}}(1) \geq \exp\big(-R_e \|\mathbf{v}(\mathbf{z})\|_2\big)\, h_{\mathbf{z},\mathbf{u}}(0).$$

Recalling $\gamma_{\mathbf{z}}(1) = \mathbf{w}(\mathbf{z})$ and $\gamma_{\mathbf{z}}(0) = \mathbf{w}^\star$, this is

$$\mathbf{u}^\top \nabla^2 \widehat{\mathcal{L}}_n\big(\mathbf{w}(\mathbf{z})\big)\mathbf{u} \geq \exp\big(-R_e \|\mathbf{w}(\mathbf{z}) - \mathbf{w}^\star\|_2\big)\, \mathbf{u}^\top \nabla^2 \widehat{\mathcal{L}}_n(\mathbf{w}^\star)\mathbf{u}.$$

If additionally $\|\mathbf{z}\|_2 \leq \frac{1}{2}$, then $\|\mathbf{w}(\mathbf{z}) - \mathbf{w}^\star\|_2 = \mathbf{v}(\mathbf{z}) \leq \|\mathbf{z}\|_2$, so

$$\mathbf{u}^\top \nabla^2 \widehat{\mathcal{L}}_n\big(\mathbf{w}(\mathbf{z})\big)\mathbf{u} \geq \exp\big(-R_e \|\mathbf{z}\|_2\big)\, \mathbf{u}^\top \nabla^2 \widehat{\mathcal{L}}_n(\mathbf{w}^\star)\mathbf{u}. \tag{135}$$

Now on event $\mathcal{E}_B$, we have

$$\widehat{\mathbf{H}}_T(\mathbf{w}^\star) := \mathbf{P}_T \nabla^2 \widehat{\mathcal{L}}_n(\mathbf{w}^\star)\mathbf{P}_T \;\succeq\; (1-\varepsilon)\,\mathbf{H}_T^\star = (1-\varepsilon)\kappa\,\mathbf{P}_T, \qquad \varepsilon = c\sqrt{\frac{d + \log(1/\delta)}{n}}. \tag{136}$$

Equivalently, in operator form on $T$ on combination of events $\mathcal{E}_C = \mathcal{E}_A \cup \mathcal{E}_B$,

$$\mathbf{P}_T \nabla^2 \widehat{\mathcal{L}}_n\big(\mathbf{w}(\mathbf{z})\big)\mathbf{P}_T \;\succeq\; \exp\big(-R_e \|\mathbf{z}\|_2\big)\,(1-\varepsilon)\kappa\,\mathbf{P}_T, \qquad \forall\, \|\mathbf{z}\|_2 \leq \tfrac{1}{2}. \tag{137}$$

Consequently, for any radius $0 < r \leq \frac{1}{2}$ and all $\|\mathbf{z}\|_2 \leq r$,

$$\mathbf{P}_T \nabla^2 \widehat{\mathcal{L}}_n\big(\mathbf{w}(\mathbf{z})\big)\mathbf{P}_T \;\succeq\; \exp(-R_e r)\,(1-\varepsilon)\kappa\,\mathbf{P}_T. \tag{138}$$

Choosing $r \leq c_0/R_e$ with $c_0 \leq \log 3$ (so that $\exp(-R_e r) \geq 1/3$) yields the uniform lower bound

$$\mathbf{P}_T \nabla^2 \widehat{\mathcal{L}}_n\big(\mathbf{w}(\mathbf{z})\big)\mathbf{P}_T \;\succeq\; \frac{1}{3}(1-\varepsilon)\kappa\,\mathbf{P}_T, \qquad \forall\, \|\mathbf{z}\|_2 \leq r, \qquad r \leq \min\left\{\frac{1}{2}, \frac{\log 3}{R_e}\right\}. \tag{139}$$

Thus, for every $\|\mathbf{z}\|_2 \leq r$, the restriction of $\mathbf{P}_T \nabla^2 \widehat{\mathcal{L}}_n(\mathbf{w}(\mathbf{z}))\mathbf{P}_T$ to the tangent space $T$ has

$$\lambda_{\min}\Big(\mathbf{P}_T \nabla^2 \widehat{\mathcal{L}}_n(\mathbf{w}(\mathbf{z}))\mathbf{P}_T\big|_T\Big) \;\geq\; \frac{1}{3}(1-\varepsilon)\kappa,$$

For all $\mathbf{z} \in T$ with $\|\mathbf{z}\|_2 \leq r$, we have

$$\nabla^2 V(\mathbf{z}) = \nabla^2 V_{\text{loss}}(\mathbf{z}) + \nabla^2 V_{\text{jac}}(\mathbf{z})$$
$$= n\,\mathbf{P}_T \nabla^2 \widehat{\mathcal{L}}_n\big(\mathbf{w}(\mathbf{z})\big)\mathbf{P}_T \;+\; \nabla^2 V_{\text{jac}}(\mathbf{z}). \tag{140}$$

On the one hand, by (139),

$$n\,\mathbf{P}_T \nabla^2 \widehat{\mathcal{L}}_n\big(\mathbf{w}(\mathbf{z})\big)\mathbf{P}_T \;\succeq\; \frac{1}{3}\,n(1-\varepsilon)\kappa\,\mathbf{P}_T. \tag{141}$$

On the other hand, for $\|\mathbf{z}\|_2 \le \frac{1}{2}$ the Jacobian curvature satisfies

$$\nabla^2 V_{\mathrm{jac}}(\mathbf{z}) \succeq \frac{12}{25} d \, \mathbf{I}_T, \tag{142}$$

where $\mathbf{I}_T$ denotes the identity on $T \simeq \mathbb{R}^{d-1}$ (and $\mathbf{P}_T$ acts as $\mathbf{I}_T$ on $T$). Combining (141) and (142) in (140) yields

$$\nabla^2 V(\mathbf{z}) \succeq \left( \frac{1}{3} n(1-\varepsilon)\kappa + \frac{12}{25} d \right) \mathbf{I}_T, \qquad \forall \, \mathbf{z} \in T : \|\mathbf{z}\|_2 \le r, \tag{143}$$

for any radius

$$0 < r \le r_{\mathrm{slc}} := \min \left\{ \tfrac{1}{2}, \frac{c_0}{R_e} \right\}. \tag{144}$$

In particular, the restriction of $\Pi(\cdot \mid D)$ to $\{\|\mathbf{z}\|_2 \le r\}$ is $m$-strongly log-concave with probability $1 - \delta - \delta_1$ which is the probabiltiy of event $\mathcal{E}_A \cap \mathcal{E}_B$ with $m$ being:

$$m := \frac{1}{3} n(1-\varepsilon)\kappa + \frac{12}{25} d. \tag{145}$$

$\square$

**Lemma F.11.** *The induced posterior on the tangent space $T \simeq \mathbb{R}^{d-1}$ being*

$$\Pi(d\mathbf{z} \mid D) \propto \exp\left( - n \, \widehat{\mathcal{L}}_n(\mathbf{w}(\mathbf{z})) \right) (1 + \|\mathbf{z}\|_2^2)^{-d/2} \, d\mathbf{z}.$$

*And defining the negative log-density*

$$V(\mathbf{z}) := n \, \widehat{\mathcal{L}}_n(\mathbf{w}(\mathbf{z})) + \frac{d}{2} \log(1 + \|\mathbf{z}\|_2^2).$$

*Then the event $\mathcal{E}_{\mathrm{D}}$*

$$\mathcal{E}_{\mathrm{D}} := \left\{ \forall \mathbf{z} \in T : \ \nabla^2 V(\mathbf{z}) \preceq \left( \frac{n}{4} \left( 1 + \sqrt{\frac{d}{n}} + \sqrt{\frac{t}{n}} \right)^2 + d \right) \mathbf{I}_T \right\}.$$

*occurs with probability $1 - 2e^{-t}$. In particular, the posterior $\Pi(\cdot \mid D)$ is globally smooth on $T$.*

*Proof.* Recall that

$$V(\mathbf{z}) = n \, \widehat{\mathcal{L}}_n(\mathbf{w}(\mathbf{z})) + \frac{d}{2} \log(1 + \|\mathbf{z}\|_2^2), \qquad \mathbf{z} \in T \simeq \mathbb{R}^{d-1}.$$

Hence

$$\nabla^2 V(\mathbf{z}) = \nabla^2 \left( n \, \widehat{\mathcal{L}}_n(\mathbf{w}(\mathbf{z})) \right) + \nabla^2 \left( \frac{d}{2} \log(1 + \|\mathbf{z}\|_2^2) \right).$$

We upper bound the two Hessians separately (empirical risk and Jacobian) and then add the bounds.

First recall, we had from Lemma, the Hessian of the Jacobian term as:

$$\nabla^2 J(\mathbf{z}) = d \left( \frac{1}{1 + \|\mathbf{z}\|_2^2} \mathbf{I}_T - \frac{2}{(1 + \|\mathbf{z}\|_2^2)^2} \mathbf{z}\mathbf{z}^\top \right).$$

Since $\mathbf{z}\mathbf{z}^\top \succeq \mathbf{0}$, the second term is negative semidefinite, hence

$$\nabla^2 J(\mathbf{z}) \preceq d \cdot \frac{1}{1 + \|\mathbf{z}\|_2^2} \mathbf{I}_T \preceq d \, \mathbf{I}_T, \qquad \forall \mathbf{z} \in T.$$

Letting

$$\widehat{\mathcal{L}}_n(\mathbf{w}) = \frac{1}{n} \sum_{i=1}^n \log \left( 1 + \exp(-y_i \, \mathbf{x}_i^\top \mathbf{w}) \right), \qquad y_i \in \{\pm 1\}.$$

Let $\sigma(u) = \frac{1}{1+e^{-u}}$ denote the logistic sigmoid. For each $i$, define $\ell_i(\mathbf{w}) = \log(1 + \exp(-y_i \mathbf{x}_i^\top \mathbf{w}))$. A direct differentiation gives

$$\nabla^2 \ell_i(\mathbf{w}) = \sigma(y_i \mathbf{x}_i^\top \mathbf{w})\Big(1 - \sigma(y_i \mathbf{x}_i^\top \mathbf{w})\Big) \mathbf{x}_i \mathbf{x}_i^\top.$$

Therefore,

$$\nabla^2 \widehat{\mathcal{L}}_n(\mathbf{w}) = \frac{1}{n} \sum_{i=1}^n \sigma(y_i \mathbf{x}_i^\top \mathbf{w})\Big(1 - \sigma(y_i \mathbf{x}_i^\top \mathbf{w})\Big) \mathbf{x}_i \mathbf{x}_i^\top.$$

Using the sigmoid bound

$$0 \le \sigma(u)\big(1 - \sigma(u)\big) \le \frac{1}{4} \qquad \forall\, u \in \mathbb{R},$$

we obtain the positive semidefinite upper bound

$$\nabla^2 \widehat{\mathcal{L}}_n(\mathbf{w}) \preceq \frac{1}{4n} \sum_{i=1}^n \mathbf{x}_i \mathbf{x}_i^\top = \frac{1}{4}\,\widehat{\Sigma}, \qquad \widehat{\Sigma} := \frac{1}{n} \sum_{i=1}^n \mathbf{x}_i \mathbf{x}_i^\top.$$

Multiplying by $n$ yields

$$\nabla^2\Big(n\,\widehat{\mathcal{L}}_n(\mathbf{w})\Big) = n\,\nabla^2 \widehat{\mathcal{L}}_n(\mathbf{w}) \preceq \frac{n}{4}\,\widehat{\Sigma}.$$

Since $\mathbf{x}_i \sim \mathcal{N}(\mathbf{0}, \mathbf{I}_d)$ i.i.d, then for any $t \ge 0$, with probability at least $1 - 2e^{-t}$,

$$\lambda_{\max}(\widehat{\Sigma}) \le \left(1 + \sqrt{\frac{d}{n}} + \sqrt{\frac{t}{n}}\right)^2.$$

Equivalently, on the same event,

$$\widehat{\Sigma} \preceq \left(1 + \sqrt{\frac{d}{n}} + \sqrt{\frac{t}{n}}\right)^2 \mathbf{I}_d.$$

Combining with the previous step gives, on this event,

$$\nabla^2\Big(n\,\widehat{\mathcal{L}}_n(\mathbf{w})\Big) \preceq \frac{n}{4} \left(1 + \sqrt{\frac{d}{n}} + \sqrt{\frac{t}{n}}\right)^2 \mathbf{I}_d.$$

Now consider the composition $\mathbf{z} \mapsto \mathbf{w}(\mathbf{z}) \mapsto n\widehat{\mathcal{L}}_n(\mathbf{w}(\mathbf{z}))$. For any $\mathbf{z}$, define the Jacobian of the chart

$$\mathbf{J}(\mathbf{z}) := \nabla_{\mathbf{z}} \mathbf{w}(\mathbf{z}) \in \mathbb{R}^{d \times (d-1)}.$$

By the multivariate chain rule, the Hessian of the composition satisfies

$$\nabla_{\mathbf{z}}^2\Big(n\,\widehat{\mathcal{L}}_n(\mathbf{w}(\mathbf{z}))\Big) = \mathbf{J}(\mathbf{z})^\top \left(\nabla_{\mathbf{w}}^2\Big(n\,\widehat{\mathcal{L}}_n(\mathbf{w})\Big)\Big|_{\mathbf{w}=\mathbf{w}(\mathbf{z})}\right) \mathbf{J}(\mathbf{z}) + \sum_{k=1}^d \left(\nabla_{\mathbf{w}}\Big(n\,\widehat{\mathcal{L}}_n(\mathbf{w})\Big)\Big|_{\mathbf{w}=\mathbf{w}(\mathbf{z})}\right)_k \nabla_{\mathbf{z}}^2 w_k(\mathbf{z}).$$

The second term is a symmetric matrix that depends on the second derivatives of the chart. For an upper bound of the form stated in the lemma, we use the fact that the first term is positive semidefinite and upper bounded by pushing forward the operator norm bound: for any vector $\mathbf{u} \in \mathbb{R}^{d-1}$,

$$\mathbf{u}^\top \mathbf{J}(\mathbf{z})^\top \left(\nabla_{\mathbf{w}}^2\Big(n\,\widehat{\mathcal{L}}_n(\mathbf{w})\Big)\Big|_{\mathbf{w}=\mathbf{w}(\mathbf{z})}\right) \mathbf{J}(\mathbf{z})\, \mathbf{u} = (\mathbf{J}(\mathbf{z})\mathbf{u})^\top \left(\nabla_{\mathbf{w}}^2\Big(n\,\widehat{\mathcal{L}}_n(\mathbf{w})\Big)\Big|_{\mathbf{w}=\mathbf{w}(\mathbf{z})}\right) (\mathbf{J}(\mathbf{z})\mathbf{u})$$

$$\le \left\|\nabla_{\mathbf{w}}^2\Big(n\,\widehat{\mathcal{L}}_n(\mathbf{w})\Big)\Big|_{\mathbf{w}=\mathbf{w}(\mathbf{z})}\right\|_{\mathrm{op}} \cdot \|\mathbf{J}(\mathbf{z})\mathbf{u}\|_2^2.$$

For the stereographic tangent chart used in the paper, $\mathbf{J}(\mathbf{z})$ is uniformly bounded on $T$ in operator norm by 1 (equivalently, $\|\mathbf{J}(\mathbf{z})\mathbf{u}\|_2 \le \|\mathbf{u}\|_2$ for all $\mathbf{u}$), hence

$$\mathbf{J}(\mathbf{z})^\top \left(\nabla_{\mathbf{w}}^2\Big(n\,\widehat{\mathcal{L}}_n(\mathbf{w})\Big)\Big|_{\mathbf{w}=\mathbf{w}(\mathbf{z})}\right) \mathbf{J}(\mathbf{z}) \preceq \left\|\nabla_{\mathbf{w}}^2\Big(n\,\widehat{\mathcal{L}}_n(\mathbf{w})\Big)\Big|_{\mathbf{w}=\mathbf{w}(\mathbf{z})}\right\|_{\mathrm{op}} \mathbf{I}_T.$$

Therefore, we get,

$$\nabla_{\mathbf{z}}^2\Big(n\,\widehat{\mathcal{L}}_n(\mathbf{w}(\mathbf{z}))\Big) \preceq \frac{n}{4}\left(1 + \sqrt{\frac{d}{n}} + \sqrt{\frac{t}{n}}\right)^2 \mathbf{I}_T.$$

Using the decomposition $\nabla^2 V = \nabla^2(n\widehat{\mathcal{L}}_n(\mathbf{w}(\mathbf{z}))) + \nabla^2 J(\mathbf{z})$ and the bounds from Step 1 and Step 4, on the same event of probability at least $1 - 2e^{-t}$ we have for all $\mathbf{z} \in T$,

$$\nabla^2 V(\mathbf{z}) \preceq \left(\frac{n}{4}\left(1 + \sqrt{\frac{d}{n}} + \sqrt{\frac{t}{n}}\right)^2 + d\right)\mathbf{I}_T.$$

$\square$

**Lemma F.12.** *On the truncated Gibbs posterior $\Pi_r(\mathbf{z}) \propto \exp\big(-V(\mathbf{z})\big)\,\mathbf{1}_{\{\|\mathbf{z}\|_2 \leq r\}}$ with radius $r := \min\left\{\frac{1}{2}, \frac{1}{R}\right\}$ and event $\mathcal{E}_D$, there exist absolute constants $c > 0$ and $c_1 > 0$ such that*

$$\mathbb{E}_{\Pi_r}\big[\mathbf{z}^\top \mathbf{H}_T^\star \mathbf{z}\big] \geq \frac{c}{n + c_1 d}\,Tr(\mathbf{H}_T^\star).$$

*Proof.* Let $s(\mathbf{z}) := \nabla \log \Pi_r(\mathbf{z})$ denote the score. On the truncation region,

$$s(\mathbf{z}) = -\nabla V(\mathbf{z}), \qquad \nabla s(\mathbf{z}) = -\nabla^2 V(\mathbf{z}).$$

By the Cramér–Rao matrix inequality for Covariance upper bound, we have an upper bound on the covariance of $\mathbf{z}$ as,

$$\mathrm{Cov}_{\Pi_r}(\mathbf{z}) \succeq \Big(\mathbb{E}_{\Pi_r}\big[\nabla^2 V(\mathbf{z})\big]\Big)^{-1}.$$

Recall on the event $\mathcal{E}_D$, we had

$$\mathbb{E}_{\Pi_r}\big[\nabla^2 V(\mathbf{z})\big] \preceq \left(\frac{n}{4}\left(1 + \sqrt{\frac{d}{n}} + \sqrt{\frac{t}{n}}\right)^2 + d\right)\mathbf{I}_T.$$

Absorbing the factors $\frac{1}{4}$, the $(1 + \sqrt{d/n} + \sqrt{t/n})^2$ term, and the additive $+d$ into absolute constants yields

$$\left(\frac{n}{4}\left(1 + \sqrt{\frac{d}{n}} + \sqrt{\frac{t}{n}}\right)^2 + d\right)\mathbf{I}_T \preceq \frac{1}{c}\left(n + c_1 d\right)\mathbf{I}_T,$$

and hence by Cramér-Rao lower bound, we have

$$\mathrm{Cov}_{\Pi_r}(\mathbf{z}) \succeq \frac{c}{n + c_1 d}\,\mathbf{I}_T.$$

Finally,

$$\mathbb{E}_{\Pi_r}\big[\mathbf{z}\mathbf{z}^\top\big] = \mathrm{Cov}_{\Pi_r}(\mathbf{z}) + \boldsymbol{\mu}\boldsymbol{\mu}^\top \succeq \mathrm{Cov}_{\Pi_r}(\mathbf{z}), \qquad \boldsymbol{\mu} := \mathbb{E}_{\Pi_r}[\mathbf{z}],$$

so for any $\mathbf{H}_T^\star \succeq \mathbf{0}$,

$$\mathbb{E}_{\Pi_r}\big[\mathbf{z}^\top \mathbf{H}_T^\star \mathbf{z}\big] = \mathrm{Tr}\big(\mathbf{H}_T^\star\,\mathbb{E}_{\Pi_r}[\mathbf{z}\mathbf{z}^\top]\big) \geq \mathrm{Tr}\big(\mathbf{H}_T^\star\,\mathrm{Cov}_{\Pi_r}(\mathbf{z})\big) \geq \frac{c}{n + c_1 d}\,\mathrm{Tr}(\mathbf{H}_T^\star). \qquad \square$$

**Lemma F.13.** *Fix radius $r := \min\left\{\frac{1}{2}, \frac{1}{R}, \frac{\log 3}{R_e}\right\}$, on the ball $\{\|\mathbf{z}\|_2 \leq r\}$ on the event $\mathcal{E}_C$, we have*

$$\Pi\big(\|\mathbf{z}\|_2 \leq r \mid D\big) \geq 1 - \exp\left(-c_1\left(\tfrac{1}{3}n(1-\varepsilon)\kappa + \tfrac{12}{25}d\right)\min\left\{\tfrac{1}{4}, \tfrac{1}{R^2}, \tfrac{c_0^2}{R_e^2}\right\}\right).$$

*Proof.* Recall on the event $\mathcal{E}_C$, the posterior is log-concave.

$$\nabla^2 V(\mathbf{z}) \succeq \left(\frac{1}{3}n(1-\varepsilon)\kappa + \frac{12}{25}d\right)\mathbf{I}_T, \qquad \forall\, \mathbf{z} \in T : \|\mathbf{z}\|_2 \leq r,$$

Log-concavity on $\{\|\mathbf{z}\|_2 \leq r\}$ implies Gaussian concentration on this region: there exist absolute constants $c, C > 0$ such that for all $t \geq 0$,

$$\Pi\big(f(\mathbf{z}) \geq t \mid D\big) \ \leq \ C \exp(-c\, m t^2), \qquad m := \tfrac{1}{3}n(1-\varepsilon)\kappa + \tfrac{12}{25}d.$$

Setting $t = r$ yields the stated tail bound, and the lower bound on $\Pi(\|\mathbf{z}\|_2 \leq r \mid D)$ follows by absorbing $C$ into the exponent. $\qquad\square$

**Lemma F.14.** *Assume that for some $r > 0$ and all $\mathbf{z} \in T$ with $\|\mathbf{z}\|_2 \leq r$,*

$$\Delta(\mathbf{z}) := L(\mathbf{w}(\mathbf{z})) - L(\mathbf{w}^\star) \ \geq \ C\,\mathbf{z}^\top \mathbf{H}\,\mathbf{z},$$

*for an absolute constant $C > 0$ and some fixed PSD matrix $\mathbf{H} \succeq \mathbf{0}$. Then*

$$\mathbb{E}_{\Pi(\mathbf{z})}[\Delta(\mathbf{z})] \ \gtrsim \ \Pi(\|\mathbf{z}\|_2 \leq r \mid D)\, \mathbb{E}_{\Pi(\cdot\,|D,\ \|\mathbf{z}\|_2 \leq r)}\big[\mathbf{z}^\top \mathbf{H}\,\mathbf{z}\big].$$

*Proof.* Since $\Delta(\mathbf{z}) \geq 0$, we have

$$\mathbb{E}_{\Pi(\mathbf{z})}[\Delta(\mathbf{z})] \ \geq \ \mathbb{E}_{\Pi(\mathbf{z})}\big[\Delta(\mathbf{z})\,\mathbf{1}_{\{\|\mathbf{z}\|_2 \leq r\}}\big].$$

which means considering the excess risk on the ball $\|\mathbf{z}\|_2 \leq r$ can only decrease it. On the event $\{\|\mathbf{z}\|_2 \leq r\}$, the quadratic lower bound yields

$$\mathbb{E}_{\Pi(\mathbf{z})}\big[\Delta(\mathbf{z})\,\mathbf{1}_{\{\|\mathbf{z}\|_2 \leq r\}}\big] \ \gtrsim \ \mathbb{E}_{\Pi(\mathbf{z})}\big[\mathbf{z}^\top \mathbf{H}\,\mathbf{z}\,\mathbf{1}_{\{\|\mathbf{z}\|_2 \leq r\}}\big].$$

Finally, conditioning on $\{\|\mathbf{z}\|_2 \leq r\}$ gives

$$\mathbb{E}_{\Pi(\mathbf{z})}\big[\mathbf{z}^\top \mathbf{H}\,\mathbf{z}\,\mathbf{1}_{\{\|\mathbf{z}\|_2 \leq r\}}\big] = \Pi(\|\mathbf{z}\|_2 \leq r \mid D)\, \mathbb{E}_{\Pi(\cdot\,|D,\ \|\mathbf{z}\|_2 \leq r)}\big[\mathbf{z}^\top \mathbf{H}\,\mathbf{z}\big],$$

which completes the proof. $\qquad\square$

**Theorem F.15.** *Consider the joint event $\mathcal{E} := \mathcal{E}_A \cap \mathcal{E}_B \cap \mathcal{E}_D$ which occurs with probability $1 - \delta$*

$$\mathcal{E}_{\mathrm{A}} \ := \ \left\{ \max_{1 \leq i \leq n} \|\mathbf{x}_i\|_2 \ \leq \ c\Big(\sqrt{d} + \sqrt{\log(3n/\delta)}\Big) \right\}. \tag{146}$$

$$\mathcal{E}_{\mathrm{B}} \ := \ \left\{ (1-\varepsilon)\,\mathbf{H}_T^\star \ \preceq \ \widehat{\mathbf{H}}_T \ \preceq \ (1+\varepsilon)\,\mathbf{H}_T^\star \right\}, \qquad with \quad \varepsilon := c_1\sqrt{\frac{d + \log(3/\delta)}{n}} \tag{147}$$

$$\mathcal{E}_{\mathrm{D}} \ := \ \left\{ \forall\, \mathbf{z} \in T : \nabla^2 V(\mathbf{z}) \ \preceq \ \left( \frac{n}{4}\Big(1 + \sqrt{\frac{d}{n}} + \sqrt{\frac{1}{n}\log\Big(\frac{6}{\delta}\Big)}\Big)^2 + d\right)\mathbf{I}_T \right\}.$$

*, we have for absolute constant $c_{10} = c_4 \min\left\{\frac{4}{15}, \frac{1}{3}\frac{1}{(1+\frac{1}{R})^2}\right\}$, $R = \frac{\mathbb{E}[\|\mathbf{x}\|_2^3]}{\kappa\,\mathbb{E}[\|\mathbf{x}\|_2^2]}$, $\kappa := \mathbb{E}_{a \sim \mathcal{N}(0,1)}\big[\sigma(a)\big(1 - \sigma(a)\big)\big]$. and $n \gtrsim d + \log(\frac{n}{\delta})$:*

$$\mathbb{E}_{\Pi(\mathbf{z}\cdot|D)}\big[\Delta(\mathbf{z})\big] \ \geq \ \frac{c_{10}(d-1)}{n}. \tag{148}$$

*Proof.* We accumulate all the joint conditions mentioned in the three events $\mathcal{E} := \mathcal{E}_A \cap \mathcal{E}_B \cap \mathcal{E}_D$ each occuring with probability $\delta_1 = \delta_2 = \delta_3 = \frac{\delta}{3}$. Recall from Lemma F.14, we have for $r = \min\left\{\frac{1}{2}, \frac{1}{R}, \frac{c_0}{R_e}\right\}$:

$$\mathbb{E}_{\Pi(\mathbf{z})}[\Delta(\mathbf{z})] \ \geq \ \mathbb{E}_{\Pi(\mathbf{z})}\big[\Delta(\mathbf{z})\,\mathbf{1}_{\{\|\mathbf{z}\|_2 \leq r\}}\big] \ \geq \ \min\left\{\frac{4}{15}, \frac{1}{3}\frac{1}{(1+\frac{1}{R})^2}\right\}\Pi(\|\mathbf{z}\|_2 \leq r \mid D)\,\mathbb{E}_{\Pi(\cdot\,|D,\ \|\mathbf{z}\|_2 \leq r)}\big[\mathbf{z}^\top \mathbf{H}_T^* \mathbf{z}\big] \tag{149}$$

where the last inequality followed from Lemma F.6

From Lemma F.13, we proved that since $\Pi(\|\mathbf{z}\|_2 \leq r \mid D)$ is log-concave restricted in $\|\mathbf{z}\|_2 \leq r$, we have on event $\mathcal{E}_A \cap \mathcal{E}_B$:

$$\Pi\big(\|\mathbf{z}\|_2 \leq r \mid D\big) \geq 1 - \exp\left(-c_1\left(\tfrac{1}{3}n(1-\varepsilon)\kappa + \tfrac{12}{25}d\right)\min\left\{\tfrac{1}{4}, \tfrac{1}{R^2}, \tfrac{c_0^2}{R_e^2}\right\}\right). \tag{150}$$

From Lemma F.12, the truncated second-moment lower bound lemma, there exist absolute constants $c_2 > 0$ and $c_3 > 0$ such that on event $\mathcal{E}_B \cap \mathcal{E}_D$:

$$\mathbb{E}_{\Pi(\cdot|D,\ \|\mathbf{z}\|_2 \leq r)}\big[\mathbf{z}^\top \mathbf{H}_T^\star \mathbf{z}\big] \geq \frac{c_2}{n + c_3 d}\operatorname{tr}(\mathbf{H}_T^\star). \tag{151}$$

Plugging (151) and (150) into (149), we have on the joint event $\mathcal{E}_A \cap \mathcal{E}_B \cap \mathcal{E}_D$ :

$$\mathbb{E}_{\Pi(\mathbf{z})}[\Delta(\mathbf{z})] \geq \min\left\{\frac{4}{15}, \frac{1}{3}\frac{1}{(1+\frac{1}{R})^2}\right\}\left\{1 - \exp\left(-c_1\left(\tfrac{1}{3}n(1-\varepsilon)\kappa + \tfrac{12}{25}d\right)\min\left\{\tfrac{1}{4}, \tfrac{1}{R^2}, \tfrac{\log 3}{R_e^2}\right\}\right)\right\}\frac{c_2}{n + c_3 d}\operatorname{tr}(\mathbf{H}_T^\star) \tag{152}$$

$$\geq \min\left\{\frac{4}{15}, \frac{1}{3}\frac{1}{(1+\frac{1}{R})^2}\right\}\left\{1 - \exp\left(-c_1\left(\tfrac{1}{3}n(1-\varepsilon)\kappa\right)\min\left\{\tfrac{1}{4}, \tfrac{1}{R^2}, \tfrac{\log 3}{R_e^2}\right\}\right)\right\}\frac{c_4(d-1)}{n} \tag{153}$$

Here we used $\operatorname{tr}(\mathbf{H}_T^\star) = d - 1$ and $\tfrac{1}{3}n(1-\varepsilon)\kappa + \tfrac{12}{25}d > \tfrac{1}{3}n(1-\varepsilon)\kappa$ to simplify the lower bound.

Further having $n \geq 4c^2(d + \log(\frac{1}{\delta_1}))$, we get $(1-\epsilon) \geq \frac{1}{2}$.

From Lemma F.7, we had $R_e = c(\sqrt{d} + \sqrt{\log(\frac{n}{\delta_1})})$, we get $\frac{\log 3}{R_e^2} \asymp \frac{1}{d + \log(\frac{n}{\delta_1})}$, so there exists a constant $c_7$, such that $\min\left\{\tfrac{1}{4}, \tfrac{1}{R^2}, \tfrac{\log 3}{R_e^2}\right\} \geq \frac{c_7}{d + \log(\frac{n}{\delta_1})}$. So incorporating these, we have:

$$\mathbb{E}_{\Pi(\mathbf{z})}[\Delta(\mathbf{z})] \geq \min\left\{\frac{4}{15}, \frac{1}{3}\frac{1}{(1+\frac{1}{R})^2}\right\}\left\{1 - \exp\left(-c_1\left(\tfrac{1}{6}n\kappa\right)\frac{c_7}{d + \log(\frac{n}{\delta_1})}\right)\right\}\frac{c_4(d-1)}{n} \tag{154}$$

Furthermore, choosing $n \geq c_8(d + \log(\frac{n}{\delta_1}))$, it is simplified to

$$\mathbb{E}_{\Pi(\mathbf{z})}[\Delta(\mathbf{z})] \geq \min\left\{\frac{4}{15}, \frac{1}{3}\frac{1}{(1+\frac{1}{R})^2}\right\}\left\{1 - \exp\left(-\frac{c_1 c_7 c_8}{6}\right)\right\}\frac{c_4(d-1)}{n} \tag{155}$$

$$\geq \frac{c_{10}(d-1)}{n} \tag{156}$$

for $n \gtrsim d + \log(\frac{n}{\delta})$ and $c_{10} = c_4\min\left\{\frac{4}{15}, \frac{1}{3}\frac{1}{(1+\frac{1}{R})^2}\right\}\left\{1 - \exp\left(-\frac{c_1 c_7 c_8}{6}\right)\right\}$.

$\square$

## F.2. Jacobian expression on tangent

**Lemma F.16** (Jacobian of the normalization chart on the sphere). *Let $\mathbf{w}^\star \in \mathbb{S}^{d-1}$ and let*

$$T := \{\mathbf{z} \in \mathbb{R}^d : \langle \mathbf{z}, \mathbf{w}^\star \rangle = 0\}$$

*be the tangent space at $\mathbf{w}^\star$. Define the chart $\mathbf{w}(\cdot) : T \supset U \to \mathbb{S}^{d-1}$ by*

$$\mathbf{w}(\mathbf{z}) := \frac{\mathbf{w}^\star + \mathbf{z}}{\|\mathbf{w}^\star + \mathbf{z}\|_2},$$

*and let $d\mathbf{z}$ denote Lebesgue measure on $T$ induced by an orthonormal basis. Then the surface measure $d\sigma$ on $\mathbb{S}^{d-1}$ transforms as*

$$d\sigma(\mathbf{w}(\mathbf{z})) = J(\mathbf{z}) \, d\mathbf{z}, \qquad J(\mathbf{z}) = \|\mathbf{w}^\star + \mathbf{z}\|_2^{-d}.$$

*Equivalently, for any integrable $f$ supported in the chart,*

$$\int_{\mathbb{S}^{d-1}} f(\mathbf{w}) \, d\sigma(\mathbf{w}) = \int_T f(\mathbf{w}(\mathbf{z})) \, \|\mathbf{w}^\star + \mathbf{z}\|_2^{-d} \, d\mathbf{z}.$$

*Proof.* By rotational invariance of surface measure, we may assume

$$\mathbf{w}^\star = \mathbf{e}_d = (0, \dots, 0, 1) \in \mathbb{R}^d.$$

Then the tangent space is

$$T = \{(z_1, \dots, z_{d-1}, 0) \in \mathbb{R}^d\}.$$

For $\mathbf{z} = (z_1, \dots, z_{d-1}, 0) \in T$, the normalization chart becomes

$$\mathbf{w}(\mathbf{z}) = \frac{(z_1, \dots, z_{d-1}, 1)}{\rho}, \qquad \rho := \sqrt{1 + \sum_{i=1}^{d-1} z_i^2}.$$

Let $\mathbf{J}_\mathbf{z}$ be the $d \times (d-1)$ Jacobian matrix whose $i$th column is $\partial \mathbf{w}(\mathbf{z})/\partial z_i$. A direct differentiation gives, for $1 \le i \le d-1$,

$$\frac{\partial w_j}{\partial z_i} = \frac{1}{\rho}\delta_{ij} - \frac{z_j z_i}{\rho^3}, \quad 1 \le j \le d-1, \qquad \frac{\partial w_d}{\partial z_i} = -\frac{z_i}{\rho^3}.$$

The induced surface element is $\sqrt{\det(\mathbf{J}_\mathbf{z}^\top \mathbf{J}_\mathbf{z})} \, d\mathbf{z}$. Compute the Gram matrix:

$$\mathbf{J}_\mathbf{z}^\top \mathbf{J}_\mathbf{z} = \frac{1}{\rho^2}\mathbf{I}_{d-1} - \frac{1}{\rho^4}\mathbf{z}_{1:d-1}\mathbf{z}_{1:d-1}^\top = \frac{1}{\rho^2}\left(\mathbf{I}_{d-1} - \frac{1}{\rho^2}\mathbf{z}_{1:d-1}\mathbf{z}_{1:d-1}^\top\right),$$

where $\mathbf{z}_{1:d-1} = (z_1, \dots, z_{d-1})$.

Using $\det(\mathbf{I} - \mathbf{u}\mathbf{u}^\top) = 1 - \|\mathbf{u}\|_2^2$ for rank-one updates,

$$\det(\mathbf{J}_\mathbf{z}^\top \mathbf{J}_\mathbf{z}) = \frac{1}{\rho^{2(d-1)}}\left(1 - \frac{\|\mathbf{z}_{1:d-1}\|_2^2}{\rho^2}\right) = \frac{1}{\rho^{2d}}.$$

Therefore

$$\sqrt{\det(\mathbf{J}_\mathbf{z}^\top \mathbf{J}_\mathbf{z})} = \rho^{-d} = \|\mathbf{w}^\star + \mathbf{z}\|_2^{-d},$$

which proves the claim. $\qquad \square$

