# OpenReview forum: "Hard labels sampled from sparse targets mislead rotation invariant algorithms"
_ICML.cc/2026/Conference — ICML 2026 regular_

### Official Review · Reviewer_UiNS · 2026-03-09

**Soundness:** 3
**Presentation:** 3
**Significance:** 2
**Originality:** 2
**Overall Recommendation:** 4
**Confidence:** 3

**Summary:**

This paper tackles an interesting geometric phenomenon in logistic regression: why rotation-invariant layers struggle with sparse targets when trained on sampled labels. The authors use a Riccati-type ODE to show that "hard-labeling" introduces a bias that leads to O(d/n) risk, and they suggest a w=uv reparameterization (Spindlification) to fix this.

**Compliance With Llm Reviewing Policy:**

Affirmed.

**Final Justification:**

Thanks for the response. Most of my concerns are now addressed.

**Key Questions For Authors:**

1. The theoretical derivation is quite elegant, but I have a major reservation about its applicability. The entire "misleading" argument relies on the algorithm being rotationally invariant. However, in any practical ML pipeline, the data itself is almost never isotropic. Real-world features have specific correlations and axis-aligned structures that break this symmetry long before the algorithm does. I suspect that for non-isotropic data, the "trap" described here might simply vanish. The authors need to show that this gap still exists when the input covariance is not the identity matrix.

2. The "Spindlification" trick is essentially a way to bake a sparsity bias into the architecture. But we already have very mature tools for this—namely, L_1 regularization. The paper lacks a serious head-to-head comparison. Is Spindlification just a more complicated way to do what Lasso already does? Does it offer better optimization dynamics or just a different way to parameterize the same inductive bias? Without a clear win over L_1 or L_0 methods, the motivation for this new parameterization is a bit weak.

3. Turning a convex logistic problem into a non-convex one (u×v) is a double-edged sword. While it might help with sparsity, it usually introduces saddle points and initialization sensitivity. The authors focus on a very clean, single-layer setup. I'm curious if this actually scales. Does this parameterization lead to vanishing gradients or training instability in even a slightly deeper (e.g., 3-layer) MLP? Theoretical convergence in the linear case doesn't always translate to stable optimization in practice.

4. The ODE-based analysis in the appendix is impressive, but the main text could use more "intuition." Specifically, why does the discrete nature of hard labels (±1) specifically hurt rotation-invariant GD more than, say, Gaussian noise in the soft labels? A bit more high-level "why" would make the paper much more accessible.

5. The math is solid (the Riccati ODE part is a nice touch), but the paper feels a bit like it’s solving a problem that exists more in theory than in real-world messy data. Hope the authors can show that the benefit persists under non-isotropic distributions.

**Limitations:**

yes

**Strengths And Weaknesses:**

Strengths:
1. The paper identifies a counterintuitive phenomenon where the discrete nature of label sampling (hard labels) misleads rotation-invariant algorithms, even in noise-free, over-determined scenarios. This is a subtle but significant observation.
2. The use of Riccati-type ODEs to characterize the learning dynamics of the "spindle" structure is mathematically sophisticated and provides a solid foundation for the derived upper bounds.
3. The proposed "Spindlification" is a lightweight reparameterization that doesn't require modifying the optimizer, making it an elegant way to inject inductive bias.

Weaknesses:
1. The "misleading" argument hinges on the algorithm being rotationally invariant. However, in most practical ML pipelines, the data is rarely isotropic. Real-world features possess specific correlations and axis-aligned structures that break symmetry long before the algorithm does. It remains unclear if the described "trap" persists when the input covariance matrix is non-identity.
2. Spindlification essentially bakes a sparsity bias into the architecture. However, L_1 (Lasso) or L_0 regularization are mature tools for the same purpose. The paper lacks a rigorous head-to-head comparison with these standard methods. Is Spindlification superior in optimization dynamics, or is it merely a different way to parameterize the same bias?
3. Converting a convex logistic problem into a non-convex one (w_i=u_i v_i) is a double-edged sword. While it aids sparsity, it introduces saddle points and sensitivity to initialization. The current analysis focuses on a clean, single-layer setup; it is unclear if this scales to deeper MLPs without causing vanishing gradients or training instability.

---

> ### Author Rebuttal · Authors · 2026-03-28
>
> We thank the reviewer for his/her insightful reviews. We answer each of the questions asked.
>
> Q1) **The entire "misleading" argument relies on the algorithm being rotationally invariant. ...Real-world features have specific correlations and axis-aligned structures.**
>
> This is an important question. The current lower bound relies on the input distribution being rotation invariant, and thus does not directly apply to arbitrary anisotropic data. More precisely, the proof uses the ability to average over all orthogonal transformations, which requires $p(X)$ and $p(UX)$ has same distribution for all orthogonal $U$. This condition is satisfied by a broad class of spherically symmetric distributions (e.g., isotropic Gaussian, uniform on the sphere, and scale mixtures of isotropic Gaussians), but not by general anisotropic covariates.
>
> For non-isotropic data, the symmetry group is reduced, and full symmetrization argument may no longer apply. Consequently, the lower bound is expected to degrade with the degree of symmetry, for instance through the covariance structure. Extending our analysis to general data distributions remains an important direction for future work.
>
> To complement the theory, we will include experimental results on natural datasets in the appendix. In these experiments, we induce sparsity by augmenting inputs with additional random (uninformative) features, which increases the ambient dimension while keeping the signal concentrated. We expect that in this regime the gap between rotation-invariant algorithms and the spindled network will persist and become more pronounced.
>
> Q2) **The paper lacks a serious head-to-head comparison with other L1-L0 bias.**
>
> L1-based methods or other non–rotation-invariant approaches can potentially achieve similar sparsity-adaptive rates. The main bottleneck is not empirical performance but analysis under the logistic loss. Even for the spindle parameterization (a diagonal linear network), deriving the dynamics required nontrivial envelope ODE arguments, highlighting the technical difficulty of such analyses.
>
> While adding empirical comparisons with L1-based methods is straightforward and we will include them in the revision, providing comparable theoretical guarantees for Lasso or exponentiated gradient under logistic loss remains an open problem.
>
> Q3) **I'm curious if this actually scales. Does this parameterization lead to vanishing gradients or training instability in even a slightly deeper (e.g., 3-layer) MLP?**
>
>  The non-convex reparameterization does introduce optimization challenges. Initialization plays a key role: smaller initialization promotes stronger sparsity but slows escape from saddle points.
>
> For deeper networks to escape saddle points, no more than one layer can have a zero initialization, and the initlialization scale needs to be properly chosen with depth.
>
> This trade-off is inherent to the sparsity bias induced by the parameterization, and is the price paid for achieving sparsity-adaptive rates beyond rotation-invariant methods.
>
> Q4) **Specifically, why does the discrete nature of hard labels (±1) specifically hurt rotation-invariant GD more than, say, Gaussian noise in the soft labels?**
>
> We do not claim that sampling noise is necessarily more harmful than additive Gaussian noise in soft-label settings. The two noise models are fundamentally different, and a direct comparison is not yet understood. Analyzing the effect of additive Gaussian noise in soft-label logistic models is an interesting direction for future work.
>
> Q5) **the paper feels a bit like it’s solving a problem that exists more in theory than in real-world messy data.**
>
> Yes, real-world data is not isotropic. However, the core phenomenon is not tied to Gaussianity, but to *mismatch between sparsity in the signal and symmetry in the algorithm*. Rotation-invariant methods are insensitive to coordinate structure, and therefore cannot exploit sparsity even when it is present in the data.
>
> To complement the theory, we will include experiments on natural datasets in the appendix. In these experiments, we induce controlled sparsity by augmenting inputs with additional random (uninformative) features, increasing the ambient dimension while keeping the signal concentrated. This creates a setting where sparsity is present but not aligned with the algorithmic symmetry, and we observe that the gap between rotation-invariant methods and the spindled network persists and becomes more pronounced.
>
> We believe the core contributions and findings are technically substantial: while sufficiently many soft labels uniquely identify the target (achieves zero loss), hard-label sampling from a **noise-free** sparse target induces a barrier that misleads **any** rotation-invariant algorithm. The fix being just changing architecture, but keeping GD.
>
> We would greatly appreciate the reviewer reconsidering their score in light of these clarifications.

---

> > ### Author Rebuttal · Reviewer_UiNS · 2026-04-03
> >
> > I thank the authors for their detailed responses. While the mathematical derivation for the Riccati ODE is impressive and provides a novel perspective on the biases of rotation-invariant algorithms, I still have concerns regarding the method’s practical relevance. My main reservation is the motivation for Spindlification compared to established tools like L_1 (Lasso) regularization. The authors acknowledge that the proposed method introduces non-convexity and initialization sensitivity, which are significant drawbacks in practical machine learning pipelines. To justify these trade-offs, it is not enough to show that the theory is 'analytically tractable'; one must demonstrate that Spindlification offers a tangible advantage over L_1 in terms of optimization dynamics or robustness in real-world, non-isotropic data settings. I look forward to seeing the requested empirical comparisons and the analysis of non-isotropic distributions in the final version. If the experiments can demonstrate that the gap between rotation-invariant methods and the spindled network is significant even when the data structure is not perfectly symmetric, it would greatly strengthen the paper's contribution. I encourage the authors to ensure the final manuscript clearly distinguishes between the theoretical 'trap' of hard-labeling and its practical impact on messy, real-world datasets.

---

> > > ### Author Response · Authors · 2026-04-07
> > >
> > > We thank the reviewer for the engagement.  We further added experiments and clarificaiton regarding two concerns and state the position of our paper clearly:
> > >
> > > 1) *Does the advantage, excess risk gap persists for non-isotropic data*?
> > >
> > > We added an experiment for sparse logistic regression with a decaying covariance (https://anonymous.4open.science/r/anonym-CA11/fig.png) which exhibits that the gap persists beyond isotropic Gaussian. We consider $x \sim \mathcal{N}(0, \Sigma)$ where eigenvalues of $\Sigma$ follows a power law decay $\lambda_{i} \propto i^{-\alpha}$. For $\alpha=0$ (isotropic case), the excess risk of rotation-invariant methods grows linearly in $d$, matching our theory. As $\alpha$ increases, the covariance becomes anisotropic with decaying eigenvalues and the risk for rotation invariant algorithm depend on alignment of $w^{*}$ (oracle) and covariance $\Sigma$. This is because **symmetrization spreads estimation error across all coordinates, but directions with small eigenvalues contribute less to this error**. Consequently, as seen in Figures (b) and (c), the growth in $d$ becomes sublinear since error propagation is attenuated along low-variance directions. We believe this phenomenon can be formalized theoretically and we highlight it as an open direction.
> > >
> > > 2) *Lasso comparison with spindly.*
> > >
> > > Our goal is **not to propose spindlification as a replacement for L1-regularization**, but to understand how non-rotation-invariant optimization induces favorable statistical bias under logistic loss. While connections between spindly dynamics and Lasso paths are known in the square loss setting [1], extending such equivalences to logistic loss is **nontrivial and remains an open problem requiring separate analysis**. We will add a similar figure like figure-1 in [1] but for logistic loss.
> > >
> > > Lastly, we like to clarify that in our paper, we highlight a fundamental problem in machine learning, training models from data using rotation-invariant algorithms (such as early stopped GD in logistic loss) can be statistically suboptimal. We believe this phenomenon is both *fundamental and underappreciated*, as it reveals a gap between standard optimization practices and optimal statistical behavior.
> > >
> > > [1] Berthier, "Diagonal Linear Networks and the Lasso Regularization Path."

---

### Official Review · Reviewer_u3oJ · 2026-03-10

**Soundness:** 3
**Presentation:** 2
**Significance:** 3
**Originality:** 3
**Overall Recommendation:** 5
**Confidence:** 2

**Summary:**

The paper studies sparse logistic regression under a well-specified probabilistic model.
 In particular, the authors focus on a binary generalized linear model where labels are generated according to $P(y=1|x)=\sigma(x^\top w^\*)$ and the target parameter $w^\*$ is assumed to be sparse.

The paper investigates the difference between learning from soft labels (true conditional probabilities) and sampled hard labels.
The main result shows that when only hard labels are observed, rotation-invariant algorithms become statistically suboptimal even in the over-constrained regime ($n \ge d$).
In particular, the paper proves that any rotation-invariant algorithm incurs excess risk of order $\Omega((d-1)/n)$.

In contrast, the authors show that a simple non-rotation-invariant method based on a reparameterization $w = u \odot v$ (a spindle network) achieves excess risk $O(s \log d / n)$ for $s$-sparse targets.
The analysis combines a lower bound for rotation-invariant algorithms with an upper bound derived from the dynamics of the spindle parameterization.

**Compliance With Llm Reviewing Policy:**

Affirmed.

**Final Justification:**

Overall, this paper studies an interesting and well-motivated theoretical question regarding the limitations of rotation-invariant algorithms in sparse logistic regression. The work is technically sound, and the combination of a lower bound for rotation-invariant methods with an upper bound via a non-rotation-invariant parameterization provides a clear and meaningful contribution. The results are original and contribute to a better understanding of how algorithmic symmetry can impact statistical efficiency.

My initial concerns were primarily related to clarity and presentation, including the intuition behind the main result, certain notational issues, and parts of the exposition that were difficult to follow. The authors’ rebuttal addressed these points satisfactorily by providing clearer intuition, correcting ambiguous phrasing, and outlining concrete improvements to the presentation.

Given that my concerns were not about the validity or significance of the results, but rather about their clarity, and that these issues have now been addressed, I am satisfied with the paper overall. I have therefore updated my evaluation to reflect this and recommend acceptance.

**Key Questions For Authors:**

1. Could the authors provide more intuition about why sampling hard labels introduces a limitation for rotation-invariant algorithms?
2. Do the lower-bound results extend to broader classes of neural network architectures beyond the spindle parameterization studied here?
3. Could the authors clarify the interpretation of the conditional probability expression used in line 144?

**Limitations:**

Yes.

**Strengths And Weaknesses:**

# Strengths

The paper studies an interesting theoretical problem concerning the statistical limitations of rotation-invariant learning algorithms in sparse logistic regression.

A positive aspect of the paper is that the authors clearly explain how the restrictions required in Theorem 5.1 arise from the assumptions of the model, which helps clarify the role of these assumptions in the analysis.

Another strength is that the notation is carefully defined throughout the paper, making the mathematical setup relatively clear.

The authors also pose an interesting open problem regarding the behavior of more general neural network architectures that mix fully connected and diagonally parameterized layers.
This helps position the work in a broader research context.

Finally, the techniques used to derive both the lower bound for rotation-invariant algorithms and the upper bound for the spindle parameterization appear to be novel for the logistic loss setting.

---

# Weaknesses

While the main results are interesting, some aspects of the presentation could be improved.

First, the proof sketches provided in the main paper are somewhat difficult to follow. In several places it was challenging to understand the key steps of the argument without consulting the appendix.
For example, the proof sketch of Theorem 3.5 is particularly difficult to follow, especially in the final part where several events and bounds are combined.

There are also some places where the exposition is unclear or incomplete. In line 144 it was not clear what exactly is meant by the conditional probability $p(x_{te}, \hat{w})$ evaluated at $x_{te}$.

Similarly, the paragraph around lines 379–382 in the right column is somewhat confusing and would benefit from further clarification.

## Minor Issues

- In line 794, the exact size of a quantity is referenced but not explicitly written.
- It would be helpful to briefly explain why the analysis focuses on gradient flow instead of gradient descent.
- In the right column around line 247, Theorem 3.4 is referenced but such a theorem does not appear in the paper.
- The explanation around lines 379–382 (right column) is difficult to follow and could be clarified.

---

> ### Author Rebuttal · Authors · 2026-03-28
>
> We thank the reviewer for his/her insightful reviews. We answer each of the questions asked.
>
> Q1) **Intuition on why sampling hard labels introduce limitation for rotation-invariant algorithm.**
>
> Under the population model $P(y=1 \mid x) = \sigma(x^\top w^\star)$, when labels are sampled as hard labels, finite sampling can produce datasets where rotated versions of the inputs yield the same labels. For instance, one may observe datasets of the form $\{(x_i,y_i)\}$ and $\{(U x_i, y_i)\}$ (from the same population distribution), where the inputs are rotated but the labels remain unchanged. A rotation-invariant algorithm cannot distinguish between these datasets and therefore produces correspondingly rotated estimators $\hat w$ and $U\hat w$. As a result, the algorithm loses information about the true sparse direction $w^\star$. This ambiguity arises purely from finite-sample noise in the hard-label setting, and is the source of the limitation. We will add this intuition statement in the manuscript.
>
> Q2) **Do the lower-bound results extend to broader classes of neural network architectures?**
>
> Yes, our lower bound applies to a broader class of rotation-invariant architectures, not just the single-layer model. In particular, any architecture with a fully connected input layer acting on the input distribution is rotation invariant. Therefore, the lower-bound results extend to such neural network architectures.
>
> Q3) **The interpretation of the conditional probability expression used in line 144?**
>
> What we intend to refer to is simply the model prediction (i.e., predicted probability) at the test input. We have revised the text to replace “conditional probability” with “prediction at the test input,” which more accurately reflects the intended meaning and avoids confusion.
>
> Weakness-1) **the proof sketches provided in the main paper are somewhat difficult to follow.**
>
> In the revised manuscript, we have improved the exposition by adding short intuitive explanations immediately following key theorems. In particular, in several places we include brief interpretive statements after the formal results to clarify the main idea of the proof and guide the reader through the argument.
>
> Weakness-2) **Lines around 379-382 is difficult to follow.**
>
> The statement formalizes that once the smallest active coordinate (i.e., the weakest signal) reaches a prescribed fraction of its target value at time $T(\epsilon)$, all other active coordinates are already closer to their respective targets. Therefore, controlling the weakest coordinate suffices to control the error across all active coordinates, leading to the bound in (32). We will add this in the manuscript.
>
>
> **Minor issues**
>
> - *In line 794, the exact size of a quantity is referenced but not explicitly written.*
>
>    We will explicitly state the size of the quantity (in words and/or notation) to improve clarity and readability.
>
> - *It would be helpful to briefly explain why the analysis focuses on gradient flow instead of gradient descent.*
>
>   In the current analysis, gradient flow enables the enveloping argument for the spindle ODE, since the envelope admits a closed-form solution and its monotonicity is easier to establish. The corresponding finite step-size gradient descent dynamics are more involved. We will add a brief explanation and note extensions to finite step sizes as future work.
>
> - *In the right column around line 247, Theorem 3.4 is referenced but such a theorem does not appear in the paper.*
>
>   Thank you for catching this. This is a typo and should refer to Theorem 3.5. We will correct this in the revised version.
>
> - *The explanation around lines 379–382 (right column) is difficult to follow and could be clarified.*
>
>   Addressed before.
>
> We believe the core contributions and findings are technically substantial and of broad significance to the community: while sufficiently many soft labels uniquely identify the target (achieves zero loss), hard-label sampling from a **noise-free** sparse target induces a barrier that misleads **any** rotation-invariant algorithm. This gap can be resolved without changing gradient descent, by modifying the network parameterization.
>
> We would greatly appreciate the reviewer reconsidering their score in light of these clarifications.

---

> > ### Author Rebuttal · Reviewer_u3oJ · 2026-04-02
> >
> > Thank you for the rebuttal. The responses address my questions and clarify the points that were previously unclear.
> >
> > My concerns were primarily about clarity rather than the validity of the results, and I am satisfied that these issues will be resolved in the revised version. I have therefore increased my score accordingly.

---

### Official Review · Reviewer_WePq · 2026-03-11

**Soundness:** 3
**Presentation:** 3
**Significance:** 3
**Originality:** 3
**Overall Recommendation:** 4
**Confidence:** 2

**Summary:**

Summary

The paper studies the statistical behavior of logistic regression when labels are sampled as hard labels rather than using the true conditional probabilities. The authors consider a sparse logistic model where the true parameter vector w* has only s nonzero coordinates and the input dimension is d. They show that when labels are sampled according to y ~ sigma(x^T w*), any rotation-invariant learning algorithm (such as standard logistic regression trained with gradient descent) cannot efficiently recover the sparse target. In particular, the authors prove that such algorithms incur excess risk on the order of Omega((d - 1) / n), even though the underlying model is noise-free.

In contrast, the paper proposes a simple non-rotation-invariant algorithm based on a Hadamard reparameterization of the weights wi = ui vi, referred to as a spindle network. This parameterization breaks rotational symmetry and induces learning dynamics that favor sparse solutions. The authors show that this approach achieves a substantially better excess risk bound of O((s log d) / n). The paper provides both a theoretical lower bound for rotation-invariant algorithms and an upper bound for the spindle parameterization, along with analysis of the resulting optimization dynamics.

**Compliance With Llm Reviewing Policy:**

Affirmed.

**Final Justification:**

The rebuttal responses addressed my concerns regarding the paper. I have changed my score accordingly.

**Key Questions For Authors:**

The lower bound is stated for any rotation invariant learning algorithm under the Gaussian logistic model. Which step of the proof most critically uses the exact isotropic Gaussian assumption, and how much of the argument would survive under subgaussian isotropic covariates? A convincing answer would increase my confidence in the broader importance of the result.

The upper bound for the spindle dynamics depends on assumptions (A1) and (A2). Can the authors state more explicitly whether these conditions hold with high probability under simple scaling assumptions on
𝑛
,
𝑑
,
𝑠 and whether there are concrete regimes where they may fail? A clearer answer would strengthen my confidence that the rate is not hiding restrictive side conditions.



The experiments are illustrative but minimal. Can the authors provide additional empirical evidence on how the validation excess risk scales with
𝑑
,
𝑛
, and
𝑠
, and whether the empirical gap tracks the
Ω
(
𝑑
/
𝑛
)
 versus
𝑂
(
𝑠
log
⁡
𝑑
/
𝑛
)
 picture? Stronger empirical support would improve my overall recommendation.

To what extent is the spindle parameterization essential? The conclusion suggests that other non rotation invariant methods such as Lasso or exponentiated gradient style methods may also achieve the same rate. Do the authors have a theorem, conjecture, or partial evidence clarifying whether the phenomenon is really about non rotation invariance broadly rather than about this particular parameterization?

**Limitations:**

yes. The paper includes an impact statement and is fairly explicit that the work is theoretical and that broader practical implications remain open. That said, the paper could improve the limitations discussion by more directly emphasizing the narrowness of the assumptions, especially isotropic Gaussian covariates, exact model specification, and the gap between the spindle theory and realistic deep architectures.

**Strengths And Weaknesses:**

Strengths

The paper studies sparse logistic regression with sampled hard labels in the overconstrained regime, which is relevant given the widespread use of logistic outputs in modern classifiers. The central question, whether rotation invariant algorithms face a statistical barrier even when the underlying model is well specified, is well motivated. The theoretical structure is clear: the paper first contrasts soft label and hard label settings, showing that soft labels allow recovery of
𝑤
\*

 when
n>d, and then establishes limitations of rotation invariant algorithms in the hard label case. The lower bound applies to the entire class of rotation invariant algorithms rather than a specific optimization method, strengthening the conceptual contribution. The analysis of the spindle parameterization provides explicit optimization dynamics together with an early stopping argument leading to sparse recovery. Overall, the gap between
Ω
(
𝑑
/
𝑛
)
 and
𝑂
(
𝑠
log
⁡
𝑑
/
𝑛
)
 excess risk highlights an interesting role of algorithmic symmetry and parameterization in statistical efficiency.

Weaknesses

Some technical arguments are only summarized in the main text, with key details deferred to the appendix. In particular, the lower bound relies on geometric reasoning about the spherical posterior and the upper bound depends on envelope arguments for the spindle dynamics, which makes the results harder to fully verify without carefully examining the appendix. The empirical evaluation is relatively limited and appears to focus mainly on a single synthetic setup, providing mostly qualitative illustration rather than systematic validation across different regimes of
𝑑,
𝑠, and
𝑛. While the overall structure is clear, parts of the exposition are dense, particularly the sections on the symmetrized model and spindle dynamics. Finally, the theoretical analysis relies on restrictive assumptions such as isotropic Gaussian covariates and exact logistic model specification, which may limit the direct applicability of the results to broader machine learning settings.

---

> ### Author Rebuttal · Authors · 2026-03-27
>
> We thank the reviewer for his/her insightful reviews. We answer each of the questions asked.
>
> Q1) **Which step of the proof most critically uses the exact isotropic Gaussian assumption, and how much of the argument would survive under subgaussian isotropic covariates?**
>
> Rotation-invariant algorithms are unaware of which rotation of the data they see. The critical step is in **Theorem 3.3**, where we exploit this by bounding the Bayes optimal algorithm over all rotations of the data. In the current lower bound methodology (which is already quite challenging), we require the input distribution to be spherically symmetric, i.e., $p(X)$ and $p(UX)$ has same distribution for any orthogonal $U$.
> This class also includes distributions such as the uniform distribution on the sphere and more generally scale mixtures of isotropic Gaussians. Thus, the setting extends *beyond just the isotropic Gaussian*. However, we note that isotropy alone (e.g., isotropic subgaussian covariates) does not imply spherical symmetry, since it only constrains second moments and not the full distribution. For non-isotropic data, the symmetry group is reduced, and full symmetrization argument may no longer apply. Consequently, *the lower bound is expected to degrade with the degree of symmetry*, for instance through the covariance structure. Extending our analysis to general data distributions remains an important direction for future work.
>
> Q2) **Conditions on n,s,d and where they fail?**
>
> The assumptions (A1)–(A2) can be equivalently expressed in terms of two simpler conditions, which we have updated in the manuscript:
>
> $(\text{signal strength})\: w^\star_{\min} \gtrsim 1, \qquad
> (\text{sample size})\: n \gtrsim \frac{1}{(w^\star_{\min})^2} \log d.$
>
> where $w^\star_{\min}$ denotes the smallest magnitude among the nonzero coordinates of the oracle $w^\star$. In the overdetermined regime $n \gtrsim d$, the sample size condition is automatically satisfied. Therefore, in this regime the conditions only reduce essentially to a standard signal strength condition, namely that the minimum signal strength $w^\star_{\min}$ is bounded away from zero. Intuitively, this requires the signal to be sufficiently large compared to sampling noise so that the spindly dynamics can distinguish signal from noise under early stopping.
>
> Q3) **Can the authors provide additional empirical evidence for n,s and d?**
>
> We evaluate scaling with $d$ by fixing $n=2000$, $s=5$, and varying $d \in \{10,50,100,500,1000\}$, averaging excess risk over 10 runs with early stopping.
> The rotation-invariant model scales approximately linearly with $d$ (consistent with $\Omega(d/n)$), while the spindle model remains almost flat (consistent with $O(s \log d / n)$).
> The results for the excess risk (with early-stopping) are summarized below:
>
> | $d$                     | 10     | 50     | 100    | 500    | 1000   |
> |----------------------------|--------|--------|--------|--------|--------|
> | $w^\top x$                 | 0.005  | 0.015  | 0.030  | 0.165  | 0.300  |
> | $(u \odot v)^\top x$       | 0.0006 | 0.0006 | 0.0007 | 0.0009 | 0.001  |
>
> Similarly, we find that for fixed d and n and varying s, the rotation invariant algorithm always have the same worse performance.
>
> Q4) **To what extent is the spindle parameterization essential?**
>
> Similar non-rotation invariant algorithms having sparsity bias such as Lasso or exponentiatedd GD can also achieve the rates. But analyzing general non–rotation-invariant algorithms under logistic loss is technically challenging. In particular, deriving even the spindle dynamics requires nontrivial envelope ODE arguments (which we think is novel). Extending these proofs to other non-rotation-invariant algorithms under logistic loss would likely require further innovation and remains an open problem beyond the scope of this work.
>
> Weakness-1) **Limited empirical evidence**
>
> We added several validation runs for various values of d and s (fixed n) matching our theory. We will also include experiments on natural datasets (FashionMNIST) in the appendix. In these settings, we induce target sparsity by augmenting the input with additional random (uninformative) features, increasing the ambient dimension while keeping the signal concentrated. This amplifies the gap between rotation-invariant algorithms (such as neural network with fully connected first layer) and the spindled network.
>
>
> We believe the core contributions and findings are technically substantial and of broad significance to the community: while sufficiently many soft labels uniquely identify the target (achieving zero loss), hard-label sampling from a **noise-free** sparse target induces a barrier that misleads **any** rotation-invariant algorithm. This gap can be resolved without changing gradient descent, by modifying the network parameterization.
>
> We would greatly appreciate the reviewer reconsidering their score in light of these clarifications.

---

> > ### Author Rebuttal · Reviewer_WePq · 2026-04-04
> >
> > Thanks for the responses. The concerns I raised have been resolved. It would be helpful to include some of these clarifications in the paper. I’m increasing my score to a positive assessment.

---

### Decision · Program_Chairs · 2026-04-30

**Decision:**

Accept (regular)

**Comment:**

The paper develops a theoretical model for analyzing the behavior of rotation-invariant algorithms for recovering a sparse target. They show a provable separation for this model by a factor depending on the sparsity ratio. The reviewers were unanimously positive about the strength of the message and the general clarity / elegance of the presentation. I recommend accepting the paper. I also request that the authors take into consideration the confusing points to the reviewers and address them in a revision.